# Unlocking TriLevel Learning with Level-Wise Zeroth Order Constraints: Distributed Algorithms and Provable Non-Asymptotic Convergence

## Abstract

Trilevel learning (TLL) found diverse applications in numerous machine learning applications, ranging from robust hyperparameter optimization to domain adaptation. However, existing researches primarily focus on scenarios where TLL can be addressed with first order information available at each level, which is inadequate in many situations involving zeroth order constraints, such as when black-box models are employed. Moreover, in trilevel learning, data may be distributed across various nodes, necessitating strategies to address TLL problems without centralizing data on servers to uphold data privacy. To this end, an effective distributed trilevel zeroth order learning framework DTZO is proposed in this work to address the TLL problems with level-wise zeroth order constraints in a distributed manner. The proposed DTZO is versatile and can be adapted to a wide range of (grey-box) TLL problems with partial zeroth order constraints. In DTZO, the cascaded polynomial approximation can be constructed without relying on gradients or sub-gradients, leveraging a novel cut, i.e., zeroth order cut. Furthermore, we theoretically carry out the non-asymptotic convergence rate analysis for the proposed DTZO in achieving the $\epsilon$-stationary point. Extensive experiments have been conducted to demonstrate and validate the superior performance of the proposed DTZO, e.g., it approximately achieves up to a 40% improvement in performance.

## 1 Introduction

Trilevel learning (TLL), also known as trilevel optimization, pertains to nested optimization problems involving three levels of optimization, thus exhibiting a trilevel hierarchical structure. Trilevel learning has been widely used in many machine learning applications, such as robust hyperparameter optimization (Sato et al., 2021), domain adaptation (Choe et al., 2023), robust neural architecture search (Guo et al., 2020; Jiao et al., 2024), and so on. The general form of a trilevel learning problem can be expressed as,

$$
\begin{aligned}
\min \quad & f_1(\boldsymbol{x}_1, \boldsymbol{x}_2, \boldsymbol{x}_3) \\
\text{s.t.} \quad & \boldsymbol{x}_2 = \arg\min_{\boldsymbol{x}_2'} f_2(\boldsymbol{x}_1, \boldsymbol{x}_2', \boldsymbol{x}_3) \\
& \quad \text{s.t.} \quad \boldsymbol{x}_3 = \arg\min_{\boldsymbol{x}_3'} f_3(\boldsymbol{x}_1, \boldsymbol{x}_2', \boldsymbol{x}_3') \\
\text{var.} \quad & \boldsymbol{x}_1, \boldsymbol{x}_2, \boldsymbol{x}_3,
\end{aligned}
\tag{1}
$$

where $f_1, f_2, f_3$ denote the first, second, and third level objectives, and $\boldsymbol{x}_1 \in \mathbb{R}^{d_1}, \boldsymbol{x}_2 \in \mathbb{R}^{d_2}, \boldsymbol{x}_3 \in \mathbb{R}^{d_3}$ are variables. Existing trilevel learning approaches focus on scenarios where TLL problems can be addressed with first order information available at each level. However, situations where first order information is unavailable (i.e., $\nabla f_1, \nabla f_2, \nabla f_3$ are non-available), such as when black-box models are employed, remain *under-explored*. Additionally, in trilevel learning applications, data may be distributed across various nodes, necessitating strategies to address trilevel learning problems without centralizing data on servers in order to uphold data privacy (Jiao et al., 2024).

**Complexity of Addressing TLL with Zeroth Order Constraints:** The complexity involved in solving problems characterized by hierarchical structures with three levels is *significantly greater* than that of bilevel learning problems (Blair, 1992; Avraamidou, 2018). It is worth mentioning that

even *finding a feasible solution* in TLL problem is **NP-hard** since it necessitates addressing the inner bilevel learning problem, which is NP-hard (Ben-Ayed and Blair, 1990; Sinha et al., 2017). Existing approaches are not applicable for addressing TLL with zeroth order constraints, as they either rely on the first order information to solve the TLL problems (Jiao et al., 2024; Sato et al., 2021) or focus on single-level and bilevel zeroth order learning problems (Fang et al., 2022; Qiu et al., 2023).

To this end, an effective **D**istributed **T**rilevel **Z**eroth **O**rder learning (DTZO) framework is proposed in this work. Specifically, we first introduce the cascaded zeroth order polynomial approximation for the trilevel learning problems, which consists of the inner layer and outer layer polynomial approximation. Next, how to generate the novel zeroth order cuts without using gradients or sub-gradients to gradually refine the cascaded polynomial approximation is discussed. Zeroth order cut is a type of cutting plane that does not rely on first order information during generation. Finally, the distributed zeroth order algorithm is developed to address trilevel zeroth order learning problems (i.e., TLL with level-wise zeroth order constraints) in a distributed manner. Theoretically, we demonstrate that the proposed zeroth order cuts can construct a polynomial relaxation for TLL problems, and this relaxation will be gradually tightened with zeroth order cuts added. Additionally, we also analyze the non-asymptotic convergence rate, i.e., iteration and communication complexities, for the proposed DTZO to achieve the $\epsilon$-stationary point. The contributions of this work are summarized as follows.

**1.** Different from the existing works on single-level and bilevel zeroth order learning, this work takes an initial step towards addressing trilevel zeroth order learning. To the best of our knowledge, this is the first work to address the trilevel zeroth order learning problems.

**2.** An effective framework DTZO with novel zeroth order cuts is proposed for tackling trilevel zeroth order learning problems in a distributed manner. Different from the existing methods, the proposed DTZO is capable of constructing the cascaded zeroth order polynomial approximation without using gradients or sub-gradients.

**3.** Extensive experiments on black-box large language models (LLMs) trilevel learning and robust hyperparameter optimization substantiate the superior performance of the proposed DTZO.

## 2 RELATED WORK

### 2.1 DISTRIBUTED ZEROTH ORDER OPTIMIZATION

Zeroth order optimization is widely-used for addressing machine learning problems where obtaining explicit gradient expressions is challenging or impractical (Liu et al., 2018c; Chen et al., 2019; Wang et al., 2018b; Chen et al., 2017; Héliou et al., 2021; Cai et al., 2021; Gao and Huang, 2020; Yue et al., 2023; Li et al., 2022; Ren et al., 2023; Nikolakakis et al., 2022; Tu et al., 2019; Rando et al., 2024). In practical applications of zeroth order optimization, data may be distributed across different nodes. To address zeroth order optimization problems in a distributed manner, the distributed zeroth order optimization methods have recently garnered significant attention, e.g., Lian et al. (2016); Tang et al. (2020); Fang et al. (2022); Chen et al. (2024a); Akhavan et al. (2021); Sahu et al. (2018); Shu et al. (2023). Furthermore, to tackle the bilevel zeroth order optimization problems in a distributed manner, the federated bilevel zeroth order optimization method FedRZO$_{bl}$ (Qiu et al., 2023) has been proposed. However, how to address the higher-nested zeroth order optimization problems, e.g., trilevel, in a distributed manner remains under-explored. To the best of our knowledge, this is the **first work** that considers how to address the trilevel zeroth order optimization problems.

### 2.2 TRILEVEL LEARNING

Trilevel learning has found applications in various fields within machine learning. A robust neural architecture search (NAS) approach that integrates adversarial learning with NAS is introduced in Guo et al. (2020). The robust NAS can be viewed as a trilevel learning problem, as discussed in Jiao et al. (2024). A trilevel learning problem comprising two levels pretraining, fine-tuning and hyperparameter optimization, is explored in Raghu et al. (2021). In Garg et al. (2022), the trilevel learning problem involves data reweight, architecture search, and model training is investigated. In Sato et al. (2021), the robust hyperparameter optimization is framed as a trilevel learning problem, and a hypergradient-based method is proposed to address such problems. In Choe et al. (2023), a general automatic differentiation technique is proposed, which can be applied to trilevel learning

problems. Additionally, a cutting plane based distributed algorithm is proposed in Jiao et al. (2024) for trilevel learning problems. Nevertheless, existing methods predominantly rely on first order information to solve trilevel learning problems. This is the **first framework** that can be used to solve trilevel learning problems *without* relying on first order information.

## 2.3 CUTTING PLANE METHOD

Cutting plane methods are widely used in convex optimization (Bertsekas, 2015; Franc et al., 2011), robust optimization (Yang et al., 2014; Bürger et al., 2013), and so on. Recently, there has been notable interest in leveraging cutting plane methods to tackle distributed nested optimization problems. It is shown in Jiao et al. (2023) that the nested optimization problem can be transformed into a *decomposable optimization problem* by utilizing cutting plane method, which *significantly facilitates* the design of distributed algorithms for nested optimization. In Jiao et al. (2023), the cutting plane method is employed to tackle bilevel optimization problems in a distributed manner. Similarly, Chen et al. (2024c) utilizes the cutting plane method to address distributed bilevel optimization problems within downlink multi-cell systems. Furthermore, Jiao et al. (2024) applies the cutting plane method to solve distributed trilevel optimization problems. However, the existing cutting plane methods for nested optimization rely on the gradients or the sub-gradients to generate cutting planes, which is not available in zeroth order optimization. In this work, the proposed framework is capable of generating zeroth order cuts for nested optimization problems without using gradients or sub-gradients.

# 3 DISTRIBUTED TRILEVEL ZEROTH ORDER LEARNING

In the practical applications of trilevel zeroth order learning, data may be distributed across multiple nodes (Jiao et al., 2024). Aggregating data on central servers may pose significant privacy risks (Subramanya and Riggio, 2021). Therefore, it is crucial to develop an effective framework to address trilevel zeroth order learning problems in a distributed manner. The distributed trilevel zeroth order learning problem can be expressed as,

$$
\begin{aligned}
\min \ & \sum_{j=1}^{N} f_{1,j}(\boldsymbol{x}_1, \boldsymbol{x}_2, \boldsymbol{x}_3) \\
\text{s.t. } & \boldsymbol{x}_2 = \arg\min_{\boldsymbol{x}_2'} \sum_{j=1}^{N} f_{2,j}(\boldsymbol{x}_1, \boldsymbol{x}_2', \boldsymbol{x}_3) \\
& \quad \text{s.t. } \boldsymbol{x}_3 = \arg\min_{\boldsymbol{x}_3'} \sum_{j=1}^{N} f_{3,j}(\boldsymbol{x}_1, \boldsymbol{x}_2', \boldsymbol{x}_3') \\
\text{var. } & \quad \boldsymbol{x}_1, \boldsymbol{x}_2, \boldsymbol{x}_3,
\end{aligned}
\tag{2}
$$

where $f_{1,j}, f_{2,j}, f_{3,j}$ respectively denote the first, second, and third level objectives in $j^{\text{th}}$ worker, $\boldsymbol{x}_1 \in \mathbb{R}^{d_1}, \boldsymbol{x}_2 \in \mathbb{R}^{d_2}, \boldsymbol{x}_3 \in \mathbb{R}^{d_3}$ are variables. The first order information of functions $f_{1,j}, f_{2,j}, f_{3,j}$, i.e., $\nabla f_{1,j}, \nabla f_{2,j}, \nabla f_{3,j}$, is not available in Eq. (2), corresponding to the level-wise zeroth order constraints. To facilitate the development of distributed algorithms in parameter-server architecture (Jiao et al., 2023; Assran et al., 2020), the distributed TLL with zeroth order constraints in Eq. (2) is equivalently reformulated as a consensus trilevel zeroth order learning problem as follows.

$$
\begin{aligned}
\min \ & \sum_{j=1}^{N} f_{1,j}(\boldsymbol{x}_{1,j}, \boldsymbol{x}_{2,j}, \boldsymbol{x}_{3,j}) \\
\text{s.t. } & \boldsymbol{x}_{1,j} = \boldsymbol{z}_1, \forall j = 1, \cdots, N \\
& \{\boldsymbol{x}_{2,j}\}, \boldsymbol{z}_2 = \arg\min_{\{\boldsymbol{x}_{2,j}'\}, \boldsymbol{z}_2'} \sum_{j=1}^{N} f_{2,j}(\boldsymbol{z}_1, \boldsymbol{x}_{2,j}', \boldsymbol{x}_{3,j}) \\
& \quad \text{s.t. } \boldsymbol{x}_{2,j}' = \boldsymbol{z}_2', \forall j = 1, \cdots, N \\
& \quad \{\boldsymbol{x}_{3,j}\}, \boldsymbol{z}_3 = \arg\min_{\{\boldsymbol{x}_{3,j}'\}, \boldsymbol{z}_3'} \sum_{j=1}^{N} f_{3,j}(\boldsymbol{z}_1, \boldsymbol{z}_2', \boldsymbol{x}_{3,j}') \\
& \qquad \text{s.t. } \boldsymbol{x}_{3,j}' = \boldsymbol{z}_3', \forall j = 1, \cdots, N \\
\text{var. } & \quad \{\boldsymbol{x}_{1,j}\}, \{\boldsymbol{x}_{2,j}\}, \{\boldsymbol{x}_{3,j}\}, \boldsymbol{z}_1, \boldsymbol{z}_2, \boldsymbol{z}_3,
\end{aligned}
\tag{3}
$$

where $\boldsymbol{x}_{1,j} \in \mathbb{R}^{d_1}, \boldsymbol{x}_{2,j} \in \mathbb{R}^{d_2}, \boldsymbol{x}_{3,j} \in \mathbb{R}^{d_3}$ denote the local variables in $j^{\text{th}}$ worker, $\boldsymbol{z}_1 \in \mathbb{R}^{d_1}, \boldsymbol{z}_2 \in \mathbb{R}^{d_2}, \boldsymbol{z}_3 \in \mathbb{R}^{d_3}$ denote the consensus variables in the master, $N$ denotes the number of workers.

**Overview of the proposed framework.** In Sec. 3.1, the construction of cascaded zeroth order polynomial approximation for the trilevel zeroth order learning problem is proposed, which consists of the inner layer and outer layer polynomial approximation. Then, how to gradually update zeroth order cuts to refine the cascaded polynomial approximation is discussed in Sec. 3.2. Finally, a distributed zeroth order algorithm is developed to effectively address the trilevel zeroth order learning problem in a distributed manner in Sec. 3.3. To improve the readability of this work, The notations used in this work and their corresponding definitions are summarized in Table 2.

### 3.1 CASCADED ZEROTH ORDER POLYNOMIAL APPROXIMATION

In this section, how to construct the cascaded zeroth order polynomial approximation for trilevel zeroth order learning is introduced. The proposed cascaded zeroth order polynomial approximation consists of two key parts: 1) the inner layer polynomial approximation and 2) the outer layer polynomial approximation, which will be discussed in detail below.

#### 3.1.1 INNER LAYER POLYNOMIAL APPROXIMATION

In trilevel learning, the third-level optimization problem can be viewed as the constraint to the second-level optimization problem (Jiao et al., 2024; Pan et al., 2024; Kwon et al., 2023; Jiang et al., 2023), it equals the constraint $\phi_{\text{in}}(\{\boldsymbol{x}_{3,j}\}, \boldsymbol{z}_1, \boldsymbol{z}_2', \boldsymbol{z}_3) = 0$, where $\phi_{\text{in}}(\{\boldsymbol{x}_{3,j}\}, \boldsymbol{z}_1, \boldsymbol{z}_2', \boldsymbol{z}_3) =$
$\|\begin{bmatrix} \{\boldsymbol{x}_{3,j}\} \\ \boldsymbol{z}_3 \end{bmatrix} - \underset{\{\boldsymbol{x}_{3,j}'\}, \boldsymbol{z}_3'}{\arg\min} \sum_j f_{3,j}(\boldsymbol{z}_1, \boldsymbol{z}_2', \boldsymbol{x}_{3,j}') \text{ s.t. } \boldsymbol{x}_{3,j}' = \boldsymbol{z}_3', \forall j\|^2$. In many bilevel and trilevel machine learning applications, e.g., neural architecture search in Liu et al. (2018a), robust hyperparameter optimization in Jiao et al. (2024), the lower-level optimization problem serves as a **soft constraint** (Kautz et al., 1996) to the upper-level optimization problem, i.e., this constraint (constraint $\phi_{\text{in}}(\{\boldsymbol{x}_{3,j}\}, \boldsymbol{z}_1, \boldsymbol{z}_2', \boldsymbol{z}_3) = 0$ in our problem) can be violated to a certain extent while still yielding a feasible and meaningful solution, more discussions are provided in Appendix E. Inspired by Jiao et al. (2023); Chen et al. (2024c), the cutting plane based method is utilized to construct a *decomposable* polynomial relaxation for this constraint, which *significantly facilitates* the development of distributed algorithms. Specifically, the inner layer zeroth order cuts are utilized to approximate the feasible region with respect to constraint $\phi_{\text{in}}(\{\boldsymbol{x}_{3,j}\}, \boldsymbol{z}_1, \boldsymbol{z}_2', \boldsymbol{z}_3) = 0$. Zeroth order cuts refer to the cutting planes that do not rely on first order information during generation. In this section, we focus on the construction of cascaded polynomial approximation, and how to generate the zeroth order cuts is discussed in detail in the next section 3.2. Consequently, the feasible region formed by inner layer zeroth order cuts in $t^{\text{th}}$ iteration can be expressed as,

$$P_{\text{in}}^t = \left\{ \sum_j \boldsymbol{a}_{j,l}^{\text{in}\top} \boldsymbol{x}_{3,j}^2 + \boldsymbol{b}_{j,l}^{\text{in}\top} \boldsymbol{x}_{3,j} + \sum_{i \in \{1,3\}} \boldsymbol{c}_{i,l}^{\text{in}\top} \boldsymbol{z}_i^2 + \boldsymbol{d}_{i,l}^{\text{in}\top} \boldsymbol{z}_i + \boldsymbol{c}_{2,l}^{\text{in}\top} \boldsymbol{z}_2^{2'} + \boldsymbol{d}_{2,l}^{\text{in}\top} \boldsymbol{z}_2' + e_l^{\text{in}} \leq \varepsilon_{\text{in}}, \forall l \right\}, \quad (4)$$

where $\boldsymbol{x}_{i,j}^2 = [x_{i,j,1}^2, \cdots, x_{i,j,d_i}^2] \in \mathbb{R}^{d_i}$, $\boldsymbol{z}_i^2 = [z_{i,1}^2, \cdots, z_{i,d_i}^2] \in \mathbb{R}^{d_i}$, $i = 1,2,3$, $\boldsymbol{a}_{j,l}^{\text{in}} \in \mathbb{R}^{d_3}$, $\boldsymbol{b}_{j,l}^{\text{in}} \in \mathbb{R}^{d_3}$, $\boldsymbol{c}_{i,l}^{\text{in}} \in \mathbb{R}^{d_i}$, $\boldsymbol{d}_{i,l}^{\text{in}} \in \mathbb{R}^{d_i}$, and $e_l^{\text{in}} \in \mathbb{R}^1$ are the parameters of $l^{\text{th}}$ inner layer zeroth order cut, $\varepsilon_{\text{in}} \geq 0$ is a constant. By using the inner layer polynomial approximation according to Eq. (4), the resulting problem can be written as,

$$
\begin{aligned}
\min \ & \textstyle\sum_{j=1}^N f_{1,j}(\boldsymbol{x}_{1,j}, \boldsymbol{x}_{2,j}, \boldsymbol{x}_{3,j}) \\
\text{s.t. } & \boldsymbol{x}_{1,j} = \boldsymbol{z}_1, \forall j = 1, \cdots, N \\
& \{\boldsymbol{x}_{2,j}\}, \boldsymbol{z}_2 = \underset{\{\boldsymbol{x}_{2,j}'\}, \boldsymbol{z}_2'}{\arg\min} \ \textstyle\sum_{j=1}^N f_{2,j}(\boldsymbol{z}_1, \boldsymbol{x}_{2,j}', \boldsymbol{x}_{3,j}) \\
& \qquad\quad \text{s.t. } \boldsymbol{x}_{2,j}' = \boldsymbol{z}_2', \forall j = 1, \cdots, N \\
& \qquad\quad (\{\boldsymbol{x}_{3,j}\}, \boldsymbol{z}_1, \boldsymbol{z}_2', \boldsymbol{z}_3) \in P_{\text{in}}^t \\
\text{var. } & \{\boldsymbol{x}_{1,j}\}, \{\boldsymbol{x}_{2,j}\}, \{\boldsymbol{x}_{3,j}\}, \boldsymbol{z}_1, \boldsymbol{z}_2, \boldsymbol{z}_3.
\end{aligned}
\quad (5)
$$

#### 3.1.2 OUTER LAYER POLYNOMIAL APPROXIMATION

Likewise, the lower-level optimization problem in Eq. (5) can be regarded as the constraint to the upper-level optimization problem. Defining $h_l^{\text{in}}(\{\boldsymbol{x}_{3,j}\}, \boldsymbol{z}_1, \boldsymbol{z}_2', \boldsymbol{z}_3) = \sum_j \boldsymbol{a}_{j,l}^{\text{in}\top} \boldsymbol{x}_{3,j}^2 + \boldsymbol{b}_{j,l}^{\text{in}\top} \boldsymbol{x}_{3,j} + \sum_{i \in \{1,3\}} \boldsymbol{c}_{i,l}^{\text{in}\top} \boldsymbol{z}_i^2 + \boldsymbol{d}_{i,l}^{\text{in}\top} \boldsymbol{z}_i + \boldsymbol{c}_{2,l}^{\text{in}\top} \boldsymbol{z}_2^{2'} + \boldsymbol{d}_{2,l}^{\text{in}\top} \boldsymbol{z}_2' + e_l^{\text{in}}$. This constraint equals $\phi_{\text{out}}(\{\boldsymbol{x}_{2,j}\}, \{\boldsymbol{x}_{3,j}\}, \boldsymbol{z}_1, \boldsymbol{z}_2, \boldsymbol{z}_3) = 0$, where

$$
\begin{aligned}
& \phi_{\text{out}}(\{\boldsymbol{x}_{2,j}\}, \{\boldsymbol{x}_{3,j}\}, \boldsymbol{z}_1, \boldsymbol{z}_2, \boldsymbol{z}_3) \\
& = \left\| \begin{bmatrix} \{\boldsymbol{x}_{2,j}\} \\ \boldsymbol{z}_2 \end{bmatrix} - \begin{array}{l} \underset{\{\boldsymbol{x}_{2,j}'\}, \boldsymbol{z}_2'}{\arg\min} \ \textstyle\sum_{j=1}^N f_{2,j}(\boldsymbol{z}_1, \boldsymbol{x}_{2,j}', \boldsymbol{x}_{3,j}) \\ \text{s.t. } \boldsymbol{x}_{2,j}' = \boldsymbol{z}_2', \forall j, h_l^{\text{in}}(\{\boldsymbol{x}_{3,j}\}, \boldsymbol{z}_1, \boldsymbol{z}_2', \boldsymbol{z}_3) \leq \varepsilon_{\text{in}}, \forall l \end{array} \right\|^2.
\end{aligned}
\quad (6)
$$

The constraint $\phi_{\text{out}}(\{\boldsymbol{x}_{2,j}\}, \{\boldsymbol{x}_{3,j}\}, \boldsymbol{z}_1, \boldsymbol{z}_2, \boldsymbol{z}_3) = 0$ also serves as a *soft constraint* to the upper-level optimization problem, more discussions about the soft constraint are provided in Appendix E. Outer layer zeroth order cuts are utilized to construct the polynomial approximation for the feasible region with respect to the constraint $\phi_{\text{out}}(\{\boldsymbol{x}_{2,j}\}, \{\boldsymbol{x}_{3,j}\}, \boldsymbol{z}_1, \boldsymbol{z}_2, \boldsymbol{z}_3) = 0$, that is,

$$P_{\text{out}}^t = \left\{ \{\boldsymbol{x}_{2,j}\}, \{\boldsymbol{x}_{3,j}\}, \boldsymbol{z}_1, \boldsymbol{z}_2, \boldsymbol{z}_3 \mid h_l^{\text{out}}(\{\boldsymbol{x}_{2,j}\}, \{\boldsymbol{x}_{3,j}\}, \boldsymbol{z}_1, \boldsymbol{z}_2, \boldsymbol{z}_3) \leq \varepsilon_{\text{out}}, \forall l \right\}, \quad (7)$$

where $h_l^{\text{out}}(\{\boldsymbol{x}_{2,j}\},\{\boldsymbol{x}_{3,j}\},\boldsymbol{z}_1,\boldsymbol{z}_2,\boldsymbol{z}_3) = \sum_{i=1}^3 \sum_{j=1}^N \boldsymbol{a}_{i,j,l}^{\text{out}}{}^\top \boldsymbol{x}_{i,j}^2 + \boldsymbol{b}_{i,j,l}^{\text{out}}{}^\top \boldsymbol{x}_{i,j} + \sum_{i=1}^3 \boldsymbol{c}_{i,l}^{\text{out}}{}^\top \boldsymbol{z}_i^2 + \boldsymbol{d}_{i,l}^{\text{out}}{}^\top \boldsymbol{z}_i + e_l^{\text{out}}$, and $\varepsilon_{\text{out}} \geq 0$ is a pre-set constant. Based on Eq. (7), the resulting cascaded zeroth order polynomial approximation problem can be written as,

$$
\begin{aligned}
&\min \sum_{j=1}^N f_{1,j}(\boldsymbol{x}_{1,j},\boldsymbol{x}_{2,j},\boldsymbol{x}_{3,j}) \\
&\text{s.t. } \boldsymbol{x}_{1,j} = \boldsymbol{z}_1, \forall j = 1,\cdots,N \\
&\sum_{i=2}^3 \sum_{j=1}^N \boldsymbol{a}_{i,j,l}^{\text{out}}{}^\top \boldsymbol{x}_{i,j}^2 + \boldsymbol{b}_{i,j,l}^{\text{out}}{}^\top \boldsymbol{x}_{i,j} + \sum_{i=1}^3 \boldsymbol{c}_{i,l}^{\text{out}}{}^\top \boldsymbol{z}_i^2 + \boldsymbol{d}_{i,l}^{\text{out}}{}^\top \boldsymbol{z}_i + e_l^{\text{out}} \leq \varepsilon_{\text{out}}, \forall l \\
&\text{var.} \quad \{\boldsymbol{x}_{1,j}\}, \{\boldsymbol{x}_{2,j}\}, \{\boldsymbol{x}_{3,j}\}, \boldsymbol{z}_1, \boldsymbol{z}_2, \boldsymbol{z}_3,
\end{aligned}
\tag{8}
$$

where $\boldsymbol{a}_{i,j,l}^{\text{out}} \in \mathbb{R}^{d_i}$, $\boldsymbol{b}_{i,j,l}^{\text{out}} \in \mathbb{R}^{d_i}$, $\boldsymbol{c}_{i,l}^{\text{out}} \in \mathbb{R}^{d_i}$, $\boldsymbol{d}_{i,l}^{\text{out}} \in \mathbb{R}^{d_i}$, and $e_l^{\text{out}} \in \mathbb{R}^1$ are the parameters of $l^{\text{th}}$ outer layer zeroth order cut.

## 3.2 Refining the Cascaded Polynomial Approximation

For every $\mathcal{T}$ iteration, the zeroth order cuts will be updated to refine the proposed cascaded polynomial approximation when $t < T_1$. Different from the existing cutting plane methods for nested optimization, the proposed zeroth order cuts can be generated without using gradients or sub-gradients, which is why we refer to them as zeroth order cuts. Specifically, in $t^{\text{th}}$ iteration, the zeroth order cuts will be updated by three key steps: 1) generating inner layer zeroth order cut; 2) generating outer layer zeroth order cut; 3) removing inactive zeroth order cuts, which will be discussed as follows. In addition, we demonstrate the proposed zeroth order cuts can construct a relaxation for the original feasible regions in Proposition 1 and 2.

### 3.2.1 Generating Inner Layer Zeroth Order Cut

At $t^{\text{th}}$ iteration, based on point $(\{\boldsymbol{x}_{3,j}^t\}, \boldsymbol{z}_1^t, \boldsymbol{z}_2^t, \boldsymbol{z}_3^t)$, the new inner layer zeroth order cut will be generated to refine the inner layer polynomial approximation, i.e., Eq. (4), as follows.

$$
\phi_{\text{in}}(\{\boldsymbol{x}_{3,j}^t\}, \boldsymbol{z}_1^t, \boldsymbol{z}_2^{t'}, \boldsymbol{z}_3^t) + G_\mu^{\text{in}}(\{\boldsymbol{x}_{3,j}^t\}, \boldsymbol{z}_1^t, \boldsymbol{z}_2^{t'}, \boldsymbol{z}_3^t)^\top \left( \begin{bmatrix} \{\boldsymbol{x}_{3,j}\} \\ \boldsymbol{z}_1 \\ \boldsymbol{z}_2' \\ \boldsymbol{z}_3 \end{bmatrix} - \begin{bmatrix} \{\boldsymbol{x}_{3,j}^t\} \\ \boldsymbol{z}_1^t \\ \boldsymbol{z}_2^{t'} \\ \boldsymbol{z}_3^t \end{bmatrix} \right)
\tag{9}
$$

$$
\leq \frac{L+1}{2} \left( \sum_j ||\boldsymbol{x}_{3,j} - \boldsymbol{x}_{3,j}^t||^2 + ||\boldsymbol{z}_1 - \boldsymbol{z}_1^t||^2 + ||\boldsymbol{z}_2' - \boldsymbol{z}_2^{t'}||^2 + ||\boldsymbol{z}_3 - \boldsymbol{z}_3^t||^2 \right) + \frac{\mu^2}{8} L^2 d_{\text{in}} + \varepsilon_{\text{in}},
$$

where $d_{\text{in}} = (d_1 + d_2 + (N+1)d_3 + 3)^3$ and

$$
G_\mu^{\text{in}}(\{\boldsymbol{x}_{3,j}^t\}, \boldsymbol{z}_1^t, \boldsymbol{z}_2^{t'}, \boldsymbol{z}_3^t) = \frac{\phi_{\text{in}}(\{\boldsymbol{x}_{3,j}^t + \mu\boldsymbol{\mu}_{x_{3,j}}\}, \boldsymbol{z}_1^t + \mu\boldsymbol{\mu}_{z_1}, \boldsymbol{z}_2^{t'} + \mu\boldsymbol{\mu}_{z_2}, \boldsymbol{z}_3^t + \mu\boldsymbol{\mu}_{z_3}) - \phi_{\text{in}}(\{\boldsymbol{x}_{3,j}^t\}, \boldsymbol{z}_1^t, \boldsymbol{z}_2^{t'}, \boldsymbol{z}_3^t)}{\mu} \boldsymbol{\mu}^{\text{in}},
\tag{10}
$$

where $\boldsymbol{\mu}^{\text{in}} = [\{\boldsymbol{\mu}_{x_{3,j}}\}, \boldsymbol{\mu}_{z_1}, \boldsymbol{\mu}_{z_2}, \boldsymbol{\mu}_{z_3}]$ is a standard Gaussian random vector, $L > 0$ is a constant, and $\mu > 0$ is the smoothing parameter (Kornowski and Shamir, 2024; Ghadimi and Lan, 2013). Then, the new generated zeroth order cut $cp_{\text{in}}^{\text{new}}$ will be added into $P_{\text{in}}^t$, i.e., $P_{\text{in}}^t = \text{Add}(P_{\text{in}}^{t-1}, cp_{\text{in}}^{\text{new}})$.

**Proposition 1** *The original feasible region of constraint $\phi_{\text{in}}(\{\boldsymbol{x}_{3,j}\}, \boldsymbol{z}_1, \boldsymbol{z}_2', \boldsymbol{z}_3) = 0$ is a subset of the feasible region formed by inner layer zeroth order cuts, i.e., $P_{\text{in}}^{t+1} = \{h_l^{\text{in}}(\{\boldsymbol{x}_{3,j}\}, \boldsymbol{z}_1, \boldsymbol{z}_2', \boldsymbol{z}_3) \leq \varepsilon_{\text{in}}, \forall l\}$ when $\phi_{\text{in}}$ has $L$-Lipschitz continuous gradient. The proof is provided in Appendix C.*

### 3.2.2 Generating Outer Layer Zeroth Order Cut

At $t^{\text{th}}$ iteration, according to point $(\{\boldsymbol{x}_{2,j}^t\}, \{\boldsymbol{x}_{3,j}^t\}, \boldsymbol{z}_1^t, \boldsymbol{z}_2^t, \boldsymbol{z}_3^t)$, the new outer layer zeroth order cut will be generated to refine the outer layer polynomial approximation in Eq. (7) as follows.

$$
\phi_{\text{out}}(\{\boldsymbol{x}_{2,j}^t\}, \{\boldsymbol{x}_{3,j}^t\}, \boldsymbol{z}_1^t, \boldsymbol{z}_2^t, \boldsymbol{z}_3^t) + G_\mu^{\text{out}}(\{\boldsymbol{x}_{2,j}^t\}, \{\boldsymbol{x}_{3,j}^t\}, \boldsymbol{z}_1^t, \boldsymbol{z}_2^t, \boldsymbol{z}_3^t)^\top \left( \begin{bmatrix} \{\boldsymbol{x}_{2,j}\} \\ \{\boldsymbol{x}_{3,j}\} \\ \boldsymbol{z}_1 \\ \boldsymbol{z}_2 \\ \boldsymbol{z}_3 \end{bmatrix} - \begin{bmatrix} \{\boldsymbol{x}_{2,j}^t\} \\ \{\boldsymbol{x}_{3,j}^t\} \\ \boldsymbol{z}_1^t \\ \boldsymbol{z}_2^t \\ \boldsymbol{z}_3^t \end{bmatrix} \right)
$$

$$
\leq \frac{L+1}{2} \left( \sum_{i=2}^3 \sum_j ||\boldsymbol{x}_{i,j} - \boldsymbol{x}_{i,j}^t||^2 + \sum_i ||\boldsymbol{z}_i - \boldsymbol{z}_i^t||^2 \right) + \frac{\mu^2}{8} L^2 (d_1 + (N+1)(d_2 + d_3) + 3)^3 + \varepsilon_{\text{out}}.
\tag{11}
$$

In Eq. (11), we have that,

$$
G_\mu^{\text{out}}(\{\boldsymbol{x}_{2,j}^t\}, \{\boldsymbol{x}_{3,j}^t\}, \boldsymbol{z}_1^t, \boldsymbol{z}_2^t, \boldsymbol{z}_3^t)
$$
$$
= \frac{\phi_{\text{out}}(\{\boldsymbol{x}_{2,j}^t + \mu\boldsymbol{\mu}_{x_{2,j}}\}, \{\boldsymbol{x}_{3,j}^t + \mu\boldsymbol{\mu}_{x_{3,j}}\}, \boldsymbol{z}_1^t + \mu\boldsymbol{\mu}_{z_1}, \boldsymbol{z}_2^t + \mu\boldsymbol{\mu}_{z_2}, \boldsymbol{z}_3^t + \mu\boldsymbol{\mu}_{z_3}) - \phi_{\text{out}}(\{\boldsymbol{x}_{2,j}^t\}, \{\boldsymbol{x}_{3,j}^t\}, \boldsymbol{z}_1^t, \boldsymbol{z}_2^t, \boldsymbol{z}_3^t)}{\mu} \boldsymbol{\mu}^{\text{out}},
$$

$$(12)$$

where $\boldsymbol{\mu}^{\text{out}} = [\{\boldsymbol{\mu}_{x_{2,j}}\}, \{\boldsymbol{\mu}_{x_{3,j}}\}, \boldsymbol{\mu}_{z_1}, \boldsymbol{\mu}_{z_2}, \boldsymbol{\mu}_{z_3}]$ is a standard Gaussian random vector. Subsequently, the new generated outer layer zeroth order cut $cp_{\text{out}}^{\text{new}}$ will be added into $P_{\text{out}}^t$, i.e., $P_{\text{out}}^t = \text{Add}(P_{\text{out}}^{t-1}, cp_{\text{out}}^{\text{new}})$.

**Proposition 2** *The original feasible region of constraint* $\phi_{\text{out}}(\{\boldsymbol{x}_{2,j}\}, \{\boldsymbol{x}_{3,j}\}, \boldsymbol{z}_1, \boldsymbol{z}_2, \boldsymbol{z}_3) = 0$ *is a subset of the feasible region formed by outer layer zeroth order cuts, i.e.,* $P_{\text{out}}^{t+1} =$

$$
\left\{ \{\boldsymbol{x}_{2,j}\}, \{\boldsymbol{x}_{3,j}\}, \boldsymbol{z}_1, \boldsymbol{z}_2, \boldsymbol{z}_3 \,\Big|\, \sum_{i=2}^3 \sum_{j=1}^N \boldsymbol{a}_{i,j,l}^{\text{out}\top} \boldsymbol{x}_{i,j}^2 + \boldsymbol{b}_{i,j,l}^{\text{out}\top} \boldsymbol{x}_{i,j} + \sum_{i=1}^3 \boldsymbol{c}_{i,l}^{\text{out}\top} \boldsymbol{z}_i^2 + \boldsymbol{d}_{i,l}^{\text{out}\top} \boldsymbol{z}_i + e_l^{\text{out}} \le \varepsilon_{\text{out}}, \forall l \right\}
$$

*when* $\phi_{\text{out}}$ *has L-Lipschitz continuous gradient. Proofs are provided in Appendix C.*

### 3.2.3 Removing Inactive Zeroth Order Cuts

To improve the effectiveness and reduce the complexity (Yang et al., 2014; Jiao et al., 2023), the inactive zeroth order cuts will be removed during the iteration process. The corresponding inner layer $P_{\text{in}}^t$ and outer layer $P_{\text{out}}^t$ will be updated as follows.

$$
P_{\text{in}}^t = \begin{cases} \text{Remove}(P_{\text{in}}^t, cp_{\text{in},l}), \text{if } h_l^{\text{in}}(\{\boldsymbol{x}_{3,j}^t\}, \boldsymbol{z}_1^t, \boldsymbol{z}_2^{t\prime}, \boldsymbol{z}_3^t) < \varepsilon_{\text{in}}, \forall l \\ P_{\text{in}}^t, \text{otherwise} \end{cases}, \tag{13}
$$

$$
P_{\text{out}}^t = \begin{cases} \text{Remove}(P_{\text{out}}^t, cp_{\text{out},l}), \text{if } h_l^{\text{out}}(\{\boldsymbol{x}_{2,j}^t\}, \{\boldsymbol{x}_{3,j}^t\}, \boldsymbol{z}_1^t, \boldsymbol{z}_2^t, \boldsymbol{z}_3^t) < \varepsilon_{\text{out}}, \forall l \\ P_{\text{out}}^t, \text{otherwise} \end{cases}, \tag{14}
$$

where $\text{Remove}(P_{\text{in}}^t, cp_{\text{in},l})$ and $\text{Remove}(P_{\text{out}}^t, cp_{\text{out},l})$ respectively represent that the $l^{\text{th}}$ inner layer and outer layer zeroth order cuts will be removed from $P_{\text{in}}^t$ and $P_{\text{out}}^t$.

### 3.3 Zeroth Order Distributed Algorithm

In this section, a distributed zeroth order algorithm is proposed. First, defining function $o(\{\boldsymbol{x}_{2,j}\}, \{\boldsymbol{x}_{3,j}\}, \boldsymbol{z}_1, \boldsymbol{z}_2, \boldsymbol{z}_3) = \sum_l \lambda_l [\max\{h_l^{\text{out}}(\{\boldsymbol{x}_{2,j}\}, \{\boldsymbol{x}_{3,j}\}, \boldsymbol{z}_1, \boldsymbol{z}_2, \boldsymbol{z}_3) - \varepsilon_{\text{out}}, 0\}]^2$, where $\lambda_l > 0$ is a penalty parameter. The constrained optimization problem described in Eq. (8) is reformulated as an unconstrained optimization problem by using the exterior penalty method (Shen and Chen, 2023; Shi and Gu, 2021; Boyd and Vandenberghe, 2004) as follows.

$$
F(\{\boldsymbol{x}_{1,j}\}, \{\boldsymbol{x}_{2,j}\}, \{\boldsymbol{x}_{3,j}\}, \boldsymbol{z}_1, \boldsymbol{z}_2, \boldsymbol{z}_3) = \sum_{j=1}^N f_{1,j}(\boldsymbol{x}_{1,j}, \boldsymbol{x}_{2,j}, \boldsymbol{x}_{3,j}) + \phi_j \|\boldsymbol{x}_{1,j} - \boldsymbol{z}_1\|^2
$$
$$
+ o(\{\boldsymbol{x}_{2,j}\}, \{\boldsymbol{x}_{3,j}\}, \boldsymbol{z}_1, \boldsymbol{z}_2, \boldsymbol{z}_3), \tag{15}
$$

where $\phi_j > 0$ is a penalty parameter. It is worth noting that the proposed DTZO is an expandable framework, allowing the incorporation of approaches beyond exterior penalty method, e.g., gradient projection based approaches (Xu et al., 2020) and Frank-Wolfe based methods (Shen et al., 2019). We chose exterior penalty method because the lower-level problem often serves as a soft constraint (as discussed in Sec. 3.1 and Appendix E) and using exterior penalty method offers comparatively *lower* complexity. In addition, we demonstrate that the optimal solution to problem in Eq. (15) is a feasible solution to the original constrained problem; 2) the gap between the problem in Eq. (15) and original constrained problem will continuously decrease as $\lambda_l, \phi_j$ increase. Detailed discussions are provided in Appendix H. In $(t+1)^{\text{th}}$ iteration, the proposed algorithm proceeds as follows.

**In Worker** $j$. After receiving the updated parameters $\boldsymbol{z}_i^t$ and $\nabla_{\boldsymbol{x}_{i,j}} o(\{\boldsymbol{x}_{2,j}^t\}, \{\boldsymbol{x}_{3,j}^t\}, \boldsymbol{z}_1^t, \boldsymbol{z}_2^t, \boldsymbol{z}_3^t)$, worker $j$ updates the local variables as follows,

$$
\boldsymbol{x}_{1,j}^{t+1} = \boldsymbol{x}_{1,j}^t - \eta_{\boldsymbol{x}_1} G_{\boldsymbol{x}_{1,j}}(\{\boldsymbol{x}_{1,j}^t\}, \{\boldsymbol{x}_{2,j}^t\}, \{\boldsymbol{x}_{3,j}^t\}, \boldsymbol{z}_1^t, \boldsymbol{z}_2^t, \boldsymbol{z}_3^t), \tag{16}
$$

$$
\boldsymbol{x}_{2,j}^{t+1} = \boldsymbol{x}_{2,j}^t - \eta_{\boldsymbol{x}_2} G_{\boldsymbol{x}_{2,j}}(\{\boldsymbol{x}_{1,j}^t\}, \{\boldsymbol{x}_{2,j}^t\}, \{\boldsymbol{x}_{3,j}^t\}, \boldsymbol{z}_1^t, \boldsymbol{z}_2^t, \boldsymbol{z}_3^t), \tag{17}
$$

$$
\boldsymbol{x}_{3,j}^{t+1} = \boldsymbol{x}_{3,j}^t - \eta_{\boldsymbol{x}_3} G_{\boldsymbol{x}_{3,j}}(\{\boldsymbol{x}_{1,j}^t\}, \{\boldsymbol{x}_{2,j}^t\}, \{\boldsymbol{x}_{3,j}^t\}, \boldsymbol{z}_1^t, \boldsymbol{z}_2^t, \boldsymbol{z}_3^t), \tag{18}
$$

we have that,

$$
G_{\boldsymbol{x}_{1,j}}(\{\boldsymbol{x}_{1,j}^t\}, \{\boldsymbol{x}_{2,j}^t\}, \{\boldsymbol{x}_{3,j}^t\}, \boldsymbol{z}_1^t, \boldsymbol{z}_2^t, \boldsymbol{z}_3^t)
$$
$$
= \frac{f_{1,j}(\boldsymbol{x}_{1,j}^t + \mu\boldsymbol{u}_{k,1}, \boldsymbol{x}_{2,j}^t, \boldsymbol{x}_{3,j}^t) - f_{1,j}(\boldsymbol{x}_{1,j}^t, \boldsymbol{x}_{2,j}^t, \boldsymbol{x}_{3,j}^t)}{\mu} \boldsymbol{u}_{k,1} + 2\phi_j(\boldsymbol{x}_{1,j}^t - \boldsymbol{z}_1^t), \tag{19}
$$

---

**Algorithm 1** DTZO: Distributed Trilevel Zeroth Order Learning

---

**Initialization:** master iteration $t = 0$, variables $\{\boldsymbol{x}_{1,j}^0\}, \{\boldsymbol{x}_{2,j}^0\}, \{\boldsymbol{x}_{3,j}^0\}, \boldsymbol{z}_1^0, \boldsymbol{z}_2^0, \boldsymbol{z}_3^0$.
**repeat**
  **for** *local worker $j$* **do**
    updates the local variables $\boldsymbol{x}_{1,j}^{t+1}, \boldsymbol{x}_{2,j}^{t+1}, \boldsymbol{x}_{3,j}^{t+1}$ according to Eq. (16)-(21);
  **end for**
  *local workers* transmit the updated variables to the master;
  **for** *master* **do**
    updates consensus variables $\boldsymbol{z}_1^{t+1}, \boldsymbol{z}_2^{t+1}, \boldsymbol{z}_3^{t+1}$ according to Eq. (22)-(24);
    computes $\nabla o(\{\boldsymbol{x}_{2,j}^{t+1}\}, \{\boldsymbol{x}_{3,j}^{t+1}\}, \boldsymbol{z}_1^{t+1}, \boldsymbol{z}_2^{t+1}, \boldsymbol{z}_3^{t+1})$;
  **end for**
  *master* broadcasts the updated parameters and gradients to workers;
  **if** $(t+1) \bmod \mathcal{T} == 0$ and $t < T_1$ **then**
    new inner layer zeroth order cuts are generated by Eq. (9) and (10);
    new outer layer zeroth order cuts are generated by Eq. (11) and (12);
    inactive zeroth order cuts are deleted by (13) and (14);
  **end if**
  $t = t + 1$;
**until** termination.

---

$$G_{\boldsymbol{x}_{2,j}}(\{\boldsymbol{x}_{1,j}^t\}, \{\boldsymbol{x}_{2,j}^t\}, \{\boldsymbol{x}_{3,j}^t\}, \boldsymbol{z}_1^t, \boldsymbol{z}_2^t, \boldsymbol{z}_3^t)$$
$$= \frac{f_{1,j}(\boldsymbol{x}_{1,j}^t, \boldsymbol{x}_{2,j}^t + \mu\boldsymbol{u}_{k,2}, \boldsymbol{x}_{3,j}^t) - f_{1,j}(\boldsymbol{x}_{1,j}^t, \boldsymbol{x}_{2,j}^t, \boldsymbol{x}_{3,j}^t)}{\mu}\boldsymbol{u}_{k,2} + \nabla_{\boldsymbol{x}_{2,j}} o(\{\boldsymbol{x}_{2,j}^t\}, \{\boldsymbol{x}_{3,j}^t\}, \boldsymbol{z}_1^t, \boldsymbol{z}_2^t, \boldsymbol{z}_3^t), \tag{20}$$

$$G_{\boldsymbol{x}_{3,j}}(\{\boldsymbol{x}_{1,j}^t\}, \{\boldsymbol{x}_{2,j}^t\}, \{\boldsymbol{x}_{3,j}^t\}, \boldsymbol{z}_1^t, \boldsymbol{z}_2^t, \boldsymbol{z}_3^t)$$
$$= \frac{f_{1,j}(\boldsymbol{x}_{1,j}^t, \boldsymbol{x}_{2,j}^t, \boldsymbol{x}_{3,j}^t + \mu\boldsymbol{u}_{k,3}) - f_{1,j}(\boldsymbol{x}_{1,j}^t, \boldsymbol{x}_{2,j}^t, \boldsymbol{x}_{3,j}^t)}{\mu}\boldsymbol{u}_{k,3} + \nabla_{\boldsymbol{x}_{3,j}} o(\{\boldsymbol{x}_{2,j}^t\}, \{\boldsymbol{x}_{3,j}^t\}, \boldsymbol{z}_1^t, \boldsymbol{z}_2^t, \boldsymbol{z}_3^t), \tag{21}$$

where $\boldsymbol{u}_{k,i} \in \mathbb{R}^{d_i}, \forall i$ are standard Gaussian random vectors, $\mu > 0$ is smoothing parameter, $\eta_{\boldsymbol{x}_i}, \forall i$ are step-sizes. Then, the updated variables $\boldsymbol{x}_{1,j}^{t+1}, \boldsymbol{x}_{2,j}^{t+1}, \boldsymbol{x}_{3,j}^{t+1}$ will be transmitted to the master.

**In Master**. After receiving updated variables from workers, the master performs the following steps,

1. Updating consensus variables,
$$\boldsymbol{z}_1^{t+1} = \boldsymbol{z}_1^t - \eta_{\boldsymbol{z}_1}\left(\sum_j 2\phi_j(\boldsymbol{z}_1^t - \boldsymbol{x}_{1,j}^t) + \nabla_{\boldsymbol{z}_1} o(\{\boldsymbol{x}_{2,j}^t\}, \{\boldsymbol{x}_{3,j}^t\}, \boldsymbol{z}_1^t, \boldsymbol{z}_2^t, \boldsymbol{z}_3^t)\right), \tag{22}$$

$$\boldsymbol{z}_2^{t+1} = \boldsymbol{z}_2^t - \eta_{\boldsymbol{z}_2}\nabla_{\boldsymbol{z}_2} o(\{\boldsymbol{x}_{2,j}^t\}, \{\boldsymbol{x}_{3,j}^t\}, \boldsymbol{z}_1^t, \boldsymbol{z}_2^t, \boldsymbol{z}_3^t), \tag{23}$$

$$\boldsymbol{z}_3^{t+1} = \boldsymbol{z}_3^t - \eta_{\boldsymbol{z}_3}\nabla_{\boldsymbol{z}_3} o(\{\boldsymbol{x}_{2,j}^t\}, \{\boldsymbol{x}_{3,j}^t\}, \boldsymbol{z}_1^t, \boldsymbol{z}_2^t, \boldsymbol{z}_3^t), \tag{24}$$

where $\eta_{\boldsymbol{z}_1}, \eta_{\boldsymbol{z}_2}$ and $\eta_{\boldsymbol{z}_3}$ are step-sizes.

2. Computing gradient of $o(\{\boldsymbol{x}_{2,j}^{t+1}\}, \{\boldsymbol{x}_{3,j}^{t+1}\}, \boldsymbol{z}_1^{t+1}, \boldsymbol{z}_2^{t+1}, \boldsymbol{z}_3^{t+1})$. Broadcasting the updated parameters $\boldsymbol{z}_i^{t+1}, i = 1, 2, 3$ and $\nabla_{\boldsymbol{x}_{i,j}} o(\{\boldsymbol{x}_{2,j}^{t+1}\}, \{\boldsymbol{x}_{3,j}^{t+1}\}, \boldsymbol{z}_1^{t+1}, \boldsymbol{z}_2^{t+1}, \boldsymbol{z}_3^{t+1}), i = 2, 3$ to workers.

**Discussion:** TLL with *level-wise* zeroth order constraints is considered in this work, where first order information at *each level* is unavailable. Note that the proposed DTZO is versatile and can be adapted to a wide range of TLL, e.g., grey-box TLL (gradients at some levels in TLL are available (Huang et al., 2024b)), with slight adjustments. For instance, if gradients at first-level in TLL are accessible, we can use gradient descent steps to replace Eq. (16)-(18). Similarly, if the second or third-level gradients are available, first order based cuts, e.g., (Jiao et al., 2024), can be employed to construct the cascaded polynomial approximation. Detailed discussions are offered in Appendix I.

## 4 THEORETICAL ANALYSIS

**Definition 1** *(**Stationarity Gap**) Following Xu et al. (2020); Jiao et al. (2023), the stationarity gap at $t^{\text{th}}$ iteration in this problem can be expressed as,*

$$\mathcal{G}^t = \begin{bmatrix} \{\nabla_{\boldsymbol{x}_{1,j}} F(\{\boldsymbol{x}_{1,j}^t\}, \{\boldsymbol{x}_{2,j}^t\}, \{\boldsymbol{x}_{3,j}^t\}, \boldsymbol{z}_1^t, \boldsymbol{z}_2^t, \boldsymbol{z}_3^t)\} \\ \{\nabla_{\boldsymbol{x}_{2,j}} F(\{\boldsymbol{x}_{1,j}^t\}, \{\boldsymbol{x}_{2,j}^t\}, \{\boldsymbol{x}_{3,j}^t\}, \boldsymbol{z}_1^t, \boldsymbol{z}_2^t, \boldsymbol{z}_3^t)\} \\ \{\nabla_{\boldsymbol{x}_{3,j}} F(\{\boldsymbol{x}_{1,j}^t\}, \{\boldsymbol{x}_{2,j}^t\}, \{\boldsymbol{x}_{3,j}^t\}, \boldsymbol{z}_1^t, \boldsymbol{z}_2^t, \boldsymbol{z}_3^t)\} \\ \nabla_{\boldsymbol{z}_1} F(\{\boldsymbol{x}_{1,j}^t\}, \{\boldsymbol{x}_{2,j}^t\}, \{\boldsymbol{x}_{3,j}^t\}, \boldsymbol{z}_1^t, \boldsymbol{z}_2^t, \boldsymbol{z}_3^t) \\ \nabla_{\boldsymbol{z}_2} F(\{\boldsymbol{x}_{1,j}^t\}, \{\boldsymbol{x}_{2,j}^t\}, \{\boldsymbol{x}_{3,j}^t\}, \boldsymbol{z}_1^t, \boldsymbol{z}_2^t, \boldsymbol{z}_3^t) \\ \nabla_{\boldsymbol{z}_3} F(\{\boldsymbol{x}_{1,j}^t\}, \{\boldsymbol{x}_{2,j}^t\}, \{\boldsymbol{x}_{3,j}^t\}, \boldsymbol{z}_1^t, \boldsymbol{z}_2^t, \boldsymbol{z}_3^t) \end{bmatrix}. \tag{25}$$

*It is seen from Eq. (25) that,*

$$||\mathcal{G}^t||^2 = \sum_{i=1}^3 \sum_{j=1}^N ||\nabla_{x_{i,j}} F(\{\boldsymbol{x}_{1,j}^t\}, \{\boldsymbol{x}_{2,j}^t\}, \{\boldsymbol{x}_{3,j}^t\}, \boldsymbol{z}_1^t, \boldsymbol{z}_2^t, \boldsymbol{z}_3^t)||^2$$
$$+ \sum_{i=1}^3 ||\nabla_{z_i} F(\{\boldsymbol{x}_{1,j}^t\}, \{\boldsymbol{x}_{2,j}^t\}, \{\boldsymbol{x}_{3,j}^t\}, \boldsymbol{z}_1^t, \boldsymbol{z}_2^t, \boldsymbol{z}_3^t)||^2. \tag{26}$$

**Definition 2** *($\epsilon$-Stationary Point)* $(\{\boldsymbol{x}_{1,j}^t\}, \{\boldsymbol{x}_{2,j}^t\}, \{\boldsymbol{x}_{3,j}^t\}, \boldsymbol{z}_1^t, \boldsymbol{z}_2^t, \boldsymbol{z}_3^t)$ *is the stationary point when* $||\mathcal{G}^t||^2 = 0$. $(\{\boldsymbol{x}_{1,j}^t\}, \{\boldsymbol{x}_{2,j}^t\}, \{\boldsymbol{x}_{3,j}^t\}, \boldsymbol{z}_1^t, \boldsymbol{z}_2^t, \boldsymbol{z}_3^t)$ *is the $\epsilon$-stationary point when* $||\mathcal{G}^t||^2 \le \epsilon$. *Defining $T(\epsilon)$ as the first iteration when* $||\mathcal{G}^t||^2 \le \epsilon$, *i.e.,* $T(\epsilon) = \min\{t \mid ||\mathcal{G}^t||^2 \le \epsilon\}$.

**Definition 3** *($\mu$-Smooth Approximation)* *Following Ghadimi and Lan (2013); Fang et al. (2022); Nesterov and Spokoiny (2017); Kornilov et al. (2024); Rando et al. (2024), the $\mu$-smooth approximation of a function $F(\boldsymbol{w}) : \mathbb{R}^d \to \mathbb{R}^1$ is given by,*

$$F_\mu(\boldsymbol{w}) = \frac{1}{(2\pi)^{\frac{d}{2}}} \int F(\boldsymbol{w} + \mu\boldsymbol{u}) e^{-\frac{1}{2}||\boldsymbol{u}||^2} d\boldsymbol{u} = \mathbb{E}_{\boldsymbol{u}} \left[ F(\boldsymbol{w} + \mu\boldsymbol{u}) \right], \tag{27}$$

*where $\boldsymbol{u} \in \mathbb{R}^d$ is a standard Gaussian random vector and $\mu > 0$ is the smoothing parameter.*

**Assumption 1** *(Boundedness)* *Following many works in machine learning, e.g., Deng et al. (2020); Jiao et al. (2023); Qian et al. (2019); Lei and Tang (2018); Zheng et al. (2017), the bounded domain is assumed, i.e., $||\boldsymbol{x}_{i,j} - \boldsymbol{x}_{i,j}^*||^2 \le \alpha_i, \forall \boldsymbol{x}_{i,j}, ||\boldsymbol{z}_i - \boldsymbol{z}_i^*||^2 \le \alpha_i, \forall \boldsymbol{z}_i$, where $\boldsymbol{x}_{i,j}^*, \boldsymbol{z}_i^*$ denote the optimal solution. Following Cutkosky and Orabona (2019); Liu et al. (2021a); Fang et al. (2022); Shaban et al. (2019), we assume the optimal value $F_\mu{}^* > -\infty$.*

**Assumption 2** *(L-smoothness)* *Following many work in nested optimization and zeroth order learning, e.g., Chen et al. (2023a); Lin et al. (2024); Ghadimi and Lan (2013), we assume the gradient of function $F$ is Lipschitz continuous with constant $L < \infty$, that is, for any point $\boldsymbol{w}, \boldsymbol{w}'$, we have that,*

$$||\nabla F(\boldsymbol{w}) - \nabla F(\boldsymbol{w}')|| \le L||\boldsymbol{w} - \boldsymbol{w}'||. \tag{28}$$

*It is worth noting that both Assumptions 1 and 2 are mild and commonly used in machine learning. Detailed discussions of these assumptions are provided in Appendix G.*

**Theorem 1** *(Iteration Complexity)* *Under Assumption 1 and 2, by setting step-sizes $\eta_{\boldsymbol{x}_i} = \eta_{\boldsymbol{z}_i} = \min\left\{\frac{1}{8L(d_1+4)}, \frac{1}{8L(d_2+4)}, \frac{1}{8L(d_3+4)}, \frac{3}{2(L+1)}, \frac{1}{\sqrt{T(\epsilon)-T_1}}\right\}, i = 1, 2, 3$ and letting smoothing parameter $0 < \mu \le \frac{1}{\sqrt{T(\epsilon)-T_1}}$, we have that,*

$$T(\epsilon) \sim \mathcal{O}\left(\left(\sum_{i=1}^3 \overline{c_i} + \overline{d} \left(\max_{t \in [T_1]} F_\mu(\{\boldsymbol{x}_{i,j}^t\}, \{\boldsymbol{z}_i^t\}) - F_\mu{}^*\right)\right)^2 \frac{1}{\epsilon^2} + T_1\right), \tag{29}$$

*where constants* $\overline{d} = 4(1 + \max\left\{8L(d_1 + 4), 8L(d_2 + 4), 8L(d_3 + 4), \frac{2(L+1)}{3}\right\})$, $\overline{c_i} = \frac{L^2(d_i+6)^3}{4(d_i+4)} + L^2(d_i+3)^3 + 4L(N+1)d_i\left(\max\left\{8L(d_1 + 4), 8L(d_2 + 4), 8L(d_3 + 4), \frac{2(L+1)}{3}\right\} + 1\right)$. $T_1 > 0$ *is a constant that controls the cascaded polynomial approximation, as discussed in Sec. 3.2. Detailed proofs of Theorem 1 are provided in Appendix A, with further discussions offered below.*

**Theorem 2** *(Communication Complexity)* *The overall communication complexity of the proposed DTZO can be divided into the communication complexity at every iteration ($C_1$) and the communication complexity of updating zeroth order cuts ($C_2$). Specifically, the overall communication complexity can be expressed as $C_1 + C_2 = T(\epsilon)(2d_1 + 3d_2 + 3d_3)N + 2N\lfloor\frac{T_1}{\mathcal{T}}\rfloor\mathcal{T}(d_2 + d_3)$. The detailed proofs are provided in Appendix B, with further discussions offered as follows.*

**Discussion:** It is seen from Theorem 1 and 2 that the proposed framework DTZO can *flexibly* control the trade-off between the performance of cascaded polynomial approximation and the iteration complexity (i.e., $T(\epsilon)$ in Theorem 1) and communication complexity (i.e., $C_1 + C_2$ in Theorem 2) by adjusting a single parameter $T_1$. Specifically, a larger $T_1$ corresponds to a better cascaded polynomial approximation, but it also entails higher iteration and communication complexity. Consequently, if the distributed system has limited computational and communication capabilities, a smaller value of $T_1$ can be selected. Conversely, if a higher quality of cascaded polynomial approximation is desired, a larger value of $T_1$ can be chosen, which demonstrates the flexibility in the

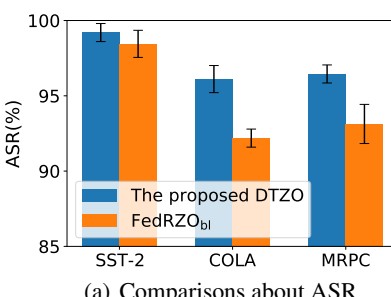 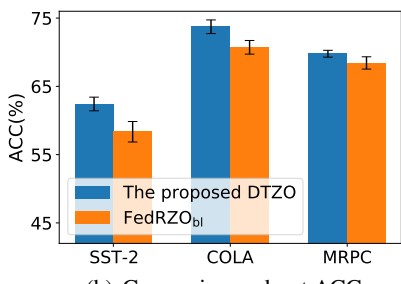

(a) Comparisons about ASR  (b) Comparisons about ACC

Figure 1: Comparisons about ASR and ACC between the proposed DTZO and the state-of-the-art distributed bilevel zeroth order learning method FedRZO$_{\text{bl}}$ (Qiu et al., 2023).

proposed framework. In addition, as shown in Theorem 1, the iteration complexity of the proposed distributed trilevel zeroth order learning framework can be written as $\mathcal{O}(\sum_i d_i^6/\epsilon^2)$. It is worth mentioning that the dimension-dependent iteration complexity is *common* in zeroth order optimization, as discussed in various works (Zhang et al., 2024b;a; Duchi et al., 2015; Sun et al., 2022; Qiu et al., 2023). For instance, the iteration complexity of the state-of-the-art distributed zeroth order bilevel learning method (Qiu et al., 2023) is given by $\mathcal{O}(d^8/\epsilon^2)$, where $d$ denotes the dimension of variables.

## 5 EXPERIMENTS

In the experiment, two distributed trilevel zeroth order learning scenarios, i.e., black-box trilevel learning on large language models (LLMs) and robust hyperparameter optimization are used to evaluate the performance of the proposed DTZO. In the zeroth order setting, the existing distributed nested optimization algorithms based on first order information, e.g., (Jiao et al., 2024), are not available in the experiment. The proposed DTZO is compared with the state-of-the-art distributed zeroth order learning method FedZOO (Fang et al., 2022) and distributed bilevel zeroth order learning method FedRZO$_{\text{bl}}$ (Qiu et al., 2023). In the experiment, all the models are implemented using PyTorch, and the experiments are conducted on a server equipped with two NVIDIA RTX 4090 GPUs. More experimental details are provided in Appendix F.

### 5.1 BLACK-BOX TRILEVEL LEARNING

Prompt learning is a key technique for enabling LLMs to efficiently and effectively adapt to various downstream tasks (Ma et al., 2024; Wang et al., 2024). In many practical scenarios involving LLMs, access to first-order information is restricted due to the proprietary nature of these models or API constraints. For instance, commercial LLM APIs only allow input-output interactions and do not provide visibility into gradients. Inspired by the black-box prompt learning (Diao et al., 2022) and backdoor attack on prompt-based LLMs (Yao et al., 2024), the backdoor attack on black-box LLMs is considered in the experiment, which can be expressed as a black-box trilevel learning problem,

$$
\begin{aligned}
\min_{\lambda} &\ \sum_{j=1}^{N} \frac{1}{|D_j^{\text{val}}|} \sum_{(\boldsymbol{s}_i, y_i) \sim D_j^{\text{val}}} L(\mathcal{G}, [\boldsymbol{k}_{\text{tri}}, \boldsymbol{p}, \boldsymbol{s}_i], y_i) \\
\text{s.t. } &\ \boldsymbol{k}_{\text{tri}} = \arg\min_{\boldsymbol{k}_{\text{tri}}'} \sum_{j=1}^{N} \frac{1}{|D_j^{\text{tr}}|} \sum_{(\boldsymbol{s}_i, y_i) \sim D_j^{\text{tr}}} L(\mathcal{G}, [\boldsymbol{k}_{\text{tri}}', \boldsymbol{p}, \boldsymbol{s}_i], y_i) + \lambda ||\boldsymbol{k}_{\text{tri}}'||^2 \\
&\ \text{s.t. } \boldsymbol{p} = \arg\min_{\boldsymbol{p}'} \sum_{j=1}^{N} \frac{1}{|D_j^{\text{tr}}|} \sum_{(\boldsymbol{s}_i, y_i) \sim D_j^{\text{tr}}} L(\mathcal{G}, [\boldsymbol{k}_{\text{tri}}', \boldsymbol{p}', \boldsymbol{s}_i], y_i) \\
\text{var. } &\ \lambda, \boldsymbol{k}_{\text{tri}}, \boldsymbol{p},
\end{aligned}
\tag{30}
$$

where $\mathcal{G}$ denotes the black-box LLM. $\lambda$, $\boldsymbol{k}_{\text{tri}}$, $\boldsymbol{p}$ respectively denote the hyperparameter, backdoor trigger, and prompt. $D_j^{\text{tr}}$ and $D_j^{\text{val}}$ denote the training and validation dataset in $j^{\text{th}}$ worker, and $N$ denotes the number of workers. $\boldsymbol{s}_i, y_i$ denote the $i^{\text{th}}$ input sentence and label. In the experiment, Qwen 1.8B-Chat (Bai et al., 2023) is utilized as the black-box LLM. The General Language Understanding Evaluation (GLUE) benchmark (Wang et al., 2018a) is used to evaluate the proposed DTZO. Specifically, the experiments are carried out on: 1) SST-2 for sentiment analysis; 2) COLA for linguistic acceptability; and 3) MRPC for semantic equivalence of sentences. In this task, we aim to obtain the effective backdoor triggers while ensuring the model performance on clean inputs (i.e., inputs without triggers). Therefore, following Yao et al. (2024), the Attack Success Rate

Table 1: Comparisons between the proposed DTZO and the state-of-the-art methods. Experiments are repeated five times and higher scores represent better performance.

| Dataset | FedZOO (Fang et al., 2022) | FedRZO$_{\mathrm{bl}}$ (Qiu et al., 2023) | **DTZO** |
|---------|---------------------------|------------------------|----------|
| MNIST | $52.89 \pm 0.49\%$ | $54.05 \pm 0.81\%$ | $\mathbf{79.27 \pm 0.19}\%$ |
| QMNIST | $52.45 \pm 0.88\%$ | $54.67 \pm 0.65\%$ | $\mathbf{78.04 \pm 0.37}\%$ |
| F-MNIST | $48.74 \pm 0.61\%$ | $50.23 \pm 0.49\%$ | $\mathbf{70.07 \pm 0.45}\%$ |
| USPS | $72.77 \pm 0.43\%$ | $73.79 \pm 0.56\%$ | $\mathbf{85.13 \pm 0.14}\%$ |

(ASR) when the triggers are activated and the Accuracy (ACC) on clean samples are utilized as the metrics in the experiments. The comparisons between the proposed DTZO and the state-of-the-art distributed bilevel zeroth order learning method FedRZO$_{\mathrm{bl}}$ are illustrated in Figure 1. It is seen from Figure 1(a) and 1(b) that the proposed DTZO can effectively tackle the distributed trilevel zeroth order learning problem and achieve superior performance than FedRZO$_{\mathrm{bl}}$ since the proposed DTZO is capable of addressing higher-nested zeroth order learning problems compared to FedRZO$_{\mathrm{bl}}$.

## 5.2 ROBUST HYPERPARAMETER OPTIMIZATION

Inspired by Sato et al. (2021); Jiao et al. (2024) in trilevel learning, the robust hyperparameter optimization is considered in the experiment, which can be formulated as follows.

$$
\begin{aligned}
&\min_{\varphi} \sum_{j=1}^{N} f_j(X_j^{\mathrm{var}}, y_j^{\mathrm{var}}, \boldsymbol{w}) \\
&\text{s.t. } \boldsymbol{w} = \arg\min_{\boldsymbol{w}'} \sum_{j=1}^{N} f_j(X_j^{\mathrm{tr}} + p_j, y_j^{\mathrm{tr}}, \boldsymbol{w}') + \varphi||\boldsymbol{w}'||^2 \\
&\qquad \text{s.t. } \boldsymbol{p} = \arg\max_{\boldsymbol{p}'} \sum_{j=1}^{N} f_j(X_j^{\mathrm{tr}} + p_j', y_j^{\mathrm{tr}}, \boldsymbol{w}') \\
&\text{var. } \qquad \varphi, \boldsymbol{w}, \boldsymbol{p},
\end{aligned}
\tag{31}
$$

where $N$ represents the number of workers in a distributed system, $\varphi$, $\boldsymbol{w}$, and $\boldsymbol{p}' = [p_1', \cdots, p_N']$ denote the regularization coefficient, model parameter, and adversarial noise, respectively. $X_j^{\mathrm{tr}}$ and $y_j^{\mathrm{tr}}$ represent the training data and labels, while $X_j^{\mathrm{var}}$ and $y_j^{\mathrm{var}}$ represent the validation data and labels, respectively. Following the setting for nondifferentiable functions as described in Qiu et al. (2023), ReLU neural networks are employed in the experiments. The digits recognition tasks in Qian et al. (2019); Wang et al. (2021) with four benchmark datasets, i.e., MNIST (LeCun et al., 1998), USPS, Fashion MNIST (Xiao et al., 2017), and QMNIST (Yadav and Bottou, 2019), are utilized to assess the performance of the proposed DTZO. The average across accuracy on clean samples and robustness against adversarial samples is used as the metric, more details about the experimental setting are provided in Appendix F. We compare the proposed DTZO with the state-of-the-art methods FedZOO (Fang et al., 2022) and FedRZO$_{\mathrm{bl}}$ (Qiu et al., 2023) in Table 1. It is seen from Table 1 that the proposed DTZO can effectively tackle the trilevel zeroth order learning problem in a distributed manner. The superior performance of DTZO, as compared to state-of-the-art methods, can be attributed to its ability to address higher-nested zeroth order learning problems.

Within the proposed framework, the trade-off between complexity and performance can be flexibly controlled by adjusting $T_1$, as discussed in Sec. 4. As shown in Figure 2 in Appendix F, the performance of DTZO improves as $T_1$ increases, we can flexibly adjust $T_1$ based on the distributed system requirements. Removing inactive cuts can significantly improve the effectiveness of cutting plane method, as discussed in Jiao et al. (2024); Yang et al. (2014). In the experiment, we also investigate the effect of removing inactive cuts within the proposed DTZO. It is seen from Figure 3 in Appendix F that pruning inactive cuts significantly reduces training time, indicating the importance of this procedure.

## 6 CONCLUSION

In this work, a distributed trilevel zeroth order learning (DTZO) framework is proposed to address the trilevel learning problems in a distributed manner without using first order information. To our best knowledge, this is the first work that considers how to tackle the trilevel zeroth order learning problems. The proposed DTZO is capable of constructing the cascaded polynomial approximation for trilevel zeroth order learning problems without using gradients or sub-gradients by utilizing the novel zeroth order cuts. Additionally, we theoretically analyze the non-asymptotic convergence rate for the proposed DTZO to achieve the $\epsilon$-stationary point. Experiments on black-box LLMs trilevel learning and robust hyperparameter optimization demonstrate the superior performance of DTZO.

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

## Appendix

To improve the readability of the Appendix, we have organized its contents as follows: In Appendix A and B, we delve into the comprehensive proofs of Theorem 1 (Iteration Complexity) and Theorem 2 (Communication Complexity). In Appendix C, the detailed proofs of Propositions 1 and 2 are provided. Furthermore, we offer the theoretical analyses about the cascaded polynomial approximation in Appendix D. Additionally, detailed discussions about the soft constraint are given in Appendix E, and the discussions about $\phi_{\text{in}}$ and $\phi_{\text{out}}$ are also conducted in this part. In Appendix F, details of the experimental setting and additional experimental results are provided. The discussions about Assumptions 1 and 2 are offered in Appendix G, we show that both Assumptions 1 and 2 are mild and widely-used in machine learning. In Appendix H, the reasons why we choose the exterior penalty method in the proposed framework are discussed, and we demonstrate the close relationship between the original constrained optimization problem and the unconstrained optimization problem. In Appendix I, we show that the proposed framework can be applied to a wide range of TLL problems, e.g., (grey-box) TLL with partial zeroth order constraints. More discussions about the cutting plane method and the choice of gradient estimator are provided in Appendix J. Lastly, the future work is discussed in Appendix K.

Furthermore, to enhance the readability of this work, the notations used in this work and their corresponding meanings are summarized in Table 2.

## Table of Contents

Table 2: Notations used in this work and the corresponding meanings.

| Notation | Meaning |
|---|---|
| $f_i(\cdot), \forall i = 1, 2, 3$ | $i^{\text{th}}$ level objective. |
| $\boldsymbol{x}_i, \forall i = 1, 2, 3$ | $i^{\text{th}}$ level variable. |
| $f_{i,j}(\cdot), \forall i = 1, 2, 3, j = 1, \cdots, N$ | $i^{\text{th}}$ level local objective in worker $j$. |
| $\boldsymbol{x}_{i,j}, \forall i = 1, 2, 3, j = 1, \cdots, N$ | $i^{\text{th}}$ level local variable in worker $j$. |
| $\boldsymbol{z}_i, \forall i = 1, 2, 3$ | $i^{\text{th}}$ level global variable in master. |
| $P_{\text{in}}, P_{\text{out}}$ | feasible regions formed by inner and outer layer zeroth order cuts. |
| $cp_{\text{in},l}, cp_{\text{out},l}$ | $l^{\text{th}}$ inner layer and outer layer zeroth order cuts. |
| $\boldsymbol{a}_{j,l}^{\text{in}}, \boldsymbol{b}_{j,l}^{\text{in}}, \boldsymbol{c}_{i,l}^{\text{in}}, \boldsymbol{d}_{i,l}^{\text{in}}, e_l^{\text{in}}$ | $l^{\text{th}}$ inner layer zeroth order cut's parameters. |
| $\boldsymbol{a}_{i,j,l}^{\text{out}}, \boldsymbol{b}_{i,j,l}^{\text{out}}, \boldsymbol{c}_{i,l}^{\text{out}}, \boldsymbol{d}_{i,l}^{\text{out}}, e_l^{\text{out}}$ | $l^{\text{th}}$ outer layer zeroth order cut's parameters. |
| $F(\cdot)$ | penalty function. |
| $F_\mu(\cdot)$ | smooth approximation of $F(\cdot)$. |
| $\mu$ | smoothing parameter. |
| $F_\mu{}^*$ | optimal objective value of $F_\mu(\cdot)$. |
| $\lambda_l, \phi_j$ | penalty parameters. |
| $\phi_{\text{in}}(\cdot), \phi_{\text{out}}(\cdot)$ | functions used in third level and second level constraint. |
| $G_{\boldsymbol{x}_{i,j}}, \forall i = 1, 2, 3, j = 1, \cdots, N$ | gradient estimator for $i^{\text{th}}$ level variable in worker $j$ |
| $\eta_{\boldsymbol{x}_i}, \eta_{\boldsymbol{z}_i}, \forall i = 1, 2, 3$ | step sizes for variables $\boldsymbol{x}_i, \boldsymbol{z}_i$. |
| $\boldsymbol{\mu}^{\text{in}}, \boldsymbol{\mu}^{\text{out}}, \boldsymbol{u}_{k,1}, \boldsymbol{u}_{k,2}, \boldsymbol{u}_{k,3}$ | standard Gaussian random vectors. |
| $\mathcal{G}^t$ | stationarity gap. |
| $T(\epsilon)$ | iteration complexity to achieve $\epsilon$-stationary point. |
| $T_1$ | parameter controls the trade-off between complexity and performance. |
| $\mathcal{T}$ | zeroth order cuts will be updated every $\mathcal{T}$ iteration. |
| $N$ | the number of workers in distributed systems. |
| $L$ | parameter in $L$-smoothness. |
| $d_i, \forall i = 1, 2, 3$ | the dimension of $i^{\text{th}}$ level variable. |

# A    PROOF OF THEOREM 1

In this section, the detailed proofs of Theorem 1, i.e., iteration complexity of the proposed DTZO, are offered. The iteration complexity refers to the number of iterations for the proposed algorithm to obtain the $\epsilon$-stationary point (Jiao et al., 2023). According to Ghadimi and Lan (2013), the gradient of the smooth approximation of $F$, i.e., $F_\mu$ (which is given in Definition 3), is also Lipschitz continuous with constant $L_\mu$ ($0 < L_\mu \leq L$), thus, we have that when $t \geq T_1$,

$$
F_\mu(\{\boldsymbol{x}_{i,j}^{t+1}\}, \{\boldsymbol{z}_i^t\})
$$

$$
\leq F_\mu(\{\boldsymbol{x}_i^t\}, \{\boldsymbol{z}_i^t\}) + \begin{bmatrix} \{\boldsymbol{x}_{1,j}^{t+1} - \boldsymbol{x}_{1,j}^t\} \\ \{\boldsymbol{x}_{2,j}^{t+1} - \boldsymbol{x}_{2,j}^t\} \\ \{\boldsymbol{x}_{3,j}^{t+1} - \boldsymbol{x}_{3,j}^t\} \end{bmatrix}^\top \begin{bmatrix} \{\nabla_{\boldsymbol{x}_{1,j}} F_\mu(\{\boldsymbol{x}_{i,j}^t\}, \{\boldsymbol{z}_i^t\})\} \\ \{\nabla_{\boldsymbol{x}_{2,j}} F_\mu(\{\boldsymbol{x}_{i,j}^t\}, \{\boldsymbol{z}_i^t\})\} \\ \{\nabla_{\boldsymbol{x}_{3,j}} F_\mu(\{\boldsymbol{x}_{i,j}^t\}, \{\boldsymbol{z}_i^t\})\} \end{bmatrix} + \frac{L}{2} || \begin{bmatrix} \{\boldsymbol{x}_{1,j}^{t+1} - \boldsymbol{x}_{1,j}^t\} \\ \{\boldsymbol{x}_{2,j}^{t+1} - \boldsymbol{x}_{2,j}^t\} \\ \{\boldsymbol{x}_{3,j}^{t+1} - \boldsymbol{x}_{3,j}^t\} \end{bmatrix} ||^2
$$

$$
= F_\mu(\{\boldsymbol{x}_{i,j}^t\}, \{\boldsymbol{z}_i^t\}) - \begin{bmatrix} \{\eta_{\boldsymbol{x}_1} G_{\boldsymbol{x}_{1,j}}(\{\boldsymbol{x}_{i,j}^t\}, \{\boldsymbol{z}_i^t\})\} \\ \{\eta_{\boldsymbol{x}_2} G_{\boldsymbol{x}_{2,j}}(\{\boldsymbol{x}_{i,j}^t\}, \{\boldsymbol{z}_i^t\})\} \\ \{\eta_{\boldsymbol{x}_3} G_{\boldsymbol{x}_{3,j}}(\{\boldsymbol{x}_{i,j}^t\}, \{\boldsymbol{z}_i^t\})\} \end{bmatrix}^T \begin{bmatrix} \{\nabla_{\boldsymbol{x}_{1,j}} F_\mu(\{\boldsymbol{x}_{i,j}^t\}, \{\boldsymbol{z}_i^t\})\} \\ \{\nabla_{\boldsymbol{x}_{2,j}} F_\mu(\{\boldsymbol{x}_{i,j}^t\}, \{\boldsymbol{z}_i^t\})\} \\ \{\nabla_{\boldsymbol{x}_{3,j}} F_\mu(\{\boldsymbol{x}_{i,j}^t\}, \{\boldsymbol{z}_i^t\})\} \end{bmatrix}
$$

$$
+ \frac{L}{2} \sum_{i=1}^{3} \sum_{j=1}^{N} \eta_{\boldsymbol{x}_i}^2 || G_{x_{i,j}}(\{\boldsymbol{x}_{i,j}^t\}, \{\boldsymbol{z}_i^t\}) ||^2. \tag{32}
$$

According to Assumption 2 (i.e., function $F$ has $L$-Lipschitz continuous gradient) and combining it with Cauchy-Schwarz inequality, we have that,

$$
F(\{\boldsymbol{x}_{i,j}^{t+1}\}, \{\boldsymbol{z}_i^{t+1}\})
$$

$$
\leq F(\{\boldsymbol{x}_{i,j}^{t+1}\}, \{\boldsymbol{z}_i^t\}) + \begin{bmatrix} \boldsymbol{z}_1^{t+1} - \boldsymbol{z}_1^t \\ \boldsymbol{z}_2^{t+1} - \boldsymbol{z}_2^t \\ \boldsymbol{z}_3^{t+1} - \boldsymbol{z}_3^t \end{bmatrix}^T \begin{bmatrix} \nabla_{\boldsymbol{z}_1} F(\{\boldsymbol{x}_{i,j}^{t+1}\}, \{\boldsymbol{z}_i^t\}) \\ \nabla_{\boldsymbol{z}_2} F(\{\boldsymbol{x}_{i,j}^{t+1}\}, \{\boldsymbol{z}_i^t\}) \\ \nabla_{\boldsymbol{z}_3} F(\{\boldsymbol{x}_{i,j}^{t+1}\}, \{\boldsymbol{z}_i^t\}) \end{bmatrix} + \frac{L}{2} || \begin{bmatrix} \boldsymbol{z}_1^{t+1} - \boldsymbol{z}_1^t \\ \boldsymbol{z}_2^{t+1} - \boldsymbol{z}_2^t \\ \boldsymbol{z}_3^{t+1} - \boldsymbol{z}_3^t \end{bmatrix} ||^2
$$

$$
= F(\{\boldsymbol{x}_{i,j}^{t+1}\}, \{\boldsymbol{z}_i^t\}) + \begin{bmatrix} \boldsymbol{z}_1^{t+1} - \boldsymbol{z}_1^t \\ \boldsymbol{z}_2^{t+1} - \boldsymbol{z}_2^t \\ \boldsymbol{z}_3^{t+1} - \boldsymbol{z}_3^t \end{bmatrix}^T \begin{bmatrix} \nabla_{\boldsymbol{z}_1} F(\{\boldsymbol{x}_{i,j}^t\}, \{\boldsymbol{z}_i^t\}) \\ \nabla_{\boldsymbol{z}_2} F(\{\boldsymbol{x}_{i,j}^t\}, \{\boldsymbol{z}_i^t\}) \\ \nabla_{\boldsymbol{z}_3} F(\{\boldsymbol{x}_{i,j}^t\}, \{\boldsymbol{z}_i^t\}) \end{bmatrix}
$$

$$
+ \begin{bmatrix} \boldsymbol{z}_1^{t+1} - \boldsymbol{z}_1^t \\ \boldsymbol{z}_2^{t+1} - \boldsymbol{z}_2^t \\ \boldsymbol{z}_3^{t+1} - \boldsymbol{z}_3^t \end{bmatrix}^T \begin{bmatrix} \nabla_{\boldsymbol{z}_1} F(\{\boldsymbol{x}_{i,j}^{t+1}\}, \{\boldsymbol{z}_i^t\}) - \nabla_{\boldsymbol{z}_1} F(\{\boldsymbol{x}_{i,j}^t\}, \{\boldsymbol{z}_i^t\}) \\ \nabla_{\boldsymbol{z}_2} F(\{\boldsymbol{x}_{i,j}^{t+1}\}, \{\boldsymbol{z}_i^t\}) - \nabla_{\boldsymbol{z}_2} F(\{\boldsymbol{x}_{i,j}^t\}, \{\boldsymbol{z}_i^t\}) \\ \nabla_{\boldsymbol{z}_3} F(\{\boldsymbol{x}_{i,j}^{t+1}\}, \{\boldsymbol{z}_i^t\}) - \nabla_{\boldsymbol{z}_3} F(\{\boldsymbol{x}_{i,j}^t\}, \{\boldsymbol{z}_i^t\}) \end{bmatrix} + \frac{L}{2} || \begin{bmatrix} \boldsymbol{z}_1^{t+1} - \boldsymbol{z}_1^t \\ \boldsymbol{z}_2^{t+1} - \boldsymbol{z}_2^t \\ \boldsymbol{z}_3^{t+1} - \boldsymbol{z}_3^t \end{bmatrix} ||^2
$$

$$
\leq F(\{\boldsymbol{x}_{i,j}^{t+1}\}, \{\boldsymbol{z}_i^t\}) - \sum_{i=1}^3 (\eta_{\boldsymbol{z}_i} - \frac{L\eta_{\boldsymbol{z}_i}^2}{2} - \frac{\eta_{\boldsymbol{z}_i}^2}{2}) ||\nabla_{\boldsymbol{z}_i} F(\{\boldsymbol{x}_{i,j}^t\}, \{\boldsymbol{z}_i^t\})||^2 + \sum_{i=1}^3 \sum_{j=1}^N \frac{L}{2} ||\boldsymbol{x}_{i,j}^{t+1} - \boldsymbol{x}_{i,j}^t||^2.
$$

$$(33)$$

Combining Eq. (33) with the Eq. (3.5) in Ghadimi and Lan (2013), we have that,

$$
F_\mu(\{\boldsymbol{x}_{i,j}^{t+1}\}, \{\boldsymbol{z}_i^{t+1}\}) - \frac{\mu^2 L(N+1)\sum_i d_i}{2}
$$

$$
\leq F(\{\boldsymbol{x}_{i,j}^{t+1}\}, \{\boldsymbol{z}_i^{t+1}\})
$$

$$
\leq F(\{\boldsymbol{x}_{i,j}^{t+1}\}, \{\boldsymbol{z}_i^t\}) - \sum_{i=1}^3 (\eta_{\boldsymbol{z}_i} - \frac{(L+1)\eta_{\boldsymbol{z}_i}^2}{2}) ||\nabla_{\boldsymbol{z}_i} F(\{\boldsymbol{x}_{i,j}^t\}, \{\boldsymbol{z}_i^t\})||^2 + \sum_{i=1}^3 \sum_{j=1}^N \frac{L}{2} ||\boldsymbol{x}_{i,j}^{t+1} - \boldsymbol{x}_{i,j}^t||^2
$$

$$
\leq F_\mu(\{\boldsymbol{x}_{i,j}^{t+1}\}, \{\boldsymbol{z}_i^t\}) - \sum_{i=1}^3 (\eta_{\boldsymbol{z}_i} - \frac{(L+1)\eta_{\boldsymbol{z}_i}^2}{2}) ||\nabla_{\boldsymbol{z}_i} F(\{\boldsymbol{x}_{i,j}^t\}, \{\boldsymbol{z}_i^t\})||^2 + \sum_{i=1}^3 \sum_{j=1}^N \frac{L}{2} ||\boldsymbol{x}_{i,j}^{t+1} - \boldsymbol{x}_{i,j}^t||^2
$$

$$
+ \frac{\mu^2 L(N+1)\sum_i d_i}{2}.
$$

$$(34)$$

Combining Eq. (32) with Eq. (34), we can obtain that,

$$
F_\mu(\{\boldsymbol{x}_{i,j}^{t+1}\}, \{\boldsymbol{z}_i^{t+1}\})
$$

$$
\leq F_\mu(\{\boldsymbol{x}_{i,j}^t\}, \{\boldsymbol{z}_i^t\}) - \begin{bmatrix} \{\eta_{\boldsymbol{x}_1} G_{\boldsymbol{x}_{1,j}}(\{\boldsymbol{x}_{i,j}^t\}, \{\boldsymbol{z}_i^t\})\} \\ \{\eta_{\boldsymbol{x}_1} G_{\boldsymbol{x}_{2,j}}(\{\boldsymbol{x}_{i,j}^t\}, \{\boldsymbol{z}_i^t\})\} \\ \{\eta_{\boldsymbol{x}_1} G_{\boldsymbol{x}_{3,j}}(\{\boldsymbol{x}_{i,j}^t\}, \{\boldsymbol{z}_i^t\})\} \end{bmatrix}^T \begin{bmatrix} \{\nabla_{\boldsymbol{x}_{1,j}} F_\mu(\{\boldsymbol{x}_{i,j}^t\}, \{\boldsymbol{z}_i^t\})\} \\ \{\nabla_{\boldsymbol{x}_{2,j}} F_\mu(\{\boldsymbol{x}_{i,j}^t\}, \{\boldsymbol{z}_i^t\})\} \\ \{\nabla_{\boldsymbol{x}_{3,j}} F_\mu(\{\boldsymbol{x}_{i,j}^t\}, \{\boldsymbol{z}_i^t\})\} \end{bmatrix}
$$

$$
+ \frac{L}{2} \sum_{i=1}^3 \sum_{j=1}^N \eta_{\boldsymbol{x}_i}^2 ||G_{x_{i,j}}(\{\boldsymbol{x}_{i,j}^t\}, \{\boldsymbol{z}_i^t\})||^2 - \sum_{i=1}^3 (\eta_{\boldsymbol{z}_i} - \frac{(L+1)\eta_{\boldsymbol{z}_i}^2}{2}) ||\nabla_{\boldsymbol{z}_i} F(\{\boldsymbol{x}_{i,j}^t\}, \{\boldsymbol{z}_i^t\})||^2
$$

$$
+ \sum_{i=1}^3 \sum_{j=1}^N \frac{L}{2} ||\boldsymbol{x}_{i,j}^{t+1} - \boldsymbol{x}_{i,j}^t||^2 + \mu^2 L(N+1)\sum_i d_i \qquad (35)
$$

$$
= F_\mu(\{\boldsymbol{x}_{i,j}^t\}, \{\boldsymbol{z}_i^t\}) - \begin{bmatrix} \{\eta_{\boldsymbol{x}_1} G_{\boldsymbol{x}_{1,j}}(\{\boldsymbol{x}_{i,j}^t\}, \{\boldsymbol{z}_i^t\})\} \\ \{\eta_{\boldsymbol{x}_1} G_{\boldsymbol{x}_{2,j}}(\{\boldsymbol{x}_{i,j}^t\}, \{\boldsymbol{z}_i^t\})\} \\ \{\eta_{\boldsymbol{x}_1} G_{\boldsymbol{x}_{3,j}}(\{\boldsymbol{x}_{i,j}^t\}, \{\boldsymbol{z}_i^t\})\} \end{bmatrix}^T \begin{bmatrix} \{\nabla_{\boldsymbol{x}_{1,j}} F_\mu(\{\boldsymbol{x}_{i,j}^t\}, \{\boldsymbol{z}_i^t\})\} \\ \{\nabla_{\boldsymbol{x}_{2,j}} F_\mu(\{\boldsymbol{x}_{i,j}^t\}, \{\boldsymbol{z}_i^t\})\} \\ \{\nabla_{\boldsymbol{x}_{3,j}} F_\mu(\{\boldsymbol{x}_{i,j}^t\}, \{\boldsymbol{z}_i^t\})\} \end{bmatrix}
$$

$$
+ \sum_{i=1}^3 \sum_{j=1}^N L\eta_{\boldsymbol{x}_i}^2 ||G_{x_{i,j}}(\{\boldsymbol{x}_{i,j}^t\}, \{\boldsymbol{z}_i^t\})||^2 - \sum_{i=1}^3 (\eta_{\boldsymbol{z}_i} - \frac{(L+1)\eta_{\boldsymbol{z}_i}^2}{2}) ||\nabla_{\boldsymbol{z}_i} F(\{\boldsymbol{x}_{i,j}^t\}, \{\boldsymbol{z}_i^t\})||^2
$$

$$
+ \mu^2 L(N+1)\sum_i d_i.
$$

Taking expectation on the both sides of Eq. (32), we can obtain that,

$$
\mathbb{E}[F_\mu(\{\boldsymbol{x}_{i,j}^{t+1}\}, \{\boldsymbol{z}_i^{t+1}\})]
$$

$$
\leq \mathbb{E}[F_\mu(\{\boldsymbol{x}_{i,j}^t\}, \{\boldsymbol{z}_i^t\})] - \sum_{i=1}^{3}\sum_{j=1}^{N} \eta_{\boldsymbol{x}_i} ||\nabla_{x_{i,j}} F_\mu(\{\boldsymbol{x}_{i,j}^t\}, \{\boldsymbol{z}_i^t\})||^2 + \mu^2 L(N+1)\sum_i d_i
$$

$$
+ \sum_{i=1}^{3}\sum_{j=1}^{N} L\eta_{\boldsymbol{x}_i}^2 \mathbb{E}[||G_{x_{i,j}}(\{\boldsymbol{x}_{i,j}^t\}, \{\boldsymbol{z}_i^t\})||^2] - \sum_{i=1}^{3}(\eta_{\boldsymbol{z}_i} - \tfrac{(L+1)\eta_{\boldsymbol{z}_i}^2}{2})||\nabla_{\boldsymbol{z}_i} F(\{\boldsymbol{x}_{i,j}^t\}, \{\boldsymbol{z}_i^t\})||^2.
$$

$$(36)$$

Combining the definition of $G_{\boldsymbol{x}_{1,j}}, G_{\boldsymbol{x}_{2,j}}, G_{\boldsymbol{x}_{3,j}}$ with the Eq. (3.12) in Ghadimi and Lan (2013), we have that,

$$
E[||G_{\boldsymbol{x}_{1,j}}(\{\boldsymbol{x}_{i,j}^t\}, \{\boldsymbol{z}_i^t\})||^2] \leq 2(d_1+4)||\nabla_{\boldsymbol{x}_{1,j}} F(\{\boldsymbol{x}_{i,j}^t\}, \{\boldsymbol{z}_i^t\})||^2 + \frac{\mu^2 L^2}{2}(d_1+6)^3, \quad (37)
$$

$$
E[||G_{\boldsymbol{x}_{2,j}}(\{\boldsymbol{x}_{i,j}^t\}, \{\boldsymbol{z}_i^t\})||^2] \leq 2(d_2+4)||\nabla_{\boldsymbol{x}_{2,j}} F(\{\boldsymbol{x}_{i,j}^t\}, \{\boldsymbol{z}_i^t\})||^2 + \frac{\mu^2 L^2}{2}(d_2+6)^3, \quad (38)
$$

$$
E[||G_{\boldsymbol{x}_{3,j}}(\{\boldsymbol{x}_{i,j}^t\}, \{\boldsymbol{z}_i^t\})||^2] \leq 2(d_3+4)||\nabla_{\boldsymbol{x}_{3,j}} F(\{\boldsymbol{x}_{i,j}^t\}, \{\boldsymbol{z}_i^t\})||^2 + \frac{\mu^2 L^2}{2}(d_3+6)^3. \quad (39)
$$

By combining Eq. (36) with Eq. (37), (38), and (39), we can get that,

$$
\mathbb{E}[F_\mu(\{\boldsymbol{x}_{i,j}^{t+1}\}, \{\boldsymbol{z}_i^{t+1}\})]
$$

$$
\leq \mathbb{E}[F_\mu(\{\boldsymbol{x}_{i,j}^t\}, \{\boldsymbol{z}_i^t\})] - \sum_{i=1}^{3}\sum_{j=1}^{N} \eta_{\boldsymbol{x}_i} ||\nabla_{x_{i,j}} F_\mu(\{\boldsymbol{x}_{i,j}^t\}, \{\boldsymbol{z}_i^t\})||^2 + \mu^2 L(N+1)\sum_i d_i
$$

$$
+ \sum_{i=1}^{3}\sum_{j=1}^{N} L\eta_{\boldsymbol{x}_i}^2 \left(2(d_i+4)||\nabla_{x_{i,j}} F(\{\boldsymbol{x}_{i,j}^t\}, \{\boldsymbol{z}_i^t\})||^2 + \tfrac{\mu^2 L^2}{2}(d_i+6)^3\right)
$$

$$
- \sum_{i=1}^{3}(\eta_{\boldsymbol{z}_i} - \tfrac{(L+1)\eta_{\boldsymbol{z}_i}^2}{2})||\nabla_{\boldsymbol{z}_i} F(\{\boldsymbol{x}_{i,j}^t\}, \{\boldsymbol{z}_i^t\})||^2,
$$

$$(40)$$

that is,

$$
\sum_{i=1}^{3}\sum_{j=1}^{N} \eta_{\boldsymbol{x}_i} ||\nabla_{x_{i,j}} F_\mu(\{\boldsymbol{x}_{i,j}^t\}, \{\boldsymbol{z}_i^t\})||^2 + \sum_{i=1}^{3}(\eta_{\boldsymbol{z}_i} - \tfrac{(L+1)\eta_{\boldsymbol{z}_i}^2}{2})||\nabla_{\boldsymbol{z}_i} F(\{\boldsymbol{x}_{i,j}^t\}, \{\boldsymbol{z}_i^t\})||^2
$$

$$
\leq \mathbb{E}[F_\mu(\{\boldsymbol{x}_{i,j}^t\}, \{\boldsymbol{z}_i^t\})] - \mathbb{E}[F_\mu(\{\boldsymbol{x}_{i,j}^{t+1}\}, \{\boldsymbol{z}_i^{t+1}\})] + \mu^2 L(N+1)\sum_i d_i
$$

$$
+ \sum_{i=1}^{3}\sum_{j=1}^{N} L\eta_{\boldsymbol{x}_i}^2 \left(2(d_i+4)||\nabla_{x_{i,j}} F(\{\boldsymbol{x}_{i,j}^t\}, \{\boldsymbol{z}_i^t\})||^2 + \tfrac{\mu^2 L^2}{2}(d_i+6)^3\right).
$$

$$(41)$$

Combining Eq. (41) with Eq. (3.8) in Ghadimi and Lan (2013), we can obtain that,

$$
\sum_{i=1}^{3}\sum_{j=1}^{N} \eta_{\boldsymbol{x}_i} \left(\tfrac{1}{2}||\nabla_{x_{i,j}} F(\{\boldsymbol{x}_{i,j}^t\}, \{\boldsymbol{z}_i^t\})||^2 - \tfrac{\mu^2 L^2}{4}(d_i+3)^3\right)
$$

$$
+ \sum_{i=1}^{3}(\eta_{\boldsymbol{z}_i} - \tfrac{(L+1)\eta_{\boldsymbol{z}_i}^2}{2})||\nabla_{\boldsymbol{z}_i} F(\{\boldsymbol{x}_{i,j}^t\}, \{\boldsymbol{z}_i^t\})||^2
$$

$$
\leq \sum_{i=1}^{3}\sum_{j=1}^{N} \eta_{\boldsymbol{x}_i} ||\nabla_{x_{i,j}} F_\mu(\{\boldsymbol{x}_{i,j}^t\}, \{\boldsymbol{z}_i^t\})||^2 + \sum_{i=1}^{3}(\eta_{\boldsymbol{z}_i} - \tfrac{(L+1)\eta_{\boldsymbol{z}_i}^2}{2})||\nabla_{\boldsymbol{z}_i} F(\{\boldsymbol{x}_{i,j}^t\}, \{\boldsymbol{z}_i^t\})||^2 \quad (42)
$$

$$
\leq \mathbb{E}[F_\mu(\{\boldsymbol{x}_{i,j}^t\}, \{\boldsymbol{z}_i^t\})] - \mathbb{E}[F_\mu(\{\boldsymbol{x}_{i,j}^{t+1}\}, \{\boldsymbol{z}_i^{t+1}\})] + \mu^2 L(N+1)\sum_i d_i
$$

$$
+ \sum_{i=1}^{3}\sum_{j=1}^{N} L\eta_{\boldsymbol{x}_i}^2 \left(2(d_i+4)||\nabla_{x_{i,j}} F(\{\boldsymbol{x}_{i,j}^t\}, \{\boldsymbol{z}_i^t\})||^2 + \tfrac{\mu^2 L^2}{2}(d_i+6)^3\right),
$$

that is,

$$
\begin{aligned}
&\sum_{i=1}^{3}\sum_{j=1}^{N}\left(\tfrac{\eta_{\boldsymbol{x}_i}}{2} - 2L(d_i+4)\eta_{\boldsymbol{x}_i}^2\right)||\nabla_{x_{i,j}}F(\{\boldsymbol{x}_{i,j}^t\},\{\boldsymbol{z}_i^t\})||^2 \\
&+\sum_{i=1}^{3}(\eta_{\boldsymbol{z}_i} - \tfrac{(L+1)\eta_{\boldsymbol{z}_i}^2}{2})||\nabla_{\boldsymbol{z}_i}F(\{\boldsymbol{x}_{i,j}^t\},\{\boldsymbol{z}_i^t\})||^2 \\
&\leq F_\mu(\{\boldsymbol{x}_{i,j}^t\},\{\boldsymbol{z}_i^t\}) - F_\mu(\{\boldsymbol{x}_{i,j}^{t+1}\},\{\boldsymbol{z}_i^{t+1}\}) + \sum_{i=1}^{3}\sum_{j=1}^{N}\tfrac{\eta_{\boldsymbol{x}_i}^2\mu^2 L^3}{2}(d_i+6)^3 \\
&+\sum_{i=1}^{3}\sum_{j=1}^{N}\tfrac{\mu^2 L^2 \eta_{\boldsymbol{x}_i}}{4}(d_i+3)^3 + \mu^2 L(N+1)\sum_i d_i.
\end{aligned}
\tag{43}
$$

According to the setting of $\eta_{\boldsymbol{x}_i}, i=1,2,3$, i.e., $0 < \eta_{\boldsymbol{x}_i} \leq \frac{1}{8L(d_i+4)}, i=1,2,3$, we have that,

$$
\frac{\eta_{\boldsymbol{x}_i}}{2} - 2L(d_i+4)\eta_{\boldsymbol{x}_i}^2 > 0, i=1,2,3.
\tag{44}
$$

Likewise, according to the setting of $\eta_{\boldsymbol{z}_i}, i=1,2,3$, i.e., $0 < \eta_{\boldsymbol{z}_i} \leq \frac{3}{2(L+1)}, i=1,2,3$, we have that,

$$
\eta_{\boldsymbol{z}_i} - \frac{(L+1)\eta_{\boldsymbol{z}_i}^2}{2} > 0, i=1,2,3.
\tag{45}
$$

Combining Eq. (43) with Eq. (44) and (45), we can obtain that,

$$
\begin{aligned}
&\sum_{i=1}^{3}\sum_{j=1}^{N}||\nabla_{x_{i,j}}F(\{\boldsymbol{x}_{i,j}^t\},\{\boldsymbol{z}_i^t\})||^2 + \sum_{i=1}^{3}||\nabla_{\boldsymbol{z}_i}F(\{\boldsymbol{x}_{i,j}^t\},\{\boldsymbol{z}_i^t\})||^2 \\
&\leq \frac{\sum_{i=1}^{3}\sum_{j=1}^{N}\left(\tfrac{\eta_{\boldsymbol{x}_i}}{2} - 2L(d_i+4)\eta_{\boldsymbol{x}_i}^2\right)||\nabla_{x_{i,j}}F(\{\boldsymbol{x}_{i,j}^t\},\{\boldsymbol{z}_i^t\})||^2}{\min\left\{\tfrac{\eta_{\boldsymbol{x}_i}}{2} - 2L(d_i+4)\eta_{\boldsymbol{x}_i}^2, \eta_{\boldsymbol{z}_i} - \tfrac{(L+1)\eta_{\boldsymbol{z}_i}^2}{2}, i=1,2,3\right\}} \\
&+ \frac{\sum_{i=1}^{3}(\eta_{\boldsymbol{z}_i} - \tfrac{(L+1)\eta_{\boldsymbol{z}_i}^2}{2})||\nabla_{\boldsymbol{z}_i}F(\{\boldsymbol{x}_{i,j}^t\},\{\boldsymbol{z}_i^t\})||^2}{\min\left\{\tfrac{\eta_{\boldsymbol{x}_i}}{2} - 2L(d_i+4)\eta_{\boldsymbol{x}_i}^2, \eta_{\boldsymbol{z}_i} - \tfrac{(L+1)\eta_{\boldsymbol{z}_i}^2}{2}, i=1,2,3\right\}} \\
&\leq \frac{F_\mu(\{\boldsymbol{x}_{i,j}^t\},\{\boldsymbol{z}_i^t\}) - F_\mu(\{\boldsymbol{x}_{i,j}^{t+1}\},\{\boldsymbol{z}_i^{t+1}\}) + \sum_{i=1}^{3}\tfrac{\eta_{\boldsymbol{x}_i}^2\mu^2 L^3 N}{2}(d_i+6)^3}{\min\left\{\tfrac{\eta_{\boldsymbol{x}_i}}{2} - 2L(d_i+4)\eta_{\boldsymbol{x}_i}^2, \eta_{\boldsymbol{z}_i} - \tfrac{(L+1)\eta_{\boldsymbol{z}_i}^2}{2}, i=1,2,3\right\}} \\
&+ \frac{\sum_{i=1}^{3}\tfrac{\mu^2 L^2 \eta_{\boldsymbol{x}_i} N}{4}(d_i+3)^3 + \mu^2 L(N+1)\sum_i d_i}{\min\left\{\tfrac{\eta_{\boldsymbol{x}_i}}{2} - 2L(d_i+4)\eta_{\boldsymbol{x}_i}^2, \eta_{\boldsymbol{z}_i} - \tfrac{(L+1)\eta_{\boldsymbol{z}_i}^2}{2}, i=1,2,3\right\}}.
\end{aligned}
\tag{46}
$$

Summing up the inequality in Eq. (46) from $t=T_1$ to $t=T(\epsilon)-1$, we have that,

$$
\frac{1}{T(\epsilon) - T_1} \sum_{t=T_1}^{T(\epsilon)-1} \left( \sum_{i=1}^{3} \sum_{j=1}^{N} ||\nabla_{x_{i,j}} F(\{\boldsymbol{x}_{i,j}^t\}, \{\boldsymbol{z}_i^t\})||^2 + \sum_{i=1}^{3} ||\nabla_{\boldsymbol{z}_i} F(\{\boldsymbol{x}_{i,j}^t\}, \{\boldsymbol{z}_i^t\})||^2 \right)
$$

$$
\leq \frac{F_\mu(\{\boldsymbol{x}_{i,j}^{T_1}\}, \{\boldsymbol{z}_i^{T_1}\}) - F_\mu(\{\boldsymbol{x}_{i,j}^{T(\epsilon)}\}, \{\boldsymbol{z}_i^{T(\epsilon)}\})}{\min \left\{ \frac{\eta_{\boldsymbol{x}_i}}{2} - 2L(d_i+4)\eta_{\boldsymbol{x}_i}^2, \eta_{\boldsymbol{z}_i} - \frac{(L+1)\eta_{\boldsymbol{z}_i}^2}{2}, i=1,2,3 \right\} (T(\epsilon) - T_1)}
$$

$$
+ \frac{\sum\limits_{i=1}^{3} \frac{\eta_{\boldsymbol{x}_i}^2 \mu^2 L^3 N}{2}(d_i+6)^3 + \sum\limits_{i=1}^{3} \frac{\mu^2 L^2 \eta_{\boldsymbol{x}_i} N}{4}(d_i+3)^3 + \mu^2 L(N+1)\sum_i d_i}{\min \left\{ \frac{\eta_{\boldsymbol{x}_i}}{2} - 2L(d_i+4)\eta_{\boldsymbol{x}_i}^2, \eta_{\boldsymbol{z}_i} - \frac{(L+1)\eta_{\boldsymbol{z}_i}^2}{2}, i=1,2,3 \right\}} \tag{47}
$$

$$
\leq \frac{\max\limits_{t \in [T_1]} F_\mu(\{\boldsymbol{x}_{i,j}^t\}, \{\boldsymbol{z}_i^t\}) - F_\mu^*}{\min \left\{ \frac{\eta_{\boldsymbol{x}_i}}{2} - 2L(d_i+4)\eta_{\boldsymbol{x}_i}^2, \eta_{\boldsymbol{z}_i} - \frac{(L+1)\eta_{\boldsymbol{z}_i}^2}{2}, i=1,2,3 \right\} (T(\epsilon) - T_1)}
$$

$$
+ \frac{\sum\limits_{i=1}^{3} \frac{\eta_{\boldsymbol{x}_i}^2 \mu^2 L^3 N}{2}(d_i+6)^3 + \sum\limits_{i=1}^{3} \frac{\mu^2 L^2 \eta_{\boldsymbol{x}_i} N}{4}(d_i+3)^3 + \mu^2 L(N+1)\sum_i d_i}{\min \left\{ \frac{\eta_{\boldsymbol{x}_i}}{2} - 2L(d_i+4)\eta_{\boldsymbol{x}_i}^2, \eta_{\boldsymbol{z}_i} - \frac{(L+1)\eta_{\boldsymbol{z}_i}^2}{2}, i=1,2,3 \right\}}.
$$

According to the setting of $\eta_{\boldsymbol{x}_i}, \eta_{\boldsymbol{z}_i}, i = 1, 2, 3$, we can obtain that,

$$
\frac{\eta_{\boldsymbol{x}_i}}{2} - 2L(d_i+4)\eta_{\boldsymbol{x}_i}^2 = \eta_{\boldsymbol{x}_i}\left(\frac{1}{2} - 2L(d_i+4)\eta_{\boldsymbol{x}_i}\right) \geq \frac{\eta_{\boldsymbol{x}_i}}{4}, i=1,2,3, \tag{48}
$$

$$
\eta_{\boldsymbol{z}_i} - \frac{(L+1)\eta_{\boldsymbol{z}_i}^2}{2} = \eta_{\boldsymbol{z}_i}(1 - \frac{(L+1)\eta_{\boldsymbol{z}_i}}{2}) \geq \frac{\eta_{\boldsymbol{z}_i}}{4}, i=1,2,3. \tag{49}
$$

Thus, we have that,

$$
\frac{1}{T(\epsilon) - T_1} \sum_{t=T_1}^{T(\epsilon)-1} \left( \sum_{i=1}^{3} \sum_{j=1}^{N} ||\nabla_{x_{i,j}} F(\{\boldsymbol{x}_{i,j}^t\}, \{\boldsymbol{z}_i^t\})||^2 + \sum_{i=1}^{3} ||\nabla_{\boldsymbol{z}_i} F(\{\boldsymbol{x}_{i,j}^t\}, \{\boldsymbol{z}_i^t\})||^2 \right)
$$

$$
\leq \frac{4\left( \max\limits_{t \in [T_1]} F_\mu(\{\boldsymbol{x}_{i,j}^t\}, \{\boldsymbol{z}_i^t\}) - F_\mu^* \right)}{\min\{\eta_{\boldsymbol{x}_1}, \eta_{\boldsymbol{x}_2}, \eta_{\boldsymbol{x}_3}, \eta_{\boldsymbol{z}_1}, \eta_{\boldsymbol{z}_2}, \eta_{\boldsymbol{z}_3}\} (T(\epsilon) - T_1)} \tag{50}
$$

$$
+ \frac{\sum\limits_{i=1}^{3} 2\eta_{\boldsymbol{x}_i}^2 \mu^2 L^3 N(d_i+6)^3 + \sum\limits_{i=1}^{3} \mu^2 L^2 \eta_{\boldsymbol{x}_i} N(d_i+3)^3 + 4\mu^2 L(N+1)\sum_i d_i}{\min\{\eta_{\boldsymbol{x}_1}, \eta_{\boldsymbol{x}_2}, \eta_{\boldsymbol{x}_3}, \eta_{\boldsymbol{z}_1}, \eta_{\boldsymbol{z}_2}, \eta_{\boldsymbol{z}_3}\}}.
$$

According to the setting that,

$$
\eta_{\boldsymbol{x}_i} = \eta_{\boldsymbol{z}_i} = \min \left\{ \frac{1}{8L(d_1+4)}, \frac{1}{8L(d_2+4)}, \frac{1}{8L(d_3+4)}, \frac{3}{2(L+1)}, \frac{1}{\sqrt{T(\epsilon)-T_1}} \right\}, i=1,2,3, \tag{51}
$$

we have that,

$$
\frac{1}{T(\epsilon) - T_1} \sum_{t=T_1}^{T(\epsilon)-1} \left( \sum_{i=1}^{3} \sum_{j=1}^{N} \|\nabla_{x_{i,j}} F(\{\boldsymbol{x}_{i,j}^t\}, \{\boldsymbol{z}_i^t\})\|^2 + \sum_{i=1}^{3} \|\nabla_{\boldsymbol{z}_i} F(\{\boldsymbol{x}_{i,j}^t\}, \{\boldsymbol{z}_i^t\})\|^2 \right)
$$

$$
\leq \frac{4 \left( \max_{t \in [T_1]} F_\mu(\{\boldsymbol{x}_{i,j}^t\}, \{\boldsymbol{z}_i^t\}) - F_\mu{}^* \right)}{\min \left\{ \frac{1}{8L(d_1+4)}, \frac{1}{8L(d_2+4)}, \frac{1}{8L(d_3+4)}, \frac{3}{2(L+1)}, \frac{1}{\sqrt{T(\epsilon)-T_1}} \right\} (T(\epsilon) - T_1)}
$$

$$
+ \sum_{i=1}^{3} 2\eta_{\boldsymbol{x}_i} \mu^2 L^3 N (d_i+6)^3 + \sum_{i=1}^{3} \mu^2 L^2 N (d_i+3)^3
$$

$$
+ \sum_{i=1}^{3} 4\mu^2 L(N+1) d_i \frac{1}{\min \left\{ \frac{1}{8L(d_1+4)}, \frac{1}{8L(d_2+4)}, \frac{1}{8L(d_3+4)}, \frac{3}{2(L+1)}, \frac{1}{\sqrt{T(\epsilon)-T_1}} \right\}}
$$

$$
\leq \frac{4 \left( \max_{t \in [T_1]} F_\mu(\{\boldsymbol{x}_{i,j}^t\}, \{\boldsymbol{z}_i^t\}) - F_\mu{}^* \right) \left( \max \left\{ 8L(d_1+4), 8L(d_2+4), 8L(d_3+4), \frac{2(L+1)}{3} \right\} \right)}{T(\epsilon) - T_1}
$$

$$
+ \frac{4 \left( \max_{t \in [T_1]} F_\mu(\{\boldsymbol{x}_{i,j}^t\}, \{\boldsymbol{z}_i^t\}) - F_\mu{}^* \right) \sqrt{T(\epsilon) - T_1}}{T(\epsilon) - T_1}
$$

$$
+ \sum_{i=1}^{3} 2\eta_{\boldsymbol{x}_i} \mu^2 L^3 N (d_i+6)^3 + \sum_{i=1}^{3} \mu^2 L^2 N (d_i+3)^3
$$

$$
+ \sum_{i=1}^{3} 4\mu^2 L(N+1) d_i \left( \max \left\{ 8L(d_1+4), 8L(d_2+4), 8L(d_3+4), \frac{2(L+1)}{3} \right\} + \sqrt{T(\epsilon) - T_1} \right).
$$

$$(52)$$

Since $\eta_{\boldsymbol{x}_i} \leq \frac{1}{8L(d_i+4)}, i = 1, 2, 3$, we can obtain that,

$$
\frac{1}{T(\epsilon) - T_1} \sum_{t=T_1}^{T(\epsilon)-1} \left( \sum_{i=1}^{3} \sum_{j=1}^{N} \|\nabla_{x_{i,j}} F(\{\boldsymbol{x}_{i,j}^t\}, \{\boldsymbol{z}_i^t\})\|^2 + \sum_{i=1}^{3} \|\nabla_{\boldsymbol{z}_i} F(\{\boldsymbol{x}_{i,j}^t\}, \{\boldsymbol{z}_i^t\})\|^2 \right)
$$

$$
\leq \frac{4 \left( \max_{t \in [T_1]} F_\mu(\{\boldsymbol{x}_{i,j}^t\}, \{\boldsymbol{z}_i^t\}) - F_\mu{}^* \right) \left( \max \left\{ 8L(d_1+4), 8L(d_2+4), 8L(d_3+4), \frac{2(L+1)}{3} \right\} \right)}{T(\epsilon) - T_1}
$$

$$
+ \frac{4 \left( \max_{t \in [T_1]} F_\mu(\{\boldsymbol{x}_{i,j}^t\}, \{\boldsymbol{z}_i^t\}) - F_\mu{}^* \right) \sqrt{T(\epsilon) - T_1}}{T(\epsilon) - T_1} + \frac{\mu^2 L^2 N}{4} \sum_{i=1}^{3} \frac{(d_i+6)^3}{d_i+4} + \mu^2 L^2 \sum_{i=1}^{3} (d_i+3)^3
$$

$$
+ \sum_{i=1}^{3} 4\mu^2 L(N+1) d_i \left( \max \left\{ 8L(d_1+4), 8L(d_2+4), 8L(d_3+4), \frac{2(L+1)}{3} \right\} + \sqrt{T(\epsilon) - T_1} \right).
$$

$$(53)$$

Because of $T(\epsilon) - T_1 \geq 1$, we have that $\frac{1}{T(\epsilon)-T_1} \leq \frac{1}{\sqrt{T(\epsilon)-T_1}}$. Combining with the setting of $\mu$, i.e., $\mu^2 \leq \frac{1}{T(\epsilon)-T_1}$, we can obtain that,

$$
\frac{1}{T(\epsilon) - T_1} \sum_{t=T_1}^{T(\epsilon)-1} (\sum_{i=1}^{3} \sum_{j=1}^{N} ||\nabla_{x_{i,j}} F(\{\boldsymbol{x}_{i,j}^t\}, \{\boldsymbol{z}_i^t\})||^2 + \sum_{i=1}^{3} ||\nabla_{\boldsymbol{z}_i} F(\{\boldsymbol{x}_{i,j}^t\}, \{\boldsymbol{z}_i^t\})||^2)
$$

$$
\leq \frac{4\max\left\{8L(d_1+4), 8L(d_2+4), 8L(d_3+4), \frac{2(L+1)}{3}\right\} \left(\max_{t\in[T_1]} F_\mu(\{\boldsymbol{x}_{i,j}^t\}, \{\boldsymbol{z}_i^t\}) - F_\mu^*\right)}{T(\epsilon) - T_1}
$$

$$
+ \frac{\max_{t\in[T_1]} F_\mu(\{\boldsymbol{x}_{i,j}^t\}, \{\boldsymbol{z}_i^t\}) - F_\mu^*}{\sqrt{T(\epsilon) - T_1}} + \frac{L^2}{4} \sum_{i=1}^{3} \frac{(d_i+6)^3}{d_i+4} \frac{1}{T(\epsilon) - T_1} + L^2 \sum_{i=1}^{3} (d_i+3)^3 \frac{1}{T(\epsilon) - T_1}
$$

$$
+ \sum_{i=1}^{3} \left(\max\left\{8L(d_1+4), 8L(d_2+4), 8L(d_3+4), \frac{2(L+1)}{3}\right\} + \sqrt{T(\epsilon) - T_1}\right) \frac{4L(N+1)d_i}{T(\epsilon) - T_1}
$$

$$
\leq \frac{4(1 + \max\left\{8L(d_1+4), 8L(d_2+4), 8L(d_3+4), \frac{2(L+1)}{3}\right\}) \left(\max_{t\in[T_1]} F_\mu(\{\boldsymbol{x}_{i,j}^t\}, \{\boldsymbol{z}_i^t\}) - F_\mu^*\right)}{\sqrt{T(\epsilon) - T_1}}
$$

$$
+ \frac{L^2}{4} \sum_{i=1}^{3} \frac{(d_i+6)^3}{d_i+4} \frac{1}{\sqrt{T(\epsilon) - T_1}} + L^2 \sum_{i=1}^{3} (d_i+3)^3 \frac{1}{\sqrt{T(\epsilon) - T_1}}
$$

$$
+ \sum_{i=1}^{3} \left(\max\left\{8L(d_1+4), 8L(d_2+4), 8L(d_3+4), \frac{2(L+1)}{3}\right\} + 1\right) 4L(N+1)d_i \frac{1}{\sqrt{T(\epsilon) - T_1}}.
$$

$$(54)$$

Combining the definition of stationarity gap and $\epsilon$-stationary point in Definition 1, 2 with Eq. (54), we have that,

$$
||\mathcal{G}^{T(\epsilon)}||^2
$$

$$
= \sum_{i=1}^{3} \sum_{j=1}^{N} ||\nabla_{x_{i,j}} F(\{\boldsymbol{x}_{i,j}^{T(\epsilon)}\}, \{\boldsymbol{z}_i^{T(\epsilon)}\})||^2 + \sum_{i=1}^{3} ||\nabla_{\boldsymbol{z}_i} F(\{\boldsymbol{x}_{i,j}^{T(\epsilon)}\}, \{\boldsymbol{z}_i^{T(\epsilon)}\})||^2
$$

$$
\leq \frac{1}{T(\epsilon) - T_1} \sum_{t=T_1}^{T(\epsilon)-1} (\sum_{i=1}^{3} \sum_{j=1}^{N} ||\nabla_{x_{i,j}} F(\{\boldsymbol{x}_{i,j}^t\}, \{\boldsymbol{z}_i^t\})||^2 + \sum_{i=1}^{3} ||\nabla_{\boldsymbol{z}_i} F(\{\boldsymbol{x}_{i,j}^t\}, \{\boldsymbol{z}_i^t\})||^2)
$$

$$
\leq \frac{4(1 + \max\left\{8L(d_1+4), 8L(d_2+4), 8L(d_3+4), \frac{2(L+1)}{3}\right\}) \left(\max_{t\in[T_1]} F_\mu(\{\boldsymbol{x}_{i,j}^t\}, \{\boldsymbol{z}_i^t\}) - F_\mu^*\right)}{\sqrt{T(\epsilon) - T_1}}
$$

$$
+ \frac{L^2}{4} \sum_{i=1}^{3} \frac{(d_i+6)^3}{d_i+4} \frac{1}{\sqrt{T(\epsilon) - T_1}} + L^2 \sum_{i=1}^{3} (d_i+3)^3 \frac{1}{\sqrt{T(\epsilon) - T_1}}
$$

$$
+ \sum_{i=1}^{3} \left(\max\left\{8L(d_1+4), 8L(d_2+4), 8L(d_3+4), \frac{2(L+1)}{3}\right\} + 1\right) 4L(N+1)d_i \frac{1}{\sqrt{T(\epsilon) - T_1}}.
$$

$$(55)$$

Thus, we can conclude that, when

$$
T(\epsilon) \geq \left(\sum_{i=1}^{3} \overline{c_i} + \overline{d} \left(\max_{t\in[T_1]} F_\mu(\{\boldsymbol{x}_{i,j}^t\}, \{\boldsymbol{z}_i^t\}) - F_\mu^*\right)\right)^2 \frac{1}{\epsilon^2} + T_1 , \tag{56}
$$

we have that $||\mathcal{G}^{T(\epsilon)}||^2 \leq \epsilon$, where constants

$$
\overline{d} = 4(1 + \max\left\{8L(d_1+4), 8L(d_2+4), 8L(d_3+4), \frac{2(L+1)}{3}\right\}), \tag{57}
$$

$$\overline{c_i} = \frac{L^2(d_i+6)^3}{4(d_i+4)} + L^2(d_i+3)^3$$

$$+ 4L(N+1)d_i \left( \max \left\{ 8L(d_1+4), 8L(d_2+4), 8L(d_3+4), \frac{2(L+1)}{3} \right\} + 1 \right). \tag{58}$$

## B  COMMUNICATION COMPLEXITY

The overall communication complexity of the proposed DTZO can be divided into 1) the communication complexity at every communication round and 2) the communication complexity of updating zeroth order cuts, which is discussed as follows.

1) The communication complexity at each iteration.

At each iteration, e.g., $(t+1)^{\text{th}}$ iteration, the workers transmit the updated variables $\boldsymbol{x}_{1,j}^{t+1}, \boldsymbol{x}_{2,j}^{t+1}, \boldsymbol{x}_{3,j}^{t+1}$ to the master, resulting in a communication complexity of $\sum_{j=1}^{N} \sum_{i=1}^{3} d_i$. Upon receiving these updated local variables, the master proceeds to update the global variables. Then, the master broadcasts the updated variables $\boldsymbol{z}_1^{t+1}, \boldsymbol{z}_2^{t+1}, \boldsymbol{z}_3^{t+1}$ and gradients $\nabla_{\boldsymbol{x}_{i,j}} o(\{\boldsymbol{x}_{2,j}^{t+1}\}, \{\boldsymbol{x}_{3,j}^{t+1}\}, \boldsymbol{z}_1^{t+1}, \boldsymbol{z}_2^{t+1}, \boldsymbol{z}_3^{t+1}), i = 2, 3$ to worker $j$. Therefore, the cumulative communication complexity from $t = 1$ to $t = T(\epsilon)$ is

$$C_1 = T(\epsilon)(2d_1 + 3d_2 + 3d_3)N. \tag{59}$$

2) The communication complexity of updating zeroth order cuts.

During every iteration $\mathcal{T}$ $(t < T_1)$, the cutting planes are updated to refine the cascaded polynomial approximation, involving two main steps:

2a) Updating the inner layer polynomial approximation: In this phase, local variables $\boldsymbol{x}_{3,j}^{k+1}$ are transmitted from worker $j$, while global variables $\boldsymbol{z}_3^{k+1}$ are sent from the master in the $(k+1)^{\text{th}}$ iteration. The communication complexity associated with updating the inner layer polynomial approximation can be expressed as follows:

$$\sum_{j=1}^{N} 2\lfloor \frac{T_1}{\mathcal{T}} \rfloor \mathcal{T} K d_3. \tag{60}$$

2b) Updating the outer layer polynomial approximation: During the $(k+1)^{\text{th}}$ iteration when updating the outer layer approximation, the worker $j$ transmits the updated variables $\boldsymbol{x}_{2,j}^{k+1}$, to the master. Subsequently, the master broadcasts the updated global variables $\boldsymbol{z}_2^{k+1}$ to worker $j$. The communication complexity involved in this process can be expressed as,

$$\sum_{j=1}^{N} 2\lfloor \frac{T_1}{\mathcal{T}} \rfloor \mathcal{T} K d_2. \tag{61}$$

Combining Eq. (60) with (61), and considering utilizing one communication round to approximate the $\phi_{\text{in}}(\{\boldsymbol{x}_{3,j}\}, \boldsymbol{z}_1, \boldsymbol{z}_2, \boldsymbol{z}_3)$ and $\phi_{\text{out}}(\{\boldsymbol{x}_{2,j}\}, \{\boldsymbol{x}_{3,j}\}, \boldsymbol{z}_1, \boldsymbol{z}_2, \boldsymbol{z}_3)$, i.e., $K = 1$, we have that the communication complexity of updating cascaded polynomial approximation is,

$$C_2 = 2N\lfloor \frac{T_1}{\mathcal{T}} \rfloor \mathcal{T}(d_2 + d_3). \tag{62}$$

Consequently, the overall communication of the proposed method is $C_1 + C_2$, which can be expressed as,

$$3T(\epsilon)(d_1 + d_2 + d_3)N + 2N\lfloor \frac{T_1}{\mathcal{T}} \rfloor \mathcal{T}(d_2 + d_3). \tag{63}$$

# C   PROOF OF PROPOSITION 1 AND 2

## C.1   PROOF OF PROPOSITION 1

For any point $(\{\boldsymbol{x}_{3,j}\}, \boldsymbol{z}_1, \boldsymbol{z}_2{'}, \boldsymbol{z}_3)$ in the original feasible region, i.e., $\phi_{\text{in}}(\{\boldsymbol{x}_{3,j}\}, \boldsymbol{z}_1, \boldsymbol{z}_2{'}, \boldsymbol{z}_3) = 0$, according to the properties of $L$-smoothness, we have that,

$$
\begin{aligned}
&\phi_{\text{in}}(\{\boldsymbol{x}_{3,j}\}, \boldsymbol{z}_1, \boldsymbol{z}_2{'}, \boldsymbol{z}_3) \\
&\geq \phi_{\text{in}}(\{\boldsymbol{x}_{3,j}^t\}, \boldsymbol{z}_1^t, \boldsymbol{z}_2^{t'}, \boldsymbol{z}_3^t) + \frac{\partial \phi_{\text{in}}(\{\boldsymbol{x}_{3,j}^t\}, \boldsymbol{z}_1^t, \boldsymbol{z}_2^{t'}, \boldsymbol{z}_3^t)}{\partial(\{\boldsymbol{x}_{3,j}\}, \boldsymbol{z}_1, \boldsymbol{z}_2{'}, \boldsymbol{z}_3)}^\top \left( \begin{bmatrix} \{\boldsymbol{x}_{3,j}\} \\ \boldsymbol{z}_1 \\ \boldsymbol{z}_2{'} \\ \boldsymbol{z}_3 \end{bmatrix} - \begin{bmatrix} \{\boldsymbol{x}_{3,j}^t\} \\ \boldsymbol{z}_1^t \\ \boldsymbol{z}_2^{t'} \\ \boldsymbol{z}_3^t \end{bmatrix} \right) \\
&\quad - \frac{L}{2} \left\| \left( \begin{bmatrix} \{\boldsymbol{x}_{3,j}\} \\ \boldsymbol{z}_1 \\ \boldsymbol{z}_2{'} \\ \boldsymbol{z}_3 \end{bmatrix} - \begin{bmatrix} \{\boldsymbol{x}_{3,j}^t\} \\ \boldsymbol{z}_1^t \\ \boldsymbol{z}_2^{t'} \\ \boldsymbol{z}_3^t \end{bmatrix} \right) \right\|^2 \\
&= \phi_{\text{in}}(\{\boldsymbol{x}_{3,j}^t\}, \boldsymbol{z}_1^t, \boldsymbol{z}_2^{t'}, \boldsymbol{z}_3^t) + G_\mu^{\text{in}}(\{\boldsymbol{x}_{3,j}^t\}, \boldsymbol{z}_1^t, \boldsymbol{z}_2^{t'}, \boldsymbol{z}_3^t)^\top \left( \begin{bmatrix} \{\boldsymbol{x}_{3,j}\} \\ \boldsymbol{z}_1 \\ \boldsymbol{z}_2{'} \\ \boldsymbol{z}_3 \end{bmatrix} - \begin{bmatrix} \{\boldsymbol{x}_{3,j}^t\} \\ \boldsymbol{z}_1^t \\ \boldsymbol{z}_2^{t'} \\ \boldsymbol{z}_3^t \end{bmatrix} \right) \\
&\quad + \left( \frac{\partial \phi_{\text{in}}(\{\boldsymbol{x}_{3,j}^t\}, \boldsymbol{z}_1^t, \boldsymbol{z}_2^{t'}, \boldsymbol{z}_3^t)}{\partial(\{\boldsymbol{x}_{3,j}\}, \boldsymbol{z}_1, \boldsymbol{z}_2{'}, \boldsymbol{z}_3)} - G_\mu^{\text{in}}(\{\boldsymbol{x}_{3,j}^t\}, \boldsymbol{z}_1^t, \boldsymbol{z}_2^{t'}, \boldsymbol{z}_3^t) \right)^\top \left( \begin{bmatrix} \{\boldsymbol{x}_{3,j}\} \\ \boldsymbol{z}_1 \\ \boldsymbol{z}_2{'} \\ \boldsymbol{z}_3 \end{bmatrix} - \begin{bmatrix} \{\boldsymbol{x}_{3,j}^t\} \\ \boldsymbol{z}_1^t \\ \boldsymbol{z}_2^{t'} \\ \boldsymbol{z}_3^t \end{bmatrix} \right) \\
&\quad - \frac{L}{2} \left\| \left( \begin{bmatrix} \{\boldsymbol{x}_{3,j}\} \\ \boldsymbol{z}_1 \\ \boldsymbol{z}_2{'} \\ \boldsymbol{z}_3 \end{bmatrix} - \begin{bmatrix} \{\boldsymbol{x}_{3,j}^t\} \\ \boldsymbol{z}_1^t \\ \boldsymbol{z}_2^{t'} \\ \boldsymbol{z}_3^t \end{bmatrix} \right) \right\|^2.
\end{aligned}
\tag{64}
$$

According to $\mathbb{E}[G_\mu^{\text{in}}(\{\boldsymbol{x}_{3,j}^t\}, \boldsymbol{z}_1^t, \boldsymbol{z}_2^{t'}, \boldsymbol{z}_3^t)] = \phi_{\mu,\text{in}}(\{\boldsymbol{x}_{3,j}^t\}, \boldsymbol{z}_1^t, \boldsymbol{z}_2^{t'}, \boldsymbol{z}_3^t)$, taking expectation on both sides of Eq. (64), we have that,

$$
\begin{aligned}
&\mathbb{E}[\phi_{\text{in}}(\{\boldsymbol{x}_{3,j}\}, \boldsymbol{z}_1, \boldsymbol{z}_2{'}, \boldsymbol{z}_3)] \\
&\geq \mathbb{E}[\phi_{\text{in}}(\{\boldsymbol{x}_{3,j}^t\}, \boldsymbol{z}_1^t, \boldsymbol{z}_2^{t'}, \boldsymbol{z}_3^t)] + \phi_{\mu,\text{in}}(\{\boldsymbol{x}_{3,j}^t\}, \boldsymbol{z}_1^t, \boldsymbol{z}_2^{t'}, \boldsymbol{z}_3^t)^\top \left( \begin{bmatrix} \{\boldsymbol{x}_{3,j}\} \\ \boldsymbol{z}_1 \\ \boldsymbol{z}_2{'} \\ \boldsymbol{z}_3 \end{bmatrix} - \begin{bmatrix} \{\boldsymbol{x}_{3,j}^t\} \\ \boldsymbol{z}_1^t \\ \boldsymbol{z}_2^{t'} \\ \boldsymbol{z}_3^t \end{bmatrix} \right) \\
&\quad + \left( \frac{\partial \phi_{\text{in}}(\{\boldsymbol{x}_{3,j}^t\}, \boldsymbol{z}_1^t, \boldsymbol{z}_2^{t'}, \boldsymbol{z}_3^t)}{\partial(\{\boldsymbol{x}_{3,j}\}, \boldsymbol{z}_1, \boldsymbol{z}_2{'}, \boldsymbol{z}_3)} - \phi_{\mu,\text{in}}(\{\boldsymbol{x}_{3,j}^t\}, \boldsymbol{z}_1^t, \boldsymbol{z}_2^{t'}, \boldsymbol{z}_3^t) \right)^\top \left( \begin{bmatrix} \{\boldsymbol{x}_{3,j}\} \\ \boldsymbol{z}_1 \\ \boldsymbol{z}_2{'} \\ \boldsymbol{z}_3 \end{bmatrix} - \begin{bmatrix} \{\boldsymbol{x}_{3,j}^t\} \\ \boldsymbol{z}_1^t \\ \boldsymbol{z}_2^{t'} \\ \boldsymbol{z}_3^t \end{bmatrix} \right) \\
&\quad - \frac{L}{2} \left\| \left( \begin{bmatrix} \{\boldsymbol{x}_{3,j}\} \\ \boldsymbol{z}_1 \\ \boldsymbol{z}_2{'} \\ \boldsymbol{z}_3 \end{bmatrix} - \begin{bmatrix} \{\boldsymbol{x}_{3,j}^t\} \\ \boldsymbol{z}_1^t \\ \boldsymbol{z}_2^{t'} \\ \boldsymbol{z}_3^t \end{bmatrix} \right) \right\|^2.
\end{aligned}
\tag{65}
$$

Combining with the Cauchy-Schwarz inequality, we have that,

$$\mathbb{E}[\phi_{\text{in}}(\{\boldsymbol{x}_{3,j}\}, \boldsymbol{z}_1, \boldsymbol{z}_2{}', \boldsymbol{z}_3)]$$

$$\geq \mathbb{E}[\phi_{\text{in}}(\{\boldsymbol{x}_{3,j}^t\}, \boldsymbol{z}_1^t, \boldsymbol{z}_2^{t}{}', \boldsymbol{z}_3^t)] + \phi_{\mu,\text{in}}(\{\boldsymbol{x}_{3,j}^t\}, \boldsymbol{z}_1^t, \boldsymbol{z}_2^{t}{}', \boldsymbol{z}_3^t)^\top \left( \begin{bmatrix} \{\boldsymbol{x}_{3,j}\} \\ \boldsymbol{z}_1 \\ \boldsymbol{z}_2{}' \\ \boldsymbol{z}_3 \end{bmatrix} - \begin{bmatrix} \{\boldsymbol{x}_{3,j}^t\} \\ \boldsymbol{z}_1^t \\ \boldsymbol{z}_2^{t}{}' \\ \boldsymbol{z}_3^t \end{bmatrix} \right)$$

$$- \frac{1}{2} \left\| \frac{\partial \phi_{\text{in}}(\{\boldsymbol{x}_{3,j}^t\}, \boldsymbol{z}_1^t, \boldsymbol{z}_2^{t}{}', \boldsymbol{z}_3^t)}{\partial(\{\boldsymbol{x}_{3,j}\}, \boldsymbol{z}_1, \boldsymbol{z}_2{}', \boldsymbol{z}_3)} - \phi_{\mu,\text{in}}(\{\boldsymbol{x}_{3,j}^t\}, \boldsymbol{z}_1^t, \boldsymbol{z}_2^{t}{}', \boldsymbol{z}_3^t) \right\|^2 - \frac{L+1}{2} \left\| \left( \begin{bmatrix} \{\boldsymbol{x}_{3,j}\} \\ \boldsymbol{z}_1 \\ \boldsymbol{z}_2{}' \\ \boldsymbol{z}_3 \end{bmatrix} - \begin{bmatrix} \{\boldsymbol{x}_{3,j}^t\} \\ \boldsymbol{z}_1^t \\ \boldsymbol{z}_2^{t}{}' \\ \boldsymbol{z}_3^t \end{bmatrix} \right) \right\|^2. \tag{66}$$

And according to Eq. (3.6) in Ghadimi and Lan (2013), we can obtain that,

$$\left\| \phi_{\mu,\text{in}}(\{\boldsymbol{x}_{3,j}^t\}, \boldsymbol{z}_1^t, \boldsymbol{z}_2^{t}{}', \boldsymbol{z}_3^t) - \frac{\partial \phi_{\text{in}}(\{\boldsymbol{x}_{3,j}^t\}, \boldsymbol{z}_1^t, \boldsymbol{z}_2^{t}{}', \boldsymbol{z}_3^t)}{\partial(\{\boldsymbol{x}_{3,j}\}, \boldsymbol{z}_1, \boldsymbol{z}_2, \boldsymbol{z}_3)} \right\|^2 \leq \frac{\mu^2}{4} L^2 (d_1 + d_2 + (N+1)d_3 + 3)^3. \tag{67}$$

By combining Eq. (66) with Eq. (67), we have that,

$$\mathbb{E}[\phi_{\text{in}}(\{\boldsymbol{x}_{3,j}\}, \boldsymbol{z}_1, \boldsymbol{z}_2{}', \boldsymbol{z}_3)]$$

$$\geq \mathbb{E}[\phi_{\text{in}}(\{\boldsymbol{x}_{3,j}^t\}, \boldsymbol{z}_1^t, \boldsymbol{z}_2^{t}{}', \boldsymbol{z}_3^t)] + \phi_{\mu,\text{in}}(\{\boldsymbol{x}_{3,j}^t\}, \boldsymbol{z}_1^t, \boldsymbol{z}_2^{t}{}', \boldsymbol{z}_3^t)^\top \left( \begin{bmatrix} \{\boldsymbol{x}_{3,j}\} \\ \boldsymbol{z}_1 \\ \boldsymbol{z}_2{}' \\ \boldsymbol{z}_3 \end{bmatrix} - \begin{bmatrix} \{\boldsymbol{x}_{3,j}^t\} \\ \boldsymbol{z}_1^t \\ \boldsymbol{z}_2^{t}{}' \\ \boldsymbol{z}_3^t \end{bmatrix} \right)$$

$$- \frac{\mu^2}{8} L^2 (d_1 + d_2 + (N+1)d_3 + 3)^3 - \frac{L+1}{2} \left\| \left( \begin{bmatrix} \{\boldsymbol{x}_{3,j}\} \\ \boldsymbol{z}_1 \\ \boldsymbol{z}_2{}' \\ \boldsymbol{z}_3 \end{bmatrix} - \begin{bmatrix} \{\boldsymbol{x}_{3,j}^t\} \\ \boldsymbol{z}_1^t \\ \boldsymbol{z}_2^{t}{}' \\ \boldsymbol{z}_3^t \end{bmatrix} \right) \right\|^2. \tag{68}$$

For any point belongs to the original feasible region, i.e., $\phi_{\text{in}}(\{\boldsymbol{x}_{3,j}\}, \boldsymbol{z}_1, \boldsymbol{z}_2{}', \boldsymbol{z}_3) = 0$, according to $\varepsilon_{\text{in}} \geq 0$, we can obtain that it also satisfies that,

$$\mathbb{E}[\phi_{\text{in}}(\{\boldsymbol{x}_{3,j}^t\}, \boldsymbol{z}_1^t, \boldsymbol{z}_2^{t}{}', \boldsymbol{z}_3^t)] + \mathbb{E}[G_\mu^{\text{in}}(\{\boldsymbol{x}_{3,j}^t\}, \boldsymbol{z}_1^t, \boldsymbol{z}_2^{t}{}', \boldsymbol{z}_3^t)]^\top \left( \begin{bmatrix} \{\boldsymbol{x}_{3,j}\} \\ \boldsymbol{z}_1 \\ \boldsymbol{z}_2{}' \\ \boldsymbol{z}_3 \end{bmatrix} - \begin{bmatrix} \{\boldsymbol{x}_{3,j}^t\} \\ \boldsymbol{z}_1^t \\ \boldsymbol{z}_2^{t}{}' \\ \boldsymbol{z}_3^t \end{bmatrix} \right) ]$$

$$\leq \frac{L+1}{2} \left\| \left( \begin{bmatrix} \{\boldsymbol{x}_{3,j}\} \\ \boldsymbol{z}_1 \\ \boldsymbol{z}_2{}' \\ \boldsymbol{z}_3 \end{bmatrix} - \begin{bmatrix} \{\boldsymbol{x}_{3,j}^t\} \\ \boldsymbol{z}_1^t \\ \boldsymbol{z}_2^{t}{}' \\ \boldsymbol{z}_3^t \end{bmatrix} \right) \right\|^2 + \frac{\mu^2}{8} L^2 (d_1 + d_2 + (N+1)d_3 + 3)^3 + \varepsilon_{\text{in}}. \tag{69}$$

According to Eq. (9), we can conclude that for any point belongs to the original feasible region of constraint $\phi_{\text{in}}(\{\boldsymbol{x}_{3,j}\}, \boldsymbol{z}_1, \boldsymbol{z}_2{}', \boldsymbol{z}_3) = 0$, it also belongs to the $P_{\text{in}}^{t+1}$, that is, the original feasible region is a subset of the feasible region formed by inner layer zeroth order cuts. Let $S_{\text{in}}$ denote the original feasible region of constraint $\phi_{\text{in}}(\{\boldsymbol{x}_{3,j}\}, \boldsymbol{z}_1, \boldsymbol{z}_2{}', \boldsymbol{z}_3) = 0$, we can obtain that the feasible region formed by inner layer zeroth order cuts will be gradually tightened with zeroth order cuts added according to Eq. (69), that is,

$$S_{\text{in}} \subseteq P_{\text{in}}^{t+1} \subseteq P_{\text{in}}^t \subseteq \cdots \subseteq P_{\text{in}}^0. \tag{70}$$

## C.2 PROOF OF PROPOSITION 2

For any point $(\{\boldsymbol{x}_{2,j}\}, \{\boldsymbol{x}_{3,j}\}, \boldsymbol{z}_1, \boldsymbol{z}_2, \boldsymbol{z}_3)$ in the original feasible region, i.e., $\phi_{\text{out}}(\{\boldsymbol{x}_{2,j}\}, \{\boldsymbol{x}_{3,j}\}, \boldsymbol{z}_1, \boldsymbol{z}_2, \boldsymbol{z}_3) = 0$, according to the properties of $L$-smoothness, we have

that,

$$\phi_{\text{out}}(\{\boldsymbol{x}_{2,j}\}, \{\boldsymbol{x}_{3,j}\}, \boldsymbol{z}_1, \boldsymbol{z}_2, \boldsymbol{z}_3)$$

$$\geq \phi_{\text{out}}(\{\boldsymbol{x}_{2,j}^t\}, \{\boldsymbol{x}_{3,j}^t\}, \boldsymbol{z}_1^t, \boldsymbol{z}_2^t, \boldsymbol{z}_3^t) + \frac{\partial \phi_{\text{out}}(\{\boldsymbol{x}_{2,j}^t\}, \{\boldsymbol{x}_{3,j}^t\}, \boldsymbol{z}_1^t, \boldsymbol{z}_2^t, \boldsymbol{z}_3^t)}{\partial(\{\boldsymbol{x}_{2,j}\}, \{\boldsymbol{x}_{3,j}\}, \boldsymbol{z}_1, \boldsymbol{z}_2, \boldsymbol{z}_3)}^{\top} \left( \begin{bmatrix} \{\boldsymbol{x}_{2,j}\} \\ \{\boldsymbol{x}_{3,j}\} \\ \boldsymbol{z}_1 \\ \boldsymbol{z}_2 \\ \boldsymbol{z}_3 \end{bmatrix} - \begin{bmatrix} \{\boldsymbol{x}_{2,j}^t\} \\ \{\boldsymbol{x}_{3,j}^t\} \\ \boldsymbol{z}_1^t \\ \boldsymbol{z}_2^t \\ \boldsymbol{z}_3^t \end{bmatrix} \right)$$

$$- \frac{L}{2} \left\| \left( \begin{bmatrix} \{\boldsymbol{x}_{2,j}\} \\ \{\boldsymbol{x}_{3,j}\} \\ \boldsymbol{z}_1 \\ \boldsymbol{z}_2 \\ \boldsymbol{z}_3 \end{bmatrix} - \begin{bmatrix} \{\boldsymbol{x}_{2,j}^t\} \\ \{\boldsymbol{x}_{3,j}^t\} \\ \boldsymbol{z}_1^t \\ \boldsymbol{z}_2^t \\ \boldsymbol{z}_3^t \end{bmatrix} \right) \right\|^2$$

$$= \phi_{\text{out}}(\{\boldsymbol{x}_{2,j}^t\}, \{\boldsymbol{x}_{3,j}^t\}, \boldsymbol{z}_1^t, \boldsymbol{z}_2^t, \boldsymbol{z}_3^t) + G_{\mu}^{\text{out}}(t)^{\top} \left( \begin{bmatrix} \{\boldsymbol{x}_{2,j}\} \\ \{\boldsymbol{x}_{3,j}\} \\ \boldsymbol{z}_1 \\ \boldsymbol{z}_2 \\ \boldsymbol{z}_3 \end{bmatrix} - \begin{bmatrix} \{\boldsymbol{x}_{2,j}^t\} \\ \{\boldsymbol{x}_{3,j}^t\} \\ \boldsymbol{z}_1^t \\ \boldsymbol{z}_2^t \\ \boldsymbol{z}_3^t \end{bmatrix} \right)$$

$$+ \left( \frac{\partial \phi_{\text{out}}(\{\boldsymbol{x}_{2,j}^t\}, \{\boldsymbol{x}_{3,j}^t\}, \boldsymbol{z}_1^t, \boldsymbol{z}_2^t, \boldsymbol{z}_3^t)}{\partial(\{\boldsymbol{x}_{2,j}\}, \{\boldsymbol{x}_{3,j}\}, \boldsymbol{z}_1, \boldsymbol{z}_2, \boldsymbol{z}_3)} - G_{\mu}^{\text{out}}(t) \right)^{\top} \left( \begin{bmatrix} \{\boldsymbol{x}_{2,j}\} \\ \{\boldsymbol{x}_{3,j}\} \\ \boldsymbol{z}_1 \\ \boldsymbol{z}_2 \\ \boldsymbol{z}_3 \end{bmatrix} - \begin{bmatrix} \{\boldsymbol{x}_{2,j}^t\} \\ \{\boldsymbol{x}_{3,j}^t\} \\ \boldsymbol{z}_1^t \\ \boldsymbol{z}_2^t \\ \boldsymbol{z}_3^t \end{bmatrix} \right)$$

$$- \frac{L}{2} \left\| \left( \begin{bmatrix} \{\boldsymbol{x}_{2,j}\} \\ \{\boldsymbol{x}_{3,j}\} \\ \boldsymbol{z}_1 \\ \boldsymbol{z}_2 \\ \boldsymbol{z}_3 \end{bmatrix} - \begin{bmatrix} \{\boldsymbol{x}_{2,j}^t\} \\ \{\boldsymbol{x}_{3,j}^t\} \\ \boldsymbol{z}_1^t \\ \boldsymbol{z}_2^t \\ \boldsymbol{z}_3^t \end{bmatrix} \right) \right\|^2,$$

$$(71)$$

where $G_{\mu}^{\text{out}}(t)$ is the simplified form of $G_{\mu}^{\text{out}}(\{\boldsymbol{x}_{2,j}^t\}, \{\boldsymbol{x}_{3,j}^t\}, \boldsymbol{z}_1^t, \boldsymbol{z}_2^t, \boldsymbol{z}_3^t)$. According to $\mathbb{E}[G_{\mu}^{\text{out}}(t)] = \phi_{\mu,\text{out}}(\{\boldsymbol{x}_{2,j}^t\}, \{\boldsymbol{x}_{3,j}^t\}, \boldsymbol{z}_1^t, \boldsymbol{z}_2^t, \boldsymbol{z}_3^t)$, taking expectation on both sides of Eq. (71), we have that,

$$\mathbb{E}[\phi_{\text{out}}(\{\boldsymbol{x}_{2,j}\}, \{\boldsymbol{x}_{3,j}\}, \boldsymbol{z}_1, \boldsymbol{z}_2, \boldsymbol{z}_3)]$$

$$\geq \mathbb{E}[\phi_{\text{out}}(\{\boldsymbol{x}_{2,j}^t\}, \{\boldsymbol{x}_{3,j}^t\}, \boldsymbol{z}_1^t, \boldsymbol{z}_2^t, \boldsymbol{z}_3^t)] + \phi_{\mu,\text{out}}(t)^{\top} \left( \begin{bmatrix} \{\boldsymbol{x}_{2,j}\} \\ \{\boldsymbol{x}_{3,j}\} \\ \boldsymbol{z}_1 \\ \boldsymbol{z}_2 \\ \boldsymbol{z}_3 \end{bmatrix} - \begin{bmatrix} \{\boldsymbol{x}_{2,j}^t\} \\ \{\boldsymbol{x}_{3,j}^t\} \\ \boldsymbol{z}_1^t \\ \boldsymbol{z}_2^t \\ \boldsymbol{z}_3^t \end{bmatrix} \right)$$

$$+ \left( \frac{\partial \phi_{\text{out}}(\{\boldsymbol{x}_{2,j}^t\}, \{\boldsymbol{x}_{3,j}^t\}, \boldsymbol{z}_1^t, \boldsymbol{z}_2^t, \boldsymbol{z}_3^t)}{\partial(\{\boldsymbol{x}_{2,j}\}, \{\boldsymbol{x}_{3,j}\}, \boldsymbol{z}_1, \boldsymbol{z}_2, \boldsymbol{z}_3)} - \phi_{\mu,\text{out}}(t) \right)^{\top} \left( \begin{bmatrix} \{\boldsymbol{x}_{2,j}\} \\ \{\boldsymbol{x}_{3,j}\} \\ \boldsymbol{z}_1 \\ \boldsymbol{z}_2 \\ \boldsymbol{z}_3 \end{bmatrix} - \begin{bmatrix} \{\boldsymbol{x}_{2,j}^t\} \\ \{\boldsymbol{x}_{3,j}^t\} \\ \boldsymbol{z}_1^t \\ \boldsymbol{z}_2^t \\ \boldsymbol{z}_3^t \end{bmatrix} \right)$$

$$(72)$$

$$- \frac{L}{2} \left\| \left( \begin{bmatrix} \{\boldsymbol{x}_{2,j}\} \\ \{\boldsymbol{x}_{3,j}\} \\ \boldsymbol{z}_1 \\ \boldsymbol{z}_2 \\ \boldsymbol{z}_3 \end{bmatrix} - \begin{bmatrix} \{\boldsymbol{x}_{2,j}^t\} \\ \{\boldsymbol{x}_{3,j}^t\} \\ \boldsymbol{z}_1^t \\ \boldsymbol{z}_2^t \\ \boldsymbol{z}_3^t \end{bmatrix} \right) \right\|^2,$$

where $\phi_{\mu,\text{out}}(t)$ is the simplified form of $\phi_{\mu,\text{out}}(\{\boldsymbol{x}_{2,j}^t\}, \{\boldsymbol{x}_{3,j}^t\}, \boldsymbol{z}_1^t, \boldsymbol{z}_2^t, \boldsymbol{z}_3^t)$. Combining with the Cauchy-Schwarz inequality, we have that,

$$
\begin{aligned}
&\mathbb{E}[\phi_{\text{out}}(\{\boldsymbol{x}_{2,j}\}, \{\boldsymbol{x}_{3,j}\}, \boldsymbol{z}_1, \boldsymbol{z}_2, \boldsymbol{z}_3)] \\
&\geq \mathbb{E}[\phi_{\text{out}}(\{\boldsymbol{x}_{2,j}^t\}, \{\boldsymbol{x}_{3,j}^t\}, \boldsymbol{z}_1^t, \boldsymbol{z}_2^t, \boldsymbol{z}_3^t)] + \phi_{\mu,\text{out}}(t)^\top \left( \begin{bmatrix} \{\boldsymbol{x}_{2,j}\} \\ \{\boldsymbol{x}_{3,j}\} \\ \boldsymbol{z}_1 \\ \boldsymbol{z}_2 \\ \boldsymbol{z}_3 \end{bmatrix} - \begin{bmatrix} \{\boldsymbol{x}_{2,j}^t\} \\ \{\boldsymbol{x}_{3,j}^t\} \\ \boldsymbol{z}_1^t \\ \boldsymbol{z}_2^t \\ \boldsymbol{z}_3^t \end{bmatrix} \right) \\
&\quad - \frac{1}{2} \| \frac{\partial \phi_{\text{out}}(\{\boldsymbol{x}_{2,j}^t\}, \{\boldsymbol{x}_{3,j}^t\}, \boldsymbol{z}_1^t, \boldsymbol{z}_2^t, \boldsymbol{z}_3^t)}{\partial(\{\boldsymbol{x}_{2,j}\}, \{\boldsymbol{x}_{3,j}\}, \boldsymbol{z}_1, \boldsymbol{z}_2, \boldsymbol{z}_3)} - \phi_{\mu,\text{out}}(t) \|^2 \\
&\quad - \frac{L+1}{2} \| \left( \begin{bmatrix} \{\boldsymbol{x}_{2,j}\} \\ \{\boldsymbol{x}_{3,j}\} \\ \boldsymbol{z}_1 \\ \boldsymbol{z}_2 \\ \boldsymbol{z}_3 \end{bmatrix} - \begin{bmatrix} \{\boldsymbol{x}_{2,j}^t\} \\ \{\boldsymbol{x}_{3,j}^t\} \\ \boldsymbol{z}_1^t \\ \boldsymbol{z}_2^t \\ \boldsymbol{z}_3^t \end{bmatrix} \right) \|^2.
\end{aligned}
\tag{73}
$$

And according to Eq. (3.6) in Ghadimi and Lan (2013), we can obtain that,

$$
\| \phi_{\mu,\text{out}}(t) - \frac{\partial \phi_{\text{out}}(\{\boldsymbol{x}_{2,j}^t\}, \{\boldsymbol{x}_{3,j}^t\}, \boldsymbol{z}_1^t, \boldsymbol{z}_2^t, \boldsymbol{z}_3^t)}{\partial(\{\boldsymbol{x}_{2,j}\}, \{\boldsymbol{x}_{3,j}\}, \boldsymbol{z}_1, \boldsymbol{z}_2, \boldsymbol{z}_3)} \|^2 \leq \frac{\mu^2}{4} L^2 (d_1 + (N+1)(d_2 + d_3) + 3)^3. \tag{74}
$$

By combining Eq. (73) with Eq. (74), we have that,

$$
\begin{aligned}
&\mathbb{E}[\phi_{\text{out}}(\{\boldsymbol{x}_{2,j}\}, \{\boldsymbol{x}_{3,j}\}, \boldsymbol{z}_1, \boldsymbol{z}_2, \boldsymbol{z}_3)] \\
&\geq \mathbb{E}[\phi_{\text{out}}(\{\boldsymbol{x}_{2,j}^t\}, \{\boldsymbol{x}_{3,j}^t\}, \boldsymbol{z}_1^t, \boldsymbol{z}_2^t, \boldsymbol{z}_3^t)] + \phi_{\mu,\text{out}}(t)^\top \left( \begin{bmatrix} \{\boldsymbol{x}_{2,j}\} \\ \{\boldsymbol{x}_{3,j}\} \\ \boldsymbol{z}_1 \\ \boldsymbol{z}_2 \\ \boldsymbol{z}_3 \end{bmatrix} - \begin{bmatrix} \{\boldsymbol{x}_{2,j}^t\} \\ \{\boldsymbol{x}_{3,j}^t\} \\ \boldsymbol{z}_1^t \\ \boldsymbol{z}_2^t \\ \boldsymbol{z}_3^t \end{bmatrix} \right) \\
&\quad - \frac{\mu^2}{8} L^2 (d_1 + (N+1)(d_2 + d_3) + 3)^3 - \frac{L+1}{2} \| \left( \begin{bmatrix} \{\boldsymbol{x}_{2,j}\} \\ \{\boldsymbol{x}_{3,j}\} \\ \boldsymbol{z}_1 \\ \boldsymbol{z}_2 \\ \boldsymbol{z}_3 \end{bmatrix} - \begin{bmatrix} \{\boldsymbol{x}_{2,j}^t\} \\ \{\boldsymbol{x}_{3,j}^t\} \\ \boldsymbol{z}_1^t \\ \boldsymbol{z}_2^t \\ \boldsymbol{z}_3^t \end{bmatrix} \right) \|^2.
\end{aligned}
\tag{75}
$$

For any point belongs to the original feasible region, i.e., $\phi_{\text{out}}(\{\boldsymbol{x}_{2,j}\}, \{\boldsymbol{x}_{3,j}\}, \boldsymbol{z}_1, \boldsymbol{z}_2, \boldsymbol{z}_3) = 0$, according to $\varepsilon_{\text{in}} \geq 0$, we can obtain that it also satisfies that,

$$
\begin{aligned}
&\phi_{\text{out}}(\{\boldsymbol{x}_{2,j}^t\}, \{\boldsymbol{x}_{3,j}^t\}, \boldsymbol{z}_1^t, \boldsymbol{z}_2^t, \boldsymbol{z}_3^t) + G_\mu^{\text{out}}(\{\boldsymbol{x}_{2,j}^t\}, \{\boldsymbol{x}_{3,j}^t\}, \boldsymbol{z}_1^t, \boldsymbol{z}_2^t, \boldsymbol{z}_3^t)^\top \left( \begin{bmatrix} \{\boldsymbol{x}_{2,j}\} \\ \{\boldsymbol{x}_{3,j}\} \\ \boldsymbol{z}_1 \\ \boldsymbol{z}_2 \\ \boldsymbol{z}_3 \end{bmatrix} - \begin{bmatrix} \{\boldsymbol{x}_{2,j}^t\} \\ \{\boldsymbol{x}_{3,j}^t\} \\ \boldsymbol{z}_1^t \\ \boldsymbol{z}_2^t \\ \boldsymbol{z}_3^t \end{bmatrix} \right) \\
&\leq \frac{L+1}{2} \left( \sum_{i=2}^3 \sum_j \|\boldsymbol{x}_{i,j} - \boldsymbol{x}_{i,j}^t\|^2 + \sum_i \|\boldsymbol{z}_i - \boldsymbol{z}_i^t\|^2 \right) + \frac{\mu^2}{8} L^2 (d_1 + (N+1)(d_2 + d_3) + 3)^3 + \varepsilon_{\text{out}}.
\end{aligned}
\tag{76}
$$

According to Eq. (11), we can conclude that for any point belongs to the original feasible region of constraint $\phi_{\text{out}}(\{\boldsymbol{x}_{2,j}\}, \{\boldsymbol{x}_{3,j}\}, \boldsymbol{z}_1, \boldsymbol{z}_2, \boldsymbol{z}_3) = 0$, it also belongs to the $P_{\text{out}}^{t+1}$, that is, the original feasible region is a subset of the feasible region formed by outer layer zeroth order cuts. In addition, let $S_{\text{out}}$ denote the original feasible region of constraint $\phi_{\text{out}}(\{\boldsymbol{x}_{2,j}\}, \{\boldsymbol{x}_{3,j}\}, \boldsymbol{z}_1, \boldsymbol{z}_2, \boldsymbol{z}_3) = 0$, based on Eq. (76), we can obtain that the feasible region formed by outer layer zeroth order cuts will be gradually tightened with zeroth order cuts added, that is,

$$
S_{\text{out}} \subseteq P_{\text{out}}^{t+1} \subseteq P_{\text{out}}^t \subseteq \cdots \subseteq P_{\text{out}}^0. \tag{77}
$$

## D  THEORETICAL ANALYSES ABOUT THE CASCADED POLYNOMIAL APPROXIMATION PROBLEM

In this section, we theoretically analyze the connections between the original distributed trilevel zeroth order optimization problem in Eq. (2) and the cascaded polynomial approximation problem in Eq. (8). To facilitate this discussion, we start by examining the distributed bilevel zeroth order optimization problem, which can be expressed as follows,

$$
\begin{aligned}
\min \ & \sum_{j=1}^{N} f_{1,j}(\boldsymbol{x}_1, \boldsymbol{x}_2) \\
\text{s.t. } & \boldsymbol{x}_2 = \arg\min_{\boldsymbol{x}_2{'}} \sum_{j=1}^{N} f_{2,j}(\boldsymbol{x}_1, \boldsymbol{x}_2{'}) \\
\text{var. } & \quad \boldsymbol{x}_1, \boldsymbol{x}_2.
\end{aligned}
\tag{78}
$$

The optimization problem in Eq. (78) can be equivalently reformulated as,

$$
\begin{aligned}
\min \ & \sum_{j=1}^{N} f_{1,j}(\boldsymbol{x}_{1,j}, \boldsymbol{x}_{2,j}) \\
\text{s.t. } & \boldsymbol{x}_{1,j} = \boldsymbol{z}_1, \forall j = 1, \cdots, N \\
& \{\boldsymbol{x}_{2,j}\}, \boldsymbol{z}_2 = \arg\min_{\{\boldsymbol{x}_{2,j}{'}\}, \boldsymbol{z}_2{'}} \sum_{j=1}^{N} f_{2,j}(\boldsymbol{z}_1, \boldsymbol{x}_{2,j}{'}) \\
& \qquad\qquad\qquad \text{s.t. } \boldsymbol{x}_{2,j}{'} = \boldsymbol{z}_2{'}, \forall j = 1, \cdots, N \\
\text{var. } & \quad \{\boldsymbol{x}_{1,j}\}, \{\boldsymbol{x}_{2,j}\}, \boldsymbol{z}_1, \boldsymbol{z}_2.
\end{aligned}
\tag{79}
$$

By utilizing the proposed polynomial approximation with zeroth order cut, we can obtain the following zeroth order polynomial approximation problem,

$$
\begin{aligned}
\min \ & \sum_{j=1}^{N} f_{1,j}(\boldsymbol{x}_{1,j}, \boldsymbol{x}_{2,j}) \\
\text{s.t. } & \boldsymbol{x}_{1,j} = \boldsymbol{z}_1, \forall j = 1, \cdots, N \\
& \sum_{j=1}^{N} \boldsymbol{a}_{2,j,l}^{\top} \boldsymbol{x}_{2,j}^2 + \boldsymbol{b}_{2,j,l}^{\top} \boldsymbol{x}_{2,j} + \sum_{i=1}^{2} \boldsymbol{c}_{i,l}^{\top} \boldsymbol{z}_i^2 + \boldsymbol{d}_{i,l}^{\top} \boldsymbol{z}_i + e_l \le \varepsilon, \forall l \\
\text{var. } & \quad \{\boldsymbol{x}_{1,j}\}, \{\boldsymbol{x}_{2,j}\}, \boldsymbol{z}_1, \boldsymbol{z}_2.
\end{aligned}
\tag{80}
$$

According to Proposition 1 and 2, we can obtain the feasible region of the problem in Eq. (79) is a subset of the feasible region of the problem in Eq. (80). Thus, we can conclude that the zeroth order polynomial approximation optimization problem in Eq. (80) is the relaxed problem of the distributed bilevel zeroth order optimization problem in Eq. (78).

For the distributed trilevel zeroth order optimization problem, we first define the following feasible regions.

$$
S_1 = \left\{ \{\boldsymbol{x}_{i,j}\}, \{\boldsymbol{z}_i\} \Big| \begin{array}{l} h_l^{\text{out}}(\{\boldsymbol{x}_{2,j}\}, \{\boldsymbol{x}_{3,j}\}, \boldsymbol{z}_1, \boldsymbol{z}_2, \boldsymbol{z}_3) \le \varepsilon_{\text{out}}, \forall l, \\ \boldsymbol{z}_1 = \boldsymbol{x}_{1,j}, \forall j \end{array} \right\},
\tag{81}
$$

$$
S_2 = 
\left\{ \{\boldsymbol{x}_{i,j}\}, \{\boldsymbol{z}_i\} \Big| \ \left\| \begin{bmatrix} \{\boldsymbol{x}_{2,j}\} \\ \boldsymbol{z}_2 \end{bmatrix} - \begin{array}{l} \arg\min_{\{\boldsymbol{x}_{2,j}{'}\}, \boldsymbol{z}_2{'}} \sum_{j=1}^{N} f_{2,j}(\boldsymbol{z}_1, \boldsymbol{x}_{2,j}{'}, \boldsymbol{x}_{3,j}) \\ \text{s.t. } \quad \boldsymbol{x}_{2,j}{'} = \boldsymbol{z}_2{'}, \forall j, \\ \qquad h_l^{\text{in}}(\{\boldsymbol{x}_{3,j}\}, \boldsymbol{z}_1, \boldsymbol{z}_2{'}, \boldsymbol{z}_3) \le \varepsilon_{\text{in}}, \forall l \end{array} \right\|^2 \le \varepsilon_{\text{out}}, \atop \boldsymbol{z}_1 = \boldsymbol{x}_{1,j}, \forall j \right\},
\tag{82}
$$

$$S_3 =$$

$$
\left\{ \{\boldsymbol{x}_{i,j}\},\{\boldsymbol{z}_i\} \Big| \; \left\| \begin{bmatrix} \{\boldsymbol{x}_{2,j}\} \\ \boldsymbol{z}_2 \end{bmatrix} - \begin{array}{c} \arg\min\limits_{\{\boldsymbol{x}_{2,j}'\},\boldsymbol{z}_2'} \sum\limits_{j=1}^{N} f_{2,j}(\boldsymbol{z}_1,\boldsymbol{x}_{2,j},\boldsymbol{x}_{3,j}) \\ \text{s.t. } \boldsymbol{x}_{2,j}' = \boldsymbol{z}_2', \forall j = 1,\cdots,N \\ \{\boldsymbol{x}_{3,j}\},\boldsymbol{z}_3 = \arg\min\limits_{\{\boldsymbol{x}_{3,j}'\},\boldsymbol{z}_3'} \sum\limits_{j=1}^{N} f_{3,j}(\boldsymbol{z}_1,\boldsymbol{z}_2',\boldsymbol{x}_{3,j}') \\ \text{s.t. } \boldsymbol{x}_{3,j}' = \boldsymbol{z}_3', \forall j = 1,\cdots,N \end{array} \right\|^2 = 0, \\ \boldsymbol{z}_1 = \boldsymbol{x}_{1,j}, \forall j \right\}.
$$

$$(83)$$

It is seen from Eq. (81) and Eq. (83) that $S_1$ and $S_3$ respectively represent the feasible region of optimization problems in Eq. (8) and Eq. (3). For any feasible solution $\{\hat{\boldsymbol{x}}_{i,j}\},\{\hat{\boldsymbol{z}}_i\}$ of optimization problem in Eq. (3), it satisfies that,

$$
\left\| \begin{bmatrix} \{\hat{\boldsymbol{x}}_{2,j}\} \\ \hat{\boldsymbol{z}}_2 \end{bmatrix} - \begin{array}{c} \arg\min\limits_{\{\boldsymbol{x}_{2,j}'\},\boldsymbol{z}_2'} \sum\limits_{j=1}^{N} f_{2,j}(\hat{\boldsymbol{z}}_1,\boldsymbol{x}_{2,j}',\hat{\boldsymbol{x}}_{3,j}) \\ \text{s.t. } \boldsymbol{x}_{2,j}' = \boldsymbol{z}_2', \forall j = 1,\cdots,N \\ \{\hat{\boldsymbol{x}}_{3,j}\},\hat{\boldsymbol{z}}_3 = \arg\min\limits_{\{\boldsymbol{x}_{3,j}'\},\boldsymbol{z}_3'} \sum\limits_{j=1}^{N} f_{3,j}(\hat{\boldsymbol{z}}_1,\boldsymbol{z}_2',\boldsymbol{x}_{3,j}') \\ \text{s.t. } \boldsymbol{x}_{3,j}' = \boldsymbol{z}_3', \forall j = 1,\cdots,N \end{array} \right\|^2 = 0. \qquad (84)
$$

Based on Proposition 1, we have that the feasible region of constraint $\phi_{\text{in}}(\{\boldsymbol{x}_{3,j}\},\boldsymbol{z}_1,\boldsymbol{z}_2',\boldsymbol{z}_3) = 0$ is a subset of the feasible region formed by inner layer zeroth order cuts, i.e., $\{\{\boldsymbol{x}_{3,j}\},\boldsymbol{z}_1,\boldsymbol{z}_2',\boldsymbol{z}_3 \,|\, h_l^{\text{in}}(\{\boldsymbol{x}_{3,j}\},\boldsymbol{z}_1,\boldsymbol{z}_2',\boldsymbol{z}_3) \leq \varepsilon_{\text{in}}, \forall l\}$. Moreover, the feasible region formed by inner layer zeroth order cuts will be continuously tightened with zeroth order cuts added. Thus, let $\beta \geq 0$ satisfy that,

$$
\left\| \begin{array}{c} \arg\min\limits_{\{\boldsymbol{x}_{2,j}'\},\boldsymbol{z}_2'} \sum\limits_{j=1}^{N} f_{2,j}(\hat{\boldsymbol{z}}_1,\boldsymbol{x}_{2,j}',\hat{\boldsymbol{x}}_{3,j}) \\ \text{s.t. } \quad \boldsymbol{x}_{2,j}' = \boldsymbol{z}_2', \forall j, \\ h_l^{\text{in}}(\{\hat{\boldsymbol{x}}_{3,j}\},\hat{\boldsymbol{z}}_1,\boldsymbol{z}_2',\hat{\boldsymbol{z}}_3) \leq \varepsilon_{\text{in}}, \forall l \end{array} - \begin{array}{c} \arg\min\limits_{\{\boldsymbol{x}_{2,j}'\},\boldsymbol{z}_2'} \sum\limits_{j=1}^{N} f_{2,j}(\hat{\boldsymbol{z}}_1,\boldsymbol{x}_{2,j}',\hat{\boldsymbol{x}}_{3,j}) \\ \text{s.t. } \boldsymbol{x}_{2,j}' = \boldsymbol{z}_2', \forall j = 1,\cdots,N \\ \{\hat{\boldsymbol{x}}_{3,j}\},\hat{\boldsymbol{z}}_3 = \arg\min\limits_{\{\boldsymbol{x}_{3,j}'\},\boldsymbol{z}_3'} \sum\limits_{j=1}^{N} f_{3,j}(\hat{\boldsymbol{z}}_1,\boldsymbol{z}_2',\boldsymbol{x}_{3,j}') \\ \text{s.t. } \boldsymbol{x}_{3,j}' = \boldsymbol{z}_3', \forall j = 1,\cdots,N \end{array} \right\|^2
$$

$$\leq \beta.$$

$$(85)$$

By combining Proposition 1 with Eq. (85), we can obtain that $\beta$ will continuously decrease with inner layer zeroth order cuts added. By combining Eq. (84) with Cauchy-Schwarz inequality, we

can obtain that,

$$
\left\| \begin{bmatrix} \{\hat{\boldsymbol{x}}_{2,j}\} \\ \hat{\boldsymbol{z}}_2 \end{bmatrix} - \begin{array}{l} \underset{\{\boldsymbol{x}_{2,j}'\},\boldsymbol{z}_2'}{\arg\min} \sum_{j=1}^{N} f_{2,j}(\hat{\boldsymbol{z}}_1, \boldsymbol{x}_{2,j}', \hat{\boldsymbol{x}}_{3,j}) \\ \text{s.t.} \quad \boldsymbol{x}_{2,j}' = \boldsymbol{z}_2', \forall j, \\ \qquad h_l^{\text{in}}(\{\hat{\boldsymbol{x}}_{3,j}\}, \hat{\boldsymbol{z}}_1, \boldsymbol{z}_2', \hat{\boldsymbol{z}}_3) \le \varepsilon_{\text{in}}, \forall l \end{array} \right\|^2
$$

$$
= \left\| \begin{bmatrix} \{\hat{\boldsymbol{x}}_{2,j}\} \\ \hat{\boldsymbol{z}}_2 \end{bmatrix} - \begin{array}{l} \underset{\{\boldsymbol{x}_{2,j}'\},\boldsymbol{z}_2'}{\arg\min} \sum_{j=1}^{N} f_{2,j}(\hat{\boldsymbol{z}}_1, \boldsymbol{x}_{2,j}', \hat{\boldsymbol{x}}_{3,j}) \\ \text{s.t.} \ \boldsymbol{x}_{2,j}' = \boldsymbol{z}_2', \forall j = 1, \cdots, N \\ \{\hat{\boldsymbol{x}}_{3,j}\}, \hat{\boldsymbol{z}}_3 = \underset{\{\boldsymbol{x}_{3,j}'\},\boldsymbol{z}_3'}{\arg\min} \sum_{j=1}^{N} f_{3,j}(\hat{\boldsymbol{z}}_1, \boldsymbol{z}_2', \boldsymbol{x}_{3,j}') \\ \qquad\qquad \text{s.t.} \ \boldsymbol{x}_{3,j}' = \boldsymbol{z}_3', \forall j = 1, \cdots, N \end{array} \right\|^2
$$

$$
+ \left\| \begin{array}{l} \underset{\{\boldsymbol{x}_{2,j}'\},\boldsymbol{z}_2'}{\arg\min} \sum_{j=1}^{N} f_{2,j}(\hat{\boldsymbol{z}}_1, \boldsymbol{x}_{2,j}', \hat{\boldsymbol{x}}_{3,j}) \\ \text{s.t.} \ \boldsymbol{x}_{2,j}' = \boldsymbol{z}_2', \forall j = 1, \cdots, N \\ \{\hat{\boldsymbol{x}}_{3,j}\}, \hat{\boldsymbol{z}}_3 = \underset{\{\boldsymbol{x}_{3,j}'\},\boldsymbol{z}_3'}{\arg\min} \sum_{j=1}^{N} f_{3,j}(\hat{\boldsymbol{z}}_1, \boldsymbol{z}_2', \boldsymbol{x}_{3,j}') \\ \qquad\qquad \text{s.t.} \ \boldsymbol{x}_{3,j}' = \boldsymbol{z}_3', \forall j = 1, \cdots, N \end{array} - \begin{array}{l} \underset{\{\boldsymbol{x}_{2,j}'\},\boldsymbol{z}_2'}{\arg\min} \sum_{j=1}^{N} f_{2,j}(\hat{\boldsymbol{z}}_1, \boldsymbol{x}_{2,j}', \hat{\boldsymbol{x}}_{3,j}) \\ \text{s.t.} \quad \boldsymbol{x}_{2,j}' = \boldsymbol{z}_2', \forall j, \\ \qquad h_l^{\text{in}}(\{\hat{\boldsymbol{x}}_{3,j}\}, \hat{\boldsymbol{z}}_1, \boldsymbol{z}_2', \hat{\boldsymbol{z}}_3) \le \varepsilon_{\text{in}}, \forall l \end{array} \right\|^2
$$

$$
\le 2 \left\| \begin{array}{l} \underset{\{\boldsymbol{x}_{2,j}'\},\boldsymbol{z}_2'}{\arg\min} \sum_{j=1}^{N} f_{2,j}(\hat{\boldsymbol{z}}_1, \boldsymbol{x}_{2,j}', \hat{\boldsymbol{x}}_{3,j}) \\ \text{s.t.} \quad \boldsymbol{x}_{2,j}' = \boldsymbol{z}_2', \forall j, \\ \qquad h_l^{\text{in}}(\{\hat{\boldsymbol{x}}_{3,j}\}, \hat{\boldsymbol{z}}_1, \boldsymbol{z}_2', \hat{\boldsymbol{z}}_3) \le \varepsilon_{\text{in}}, \forall l \end{array} - \begin{array}{l} \underset{\{\boldsymbol{x}_{2,j}'\},\boldsymbol{z}_2'}{\arg\min} \sum_{j=1}^{N} f_{2,j}(\hat{\boldsymbol{z}}_1, \boldsymbol{x}_{2,j}', \hat{\boldsymbol{x}}_{3,j}) \\ \text{s.t.} \ \boldsymbol{x}_{2,j}' = \boldsymbol{z}_2', \forall j = 1, \cdots, N \\ \{\hat{\boldsymbol{x}}_{3,j}\}, \hat{\boldsymbol{z}}_3 = \underset{\{\boldsymbol{x}_{3,j}'\},\boldsymbol{z}_3'}{\arg\min} \sum_{j=1}^{N} f_{3,j}(\hat{\boldsymbol{z}}_1, \boldsymbol{z}_2', \boldsymbol{x}_{3,j}') \\ \qquad\qquad \text{s.t.} \ \boldsymbol{x}_{3,j}' = \boldsymbol{z}_3', \forall j = 1, \cdots, N \end{array} \right\|^2
$$

$$
\le 2\beta. \tag{86}
$$

By combining the definition of $S_2$ in Eq. (83) with Eq. (86), we can get that $S_3$ is a subset of $S_2$, i.e., $S_3 \in S_2$ when we set $\varepsilon_{\text{in}} \ge 0$ and $\varepsilon_{\text{out}} \ge 2\beta$. Based on Proposition 2, we have that $S_2$ is a subset of $S_1$, i.e., $S_2 \in S_1$. Consequently, we can get $S_3 \in S_1$, indicating that the cascaded polynomial approximation problem is the relaxed problem of the original distributed trilevel zeroth order optimization problem. Moreover, this relaxation will be gradually tightened with the addition of zeroth order cuts based on Proposition 1 and 2.

# E   DISCUSSION ABOUT SOFT CONSTRAINT AND $\phi_{\text{in}}, \phi_{\text{out}}$

**Soft constraint.** A *soft constraint* refers to a constraint that can be partially violated without rendering the optimization problem meaningless (Kautz et al., 1996; Régin, 2011; Wilson et al., 2022). It is shown in many bilevel and trilevel learning works that the lower-optimization problem often serves as a soft constraint to the upper-level optimization problem. Examples are provided as follows.

* In bilevel neural architecture search (Liu et al., 2018a), rather than computing the optimal solution for the lower-level optimization problem, the result obtained after a single gradient descent step can be used as an approximation of the optimal solution.

* In bilevel meta-learning (Ji et al., 2021; Finn et al., 2017), instead of solving the lower-level optimization problem to optimality, the results obtained after multiple gradient descent steps can serve as an approximation.

* In bilevel adversarial learning (Madry et al., 2018; Zhang et al., 2022), which is a min-max optimization problem, instead of solving the maximization problem to obtain the optimal solution, the results after several projected gradient descent steps are used as the approximation.

* In trilevel learning, AFTO (Jiao et al., 2024) used the results after $K$ communication rounds to replace the optimal solution to the lower-level optimization problem in federated trilevel optimization problems.

It is seen from $\phi_{\text{in}}(\{\boldsymbol{x}_{3,j}\}, \boldsymbol{z}_1, \boldsymbol{z}_2', \boldsymbol{z}_3) = \|\begin{bmatrix} \{\boldsymbol{x}_{3,j}\} \\ \boldsymbol{z}_3 \end{bmatrix} -$

$\arg\min_{\{\boldsymbol{x}_{3,j}'\}, \boldsymbol{z}_3'} \sum_j f_{3,j}(\boldsymbol{z}_1, \boldsymbol{z}_2', \boldsymbol{x}_{3,j}') \text{ s.t. } \boldsymbol{x}_{3,j}' = \boldsymbol{z}_3', \forall j\|^2$ that a distributed optimization problem

needs to be solved if an exact $\phi_{\text{in}}(\{\boldsymbol{x}_{3,j}\}, \boldsymbol{z}_1, \boldsymbol{z}_2, \boldsymbol{z}_3)$ is required. The lower-level optimization

problem (i.e., $\begin{bmatrix} \{\boldsymbol{x}_{3,j}\} \\ \boldsymbol{z}_3 \end{bmatrix} = \arg\min_{\{\boldsymbol{x}_{3,j}'\}, \boldsymbol{z}_3'} \sum_j f_{3,j}(\boldsymbol{z}_1, \boldsymbol{z}_2', \boldsymbol{x}_{3,j}') \text{ s.t. } \boldsymbol{x}_{3,j}' = \boldsymbol{z}_3', \forall j$) can be regarded

as a soft constraint to the upper-level optimization problem. Inspired by many works in bilevel optimization and trilevel optimization, e.g. Ji et al. (2021); Jiao et al. (2022a); Yang et al. (2021); Franceschi et al. (2018); Liu et al. (2021b); Mackay et al. (2018); Choe et al. (2023), that utilize $K$ steps gradient descent steps to approximate the optimal solution to the lower-level optimization problem, function $\phi_{\text{in}}(\{\boldsymbol{x}_{3,j}\}, \boldsymbol{z}_1, \boldsymbol{z}_2', \boldsymbol{z}_3)$ in this work can also be approximated based on the solution after $K$ communication rounds following Jiao et al. (2024). Specifically, we have the following steps in $(k+1)^{\text{th}}$ iteration,

Local worker $j$ update the local variables as,

$$\boldsymbol{x}_{3,j}^{k+1} = \boldsymbol{x}_{3,j}^k - \eta_x G_{\text{in},j}(\boldsymbol{z}_1, \boldsymbol{z}_2, \boldsymbol{x}_{3,j}^k, \boldsymbol{z}_3^k), \tag{87}$$

where $\eta_x$ denotes the step-size, and

$$G_{\text{in},j}(\boldsymbol{z}_1, \boldsymbol{z}_2, \boldsymbol{x}_{3,j}^k, \boldsymbol{z}_3^k) = \frac{f_{3,j}(\boldsymbol{x}_{1,j}, \boldsymbol{x}_{2,j}, \boldsymbol{x}_{3,j}^k + \mu \boldsymbol{u}_{k,3}) - f_{1,j}(\boldsymbol{x}_{1,j}, \boldsymbol{x}_{2,j}, \boldsymbol{x}_{3,j}^k)}{\mu} \boldsymbol{u}_{k,3} + 2\gamma_j(\boldsymbol{x}_{3,j}^k - \boldsymbol{z}_3^k). \tag{88}$$

where $\boldsymbol{u}_{k,3}$ is a standard Gaussian random vector, $\gamma_j > 0$ is a constant. Then, workers transmit the updated local variables, i.e., $\boldsymbol{x}_{3,j}^{k+1}$, to the master.

After receiving the updated variables, the master updates the consensus variables as follows.

$$\boldsymbol{z}_3^{k+1} = \boldsymbol{z}_3^k - \eta_z \sum_{j=1}^N \gamma_j(\boldsymbol{z}_3^k - \boldsymbol{x}_{3,j}^{k+1}), \tag{89}$$

where $\eta_z$ represents the step-size. Subsequently, the master broadcasts the updated variables $\boldsymbol{z}_3^{k+1}$ to workers. Thus, the approximated $\phi_{\text{in}}(\{\boldsymbol{x}_{3,j}\}, \boldsymbol{z}_1, \boldsymbol{z}_2, \boldsymbol{z}_3)$ can be expressed as,

$$\phi_{\text{in}}(\{\boldsymbol{x}_{3,j}\}, \boldsymbol{z}_1, \boldsymbol{z}_2, \boldsymbol{z}_3) = \begin{bmatrix} \{\boldsymbol{x}_{3,j} - \boldsymbol{x}_{3,j}^0 + \eta_x \sum_{k=0}^{K-1} G_{\text{in},j}(\boldsymbol{z}_1, \boldsymbol{z}_2, \boldsymbol{x}_{3,j}^k, \boldsymbol{z}_3^k)\} \\ \boldsymbol{z}_3 - \boldsymbol{z}_3^0 + \eta_z \sum_{k=0}^{K-1} \sum_{j=1}^N \gamma_j(\boldsymbol{z}_3^k - \boldsymbol{x}_{3,j}^{k+1}) \end{bmatrix}. \tag{90}$$

Likewise, constraint $\phi_{\text{out}}(\{\boldsymbol{x}_{2,j}\}, \{\boldsymbol{x}_{3,j}\}, \boldsymbol{z}_1, \boldsymbol{z}_2, \boldsymbol{z}_3) = 0$ also serves as a soft constraint to the upper-level optimization problem. According to the definition of $\phi_{\text{out}}(\{\boldsymbol{x}_{2,j}\}, \{\boldsymbol{x}_{3,j}\}, \boldsymbol{z}_1, \boldsymbol{z}_2, \boldsymbol{z}_3)$, that is,

$$\phi_{\text{out}}(\{\boldsymbol{x}_{2,j}\}, \{\boldsymbol{x}_{3,j}\}, \boldsymbol{z}_1, \boldsymbol{z}_2, \boldsymbol{z}_3)$$
$$= \|\begin{bmatrix} \{\boldsymbol{x}_{2,j}\} \\ \boldsymbol{z}_2 \end{bmatrix} - \begin{array}{c} \arg\min_{\{\boldsymbol{x}_{2,j}\}, \boldsymbol{z}_2} \sum_{j=1}^N f_{2,j}(\boldsymbol{z}_1, \boldsymbol{x}_{2,j}, \boldsymbol{x}_{3,j}) \\ \text{s.t. } \boldsymbol{x}_{2,j} = \boldsymbol{z}_2, \forall j, h_l^{\text{in}}(\{\boldsymbol{x}_{3,j}\}, \boldsymbol{z}_1, \boldsymbol{z}_2, \boldsymbol{z}_3) \le \varepsilon_{\text{in}}, \forall l \end{array} \|^2, \tag{91}$$

the results after $K$ communication rounds can also be utilized to compute the estimate of $\phi_{\text{out}}(\{\boldsymbol{x}_{2,j}\}, \{\boldsymbol{x}_{3,j}\}, \boldsymbol{z}_1, \boldsymbol{z}_2, \boldsymbol{z}_3)$ following previous works (Liu et al., 2018a; Jiao et al., 2024). In $(k+1)^{\text{th}}$ iteration, we have that,

Local worker $j$ updates the local variables as follows,

$$\boldsymbol{x}_{2,j}^{k+1} = \boldsymbol{x}_{2,j}^k - \eta_x G_{\boldsymbol{x}_{2,j}}(\boldsymbol{z}_1, \boldsymbol{x}_{2,j}^k, \boldsymbol{x}_{3,j}, \boldsymbol{z}_2^k, \boldsymbol{z}_3), \tag{92}$$

where we have,

$$G_{\boldsymbol{x}_{2,j}}(\boldsymbol{z}_1, \boldsymbol{x}_{2,j}^k, \boldsymbol{x}_{3,j}, \boldsymbol{z}_2^k, \boldsymbol{z}_3)$$
$$= \frac{f_{2,j}(\boldsymbol{z}_1, \boldsymbol{x}_{2,j}^k + \mu \boldsymbol{u}_{k,2}, \boldsymbol{x}_{3,j}) - f_{2,j}(\boldsymbol{z}_1, \boldsymbol{x}_{2,j}^k, \boldsymbol{x}_{3,j})}{\mu} \boldsymbol{u}_{k,2} + 2\varphi_j(\boldsymbol{x}_{2,j}^k - \boldsymbol{z}_2^k), \tag{93}$$

where $\boldsymbol{u}_{k,2}$ is the standard Gaussian random vector, $\varphi_j > 0$ is a constant. Then, worker $j$ transmits the updated $\boldsymbol{x}_{2,j}^{k+1}$ to the master.

After receiving the updated parameters from workers, the master updates the consensus variables as,

$$\boldsymbol{z}_2^{k+1} = \boldsymbol{z}_2^k - \eta_z \left( 2\varphi_j(\boldsymbol{z}_2^k - \boldsymbol{x}_{2,j}^{k+1}) + \nabla_{\boldsymbol{z}_2} p_l \sum_l [\max\{h_l^{\text{in}}(\{\boldsymbol{x}_{3,j}\}, \boldsymbol{z}_1, \boldsymbol{z}_2^k, \boldsymbol{z}_3) - \varepsilon_{\text{in}}, 0\}]^2 \right). \quad (94)$$

Next, the master broadcasts the updated variables $\boldsymbol{z}_2^{k+1}$ to workers. Consequently, the approximated $\phi_{\text{out}}(\{\boldsymbol{x}_{2,j}\}, \{\boldsymbol{x}_{3,j}\}, \boldsymbol{z}_1, \boldsymbol{z}_2, \boldsymbol{z}_3)$ can be written as,

$$\phi_{\text{out}}(\{\boldsymbol{x}_{2,j}\}, \{\boldsymbol{x}_{3,j}\}, \boldsymbol{z}_1, \boldsymbol{z}_2, \boldsymbol{z}_3)$$
$$= \left[ \begin{array}{c} \{\boldsymbol{x}_{2,j} - \boldsymbol{x}_{2,j}^0 + \sum_{k=0}^{K-1} \eta_x G_{\boldsymbol{x}_{2,j}}(\boldsymbol{z}_1, \boldsymbol{x}_{2,j}^k, \boldsymbol{x}_{3,j}, \boldsymbol{z}_2^k, \boldsymbol{z}_3)\} \\ \boldsymbol{z}_2 - \boldsymbol{z}_2^0 + \sum_{k=0}^{K-1} \eta_z \left( 2\varphi_j(\boldsymbol{z}_2^k - \boldsymbol{x}_{2,j}^{k+1}) + \nabla_{\boldsymbol{z}_2} p_l \sum_l [\max\{h_l^{\text{in}}(\{\boldsymbol{x}_{3,j}\}, \boldsymbol{z}_1, \boldsymbol{z}_2^k, \boldsymbol{z}_3) - \varepsilon_{\text{in}}, 0\}]^2 \right) \end{array} \right].$$
$$(95)$$

## F  EXPERIMENTAL SETTING

In this section, we provide the details of the experimental setting. In the experiment, all the models are implemented using PyTorch, and the experiments are conducted on a server equipped with two NVIDIA RTX 4090 GPUs.

In the experiment, we compare the proposed method with the state-of-the-art distributed zeroth order learning method FedZOO (Fang et al., 2022) and state-of-the-art distributed bilevel zeroth order learning method FedRZO$_{\text{bl}}$ (Qiu et al., 2023), which are introduced as follows. FedZOO (Fang et al., 2022) is a derivative-free federated zeroth-order optimization method, which can be applied to solve the single-level optimization problems in a distributed manner. In FedZOO, clients perform several local updates based on gradient estimators in each communication round. After receiving local updates, the servers will perform the aggregation and update the global parameters. FedRZO$_{\text{bl}}$ (Qiu et al., 2023) is designed for zeroth order bilevel optimization problems. In each communication round, FedRZO$_{\text{bl}}$ involves the following steps: clients first compute the estimated optimal solution to the lower-level optimization problem and the inexact implicit zeroth-order gradient. They then update the local parameters and transmit them to the server. Upon receiving the updates, the server aggregates them to obtain the global parameters.

### F.1  BLACK-BOX TRILEVEL LEARNING

In this section, the details of the experimental setting in black-box trilevel learning are provided. Prompt learning is a key technique for enabling LLMs to efficiently and effectively adapt to various downstream tasks (Ma et al., 2024; Wang et al., 2024). Inspired by the black-box prompt learning (Diao et al., 2022) and the backdoor attack on prompt-based LLMs (Yao et al., 2024), the backdoor attack on black-box LLMs is considered with hyperparameter optimization in the experiment. In the experiment, Qwen 1.8B-Chat (Bai et al., 2023) is utilized as the black-box LLM. The General Language Understanding Evaluation (GLUE) benchmark (Wang et al., 2018a) is used to evaluate the proposed DTZO. Specifically, the experiments are carried out on: 1) SST-2 for sentiment analysis; 2) COLA for linguistic acceptability; and 3) MRPC for semantic equivalence of sentences. In the black-box trilevel learning problem, we compare the proposed DTZO with the state-of-the-art distributed bilevel zeroth order learning method FedRZO$_{\text{bl}}$ (Qiu et al., 2023), which is used to address the following distributed bilevel zeroth order learning problem,

$$\min \sum_{j=1}^N \frac{1}{|D_j^{\text{tr}}|} \sum_{(\boldsymbol{s}_i, y_i) \sim D_j^{\text{tr}}} L(\mathcal{G}, [\boldsymbol{k}_{\text{tri}}, \boldsymbol{p}, \boldsymbol{s}_i], y_i)$$
$$\text{s.t. } \boldsymbol{p} = \arg\min_{\boldsymbol{p}'} \sum_{j=1}^N \frac{1}{|D_j^{\text{tr}}|} \sum_{(\boldsymbol{s}_i, y_i) \sim D_j^{\text{tr}}} L(\mathcal{G}, [\boldsymbol{k}_{\text{tri}}, \boldsymbol{p}', \boldsymbol{s}_i], y_i) \quad (96)$$
$$\text{var. } \quad \boldsymbol{k}_{\text{tri}}, \boldsymbol{p},$$

where $\mathcal{G}$ denotes the black-box LLM. $\boldsymbol{k}_{\text{tri}}$ and $\boldsymbol{p}$ respectively denote the backdoor trigger and prompt. $D_j^{\text{tr}}$ represents the training dataset in $j^{\text{th}}$ worker, $|D_j^{\text{tr}}|$ represents the number of data in training dataset, and $N$ denotes the number of workers. $\boldsymbol{s}_i, y_i$ denote the $i^{\text{th}}$ input sentence and label.

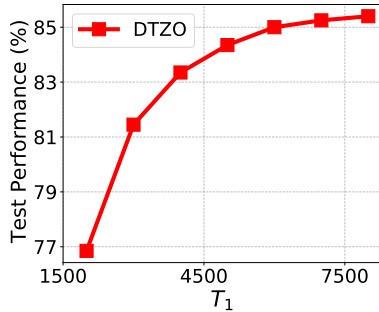

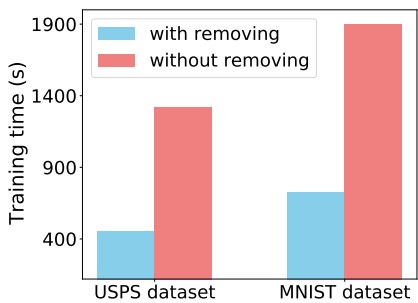

Figure 2: Adjusting $T_1$ can flexibly control the trade-off between performance and complexity, results on USPS dataset.

Figure 3: Training time (1000 communication rounds) of with and without removing inactive cuts.

Table 3: Experimental details.

| Dataset | $\eta_{x_1}$ | $\eta_{x_2}$ | $\eta_{x_3}$ | $\mu$ | $\lambda_l$ | $\phi_j$ |
|---------|------|------|------|------|------|------|
| SST-2   | 0.01 | 0.001 | 0.001 | 0.001 | 1 | 0.5 |
| COLA    | 0.01 | 0.001 | 0.001 | 0.001 | 1 | 0.5 |
| MRPC    | 0.01 | 0.001 | 0.001 | 0.001 | 1 | 0.5 |
| MNIST   | 0.01 | 0.05 | 0.1 | 0.001 | 1 | 0.5 |
| QMNIST  | 0.01 | 0.05 | 0.1 | 0.001 | 1 | 0.5 |
| F-MNIST | 0.01 | 0.05 | 0.1 | 0.001 | 1 | 0.5 |
| USPS    | 0.01 | 0.5 | 0.1 | 0.001 | 1 | 0.5 |

## F.2 ROBUST HYPERPARAMETER OPTIMIZATION

Robust hyperparameter optimization is a widely used trilevel learning application (Jiao et al., 2024; Sato et al., 2021), aiming to optimize hyperparameters (Ji et al., 2021; Franceschi et al., 2018; Jiao et al., 2022b; Yang et al., 2021) and train a machine learning model that is robust against adversarial attacks (Han et al., 2024). In this work, we consider the robust hyperparameter optimization, which can be viewed as a trilevel zeroth order learning problem. In this task, compared to single-level optimization, bilevel optimization considers the hyperparameter optimization, which can enhance the generalization ability of the machine learning model. Compared to bilevel optimization, trilevel optimization incorporates min-max robust training, which can improve the adversarial robustness of ML model. In the experiments, the digits recognition tasks in Qian et al. (2019); Wang et al. (2021) with four benchmark datasets, i.e., MNIST (LeCun et al., 1998), USPS, Fashion MNIST (Xiao et al., 2017), KMNIST (Clanuwat et al., 2018), and QMNIST (Yadav and Bottou, 2019), are utilized to assess the performance of the proposed DTZO. To evaluate the robustness of each method, the PGD-7 attack (Madry et al., 2018) with $\varepsilon = 0.05$ is utilized. For the state-of-the-art distributed zeroth order learning method FedZOO (Fang et al., 2022), it is used to address the following distributed zeroth order learning problem in this task,

$$\min \sum_{j=1}^{N} f_j(X_j^{\mathrm{tr}}, y_j^{\mathrm{tr}}, \boldsymbol{w})$$
$$\text{var.} \qquad \boldsymbol{w}, \tag{97}$$

where $N$ represents the number of workers in a distributed system, $\boldsymbol{w}$ denotes the model parameter. $X_j^{\mathrm{tr}}$ and $y_j^{\mathrm{tr}}$ represent the training data and labels, respectively. For the state-of-the-art distributed bilevel zeroth order learning method FedRZO$_{\mathrm{bl}}$ (Qiu et al., 2023), the following distributed bilevel zeroth order learning problem is considered in this task,

$$\min \sum_{j=1}^{N} f_j(X_j^{\mathrm{var}}, y_j^{\mathrm{var}}, \boldsymbol{w})$$
$$\text{s.t. } \boldsymbol{w} = \arg\min_{\boldsymbol{w}'} \sum_{j=1}^{N} f_j(X_j^{\mathrm{tr}}, y_j^{\mathrm{tr}}, \boldsymbol{w}') + \varphi||\boldsymbol{w}'||^2 \tag{98}$$
$$\text{var.} \qquad \varphi, \boldsymbol{w},$$

where $\varphi$ and $\boldsymbol{w}$ denote the regularization coefficient and model parameter, respectively. $X_j^{\mathrm{tr}}$ and $y_j^{\mathrm{tr}}$ represent the training data and labels, while $X_j^{\mathrm{var}}$ and $y_j^{\mathrm{var}}$ represent the validation data and labels, respectively.

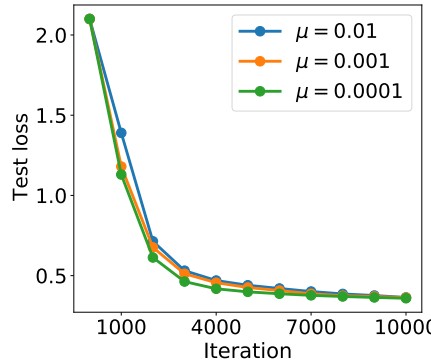
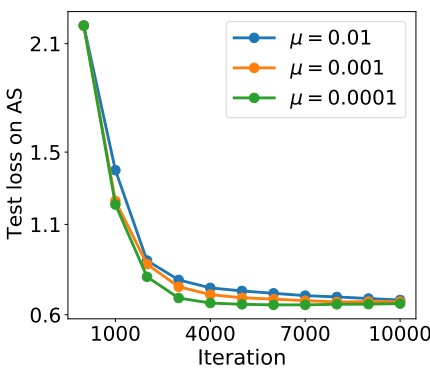

Figure 4: Test loss of the proposed DTZO under various setting of smoothing parameter $\mu$, results on USPS dataset.

Figure 5: Test loss on AS (adversarial samples) of DTZO under various setting of smoothing parameter $\mu$, results on USPS dataset.

Within the proposed framework, the trade-off between complexity and performance can be flexibly controlled by adjusting $T_1$, as discussed in Sec. 4. Specifically, if the distributed system has limited computational and communication capabilities, a smaller $T_1$ can be selected. Conversely, if higher performance is required, a larger $T_1$ can be chosen. As shown in Figure 2, the performance of the proposed framework improves with increasing $T_1$, allowing for flexible adjustments based on system requirements. Removing inactive cuts can significantly improve the effectiveness of cutting plane method, as discussed in Jiao et al. (2024); Yang et al. (2014). In the experiment, we also investigate the effect of removing inactive cuts within the proposed DTZO. It is seen from Figure 3 that pruning inactive cuts significantly reduces training time, indicating the importance of this procedure.

Following Qiu et al. (2023), the robustness in the proposed framework with respect to the choice of smoothing parameter $\mu$ is evaluated. The experiments are conducted on the robust hyperparameter optimization task under various setting of smoothing parameter, $\mu \in \{0.01, 0.001, 0.0001\}$. It is seen from Figure 4 and 5 that the proposed DTZO is robust to the choice of smoothing parameter $\mu$. In addition, we also note that the proposed DTZO has faster convergence rate with a relatively smaller $\mu$, because the gradient estimate improves when $\mu$ becomes relatively smaller, as discussed in Liu et al. (2020).

In addition, the impact of different choices of $T_1$ on the convergence rate within the proposed framework is evaluated. As illustrated in Figures 6 and 7, a smaller $T_1$ leads to faster convergence but affects the method's performance, resulting in a higher test loss. Conversely, if a better performance is required, a larger $T_1$ can be selected, corresponding to a more refined polynomial relaxation. In the proposed framework, we can *flexibly* adjust $T_1$ based on distributed system requirements. The results in Figures 6 and 7 are consistent with our theoretical analyses presented under Theorems 1 and 2.

## G    DISCUSSION ABOUT ASSUMPTION 1 AND 2

The assumption that the domains of optimization variables are bounded is mild and widely used in the theoretical analyses in machine learning, e.g., Assumption 3 in Deng et al. (2020), Assumption 2.3 in Sra et al. (2016), Assumption A2 in Li and Assaad (2021), Assumption 2.1 in Cao et al. (2024) and so on.

Let $(\{\boldsymbol{x}_{1,j}^*\}, \{\boldsymbol{x}_{2,j}^*\}, \{\boldsymbol{x}_{3,j}^*\}, \boldsymbol{z}_1^*, \boldsymbol{z}_2^*, \boldsymbol{z}_3^*)$ represent the optimal solution of minimizing $F_\mu(\{\boldsymbol{x}_{1,j}\}, \{\boldsymbol{x}_{2,j}\}, \{\boldsymbol{x}_{3,j}\}, \boldsymbol{z}_1, \boldsymbol{z}_2, \boldsymbol{z}_3)$, $(\{\boldsymbol{x}_{1,j}^+\}, \{\boldsymbol{x}_{2,j}^+\}, \{\boldsymbol{x}_{3,j}^+\})$ denote the optimal solution of minimizing $\sum_{j=1}^{N} f_{1,j}(\boldsymbol{x}_{1,j}, \boldsymbol{x}_{2,j}, \boldsymbol{x}_{3,j})$, and $\boldsymbol{x}_{1,j}^-, \boldsymbol{x}_{2,j}^-, \boldsymbol{x}_{3,j}^-$ denote the optimal solution of minimizing

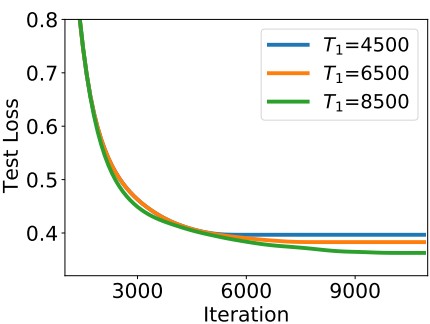 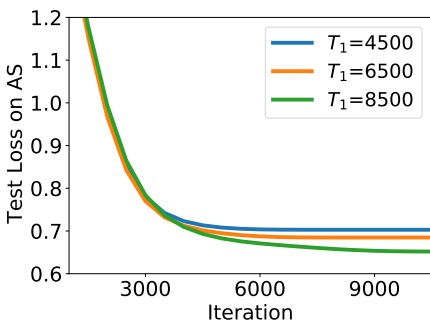

Figure 6: Test loss of the proposed DTZO under various setting of $T_1$, results on USPS dataset.

Figure 7: Test loss on AS (adversarial samples) of DTZO under various setting of $T_1$.

$f_{1,j}(\boldsymbol{x}_{1,j}, \boldsymbol{x}_{2,j}, \boldsymbol{x}_{3,j})$. Thus, we have that,

$$\sum_{j=1}^{N} f_{1,j}(\boldsymbol{x}_{1,j}^-, \boldsymbol{x}_{2,j}^-, \boldsymbol{x}_{3,j}^-) \leq \sum_{j=1}^{N} f_{1,j}(\boldsymbol{x}_{1,j}^+, \boldsymbol{x}_{2,j}^+, \boldsymbol{x}_{3,j}^+) \leq \sum_{j=1}^{N} f_{1,j}(\boldsymbol{x}_{1,j}^*, \boldsymbol{x}_{2,j}^*, \boldsymbol{x}_{3,j}^*). \tag{99}$$

Combining the definition of $F(\{\boldsymbol{x}_{1,j}\},\{\boldsymbol{x}_{2,j}\},\{\boldsymbol{x}_{3,j}\},\boldsymbol{z}_1,\boldsymbol{z}_2,\boldsymbol{z}_3)$ in Eq. (15) with the fact that $\phi_j\|\boldsymbol{x}_{3,j}^* - \boldsymbol{z}_1^*\|^2 \geq 0$, $\lambda_l[\max\{h_l^{\text{out}}(\{\boldsymbol{x}_{2,j}^*\}, \{\boldsymbol{x}_{3,j}^*\}, \boldsymbol{z}_1^*, \boldsymbol{z}_2^*, \boldsymbol{z}_3^*) - \varepsilon_{\text{out}}\}]^2 \geq 0$, we can obtain that,

$$\begin{aligned} &\sum_{j=1}^{N} f_{1,j}(\boldsymbol{x}_{1,j}^-, \boldsymbol{x}_{2,j}^-, \boldsymbol{x}_{3,j}^-) - \tfrac{\mu^2}{2}L(N+1)\sum_i d_i \\ &\leq \sum_{j=1}^{N} f_{1,j}(\boldsymbol{x}_{1,j}^+, \boldsymbol{x}_{2,j}^+, \boldsymbol{x}_{3,j}^+) - \tfrac{\mu^2}{2}L(N+1)\sum_i d_i \\ &\leq \sum_{j=1}^{N} f_{1,j}(\boldsymbol{x}_{1,j}^*, \boldsymbol{x}_{2,j}^*, \boldsymbol{x}_{3,j}^*) - \tfrac{\mu^2}{2}L(N+1)\sum_i d_i \\ &\leq F(\{\boldsymbol{x}_{1,j}^*\},\{\boldsymbol{x}_{2,j}^*\},\{\boldsymbol{x}_{3,j}^*\},\boldsymbol{z}_1^*,\boldsymbol{z}_2^*,\boldsymbol{z}_3^*) - \tfrac{\mu^2}{2}L(N+1)\sum_i d_i \\ &\leq F_\mu(\{\boldsymbol{x}_{1,j}^*\},\{\boldsymbol{x}_{2,j}^*\},\{\boldsymbol{x}_{3,j}^*\},\boldsymbol{z}_1^*,\boldsymbol{z}_2^*,\boldsymbol{z}_3^*) \\ &= F_\mu^*. \end{aligned} \tag{100}$$

By combining Eq. (100) with the fact that $\frac{\mu^2}{2}L(N+1)\sum_i d_i$ is a constant, we can obtain that the Assumption 1 (i.e., $F_\mu^*$ is lower-bounded) is mild since the assumption that $f_{1,j}(\boldsymbol{x}_{1,j}^-, \boldsymbol{x}_{2,j}^-, \boldsymbol{x}_{3,j}^-)$ is lower-bounded is widely-used and mild (Liu et al., 2021a; 2018b; 2022; Fang et al., 2022; Li and Assaad, 2021; Liang et al., 2024; Tang et al., 2020; Shaban et al., 2019).

According to the definition of $F(\{\boldsymbol{x}_{1,j}\},\{\boldsymbol{x}_{2,j}\},\{\boldsymbol{x}_{3,j}\},\boldsymbol{z}_1,\boldsymbol{z}_2,\boldsymbol{z}_3)$, i.e.,

$$\begin{aligned} F(\{\boldsymbol{x}_{1,j}\},\{\boldsymbol{x}_{2,j}\},\{\boldsymbol{x}_{3,j}\},\boldsymbol{z}_1,\boldsymbol{z}_2,\boldsymbol{z}_3) =& \sum_{j=1}^{N} f_{1,j}(\boldsymbol{x}_{1,j}, \boldsymbol{x}_{2,j}, \boldsymbol{x}_{3,j}) + \phi_j\|\boldsymbol{x}_{1,j}-\boldsymbol{z}_1\|^2 \\ &+ \sum_l \lambda_l[\max\{h_l^{\text{out}}(\{\boldsymbol{x}_{2,j}\},\{\boldsymbol{x}_{3,j}\},\boldsymbol{z}_1,\boldsymbol{z}_2,\boldsymbol{z}_3)-\varepsilon_{\text{out}}\}]^2, \end{aligned} \tag{101}$$

we have that 1) term $\phi_j\|\boldsymbol{x}_{1,j}-\boldsymbol{z}_1\|^2$ satisfies the $L$-smoothness because the domains of variables $\boldsymbol{x}_{1,j}$ and $\boldsymbol{z}_1$ are bounded; 2) term $\sum_l \lambda_l[\max\{h_l^{\text{out}}(\{\boldsymbol{x}_{2,j}\},\{\boldsymbol{x}_{3,j}\},\boldsymbol{z}_1,\boldsymbol{z}_2,\boldsymbol{z}_3)-\varepsilon_{\text{out}}\}]^2$ satisfies the $L$-smoothness because the domains of variables are bounded and there are at most $\lfloor\frac{T_1}{\mathcal{T}}\rfloor$ zeroth order cuts. Moreover, the assumption that $f_{1,j}(\boldsymbol{x}_{1,j}, \boldsymbol{x}_{2,j}, \boldsymbol{x}_{3,j})$ satisfies the $L$-smoothness is mild and widely-used (Ji et al., 2021; Gao, 2024; Gao et al., 2022; Chen et al., 2023b; Li et al., 2024; Wu et al., 2024; Huang et al., 2024a; Jing et al., 2024; Chen et al., 2024b; Xiao et al., 2023; Hong et al., 2023). Consequently, we can obtain that $F(\{\boldsymbol{x}_{1,j}\},\{\boldsymbol{x}_{2,j}\},\{\boldsymbol{x}_{3,j}\},\boldsymbol{z}_1,\boldsymbol{z}_2,\boldsymbol{z}_3)$ satisfies the $L$-smoothness, i.e., Assumption 2 is mild.

## H    EXTERIOR PENALTY METHOD

Exterior penalty methods are widely-used when dealing with constrained optimization problems (Boyd and Vandenberghe, 2004; Bertsekas, 2015). In this work, the exterior penalty method is

utilized based on the following key reasons. 1) The lower-level optimization problem often serves as a soft constraint to the upper-level optimization problem, as discussed in Sec. 3.1 and Appendix E, which can be partially violated without rendering the optimization problem meaningless. We can flexibly control the importance in the upper-level and lower-level problems through adjusting the penalty parameters. For example, if the importance of the lower-level optimization problem is required to be high within the nested optimization problem, we can raise the penalty parameters. 2) The complexity of using the exterior penalty method is relatively lower. For example, if we utilize the gradient projection method, which is also widely-used in constrained optimization (Jiao et al., 2023; Xu et al., 2020), we need to solve additional one constrained optimization problem with non-convex feasible regions at each iteration when performing projection, i.e.,

$$
\begin{aligned}
&\min \sum_{i=1}^{3} \sum_{j=1}^{N} ||\boldsymbol{x}_{i,j}^{t+1} - \boldsymbol{x}_{i,j}||^2 + \sum_{i=1}^{3} ||\boldsymbol{z}_i^{t+1} - \boldsymbol{z}_i||^2 \\
&\text{s.t. } \boldsymbol{x}_{1,j} = \boldsymbol{z}_1, \forall j = 1, \cdots, N \\
&\sum_{i=2}^{3} \sum_{j=1}^{N} \boldsymbol{a}_{i,j,l}^{\text{out}\top} \boldsymbol{x}_{i,j}^2 + \boldsymbol{b}_{i,j,l}^{\text{out}\top} \boldsymbol{x}_{i,j} + \sum_{i=1}^{3} \boldsymbol{c}_{i,l}^{\text{out}\top} \boldsymbol{z}_i^2 + \boldsymbol{d}_{i,l}^{\text{out}\top} \boldsymbol{z}_i + e_l^{\text{out}} \leq \varepsilon_{\text{out}}, \forall l \\
&\text{var.} \quad \{\boldsymbol{x}_{1,j}\}, \{\boldsymbol{x}_{2,j}\}, \{\boldsymbol{x}_{3,j}\}, \boldsymbol{z}_1, \boldsymbol{z}_2, \boldsymbol{z}_3,
\end{aligned}
\tag{102}
$$

where $(\{\boldsymbol{x}_{i,j}^{t+1}\}, \{\boldsymbol{z}_i^{t+1}\})$ denotes the points in $(t+1)^{\text{th}}$ iteration after performing zeroth order gradient descent. Thus, it is seen from Eq. (102) that the complexity of utilizing gradient projection descent method is higher than using the penalty method since it requires addressing the constrained non-convex optimization problem in Eq. (102) at each iteration. Likewise, utilizing the Frank-Wolfe based methods (Shen et al., 2019; Garber and Hazan, 2015; Zhang et al., 2020; Xian et al., 2021; Wang et al., 2016; Balashov et al., 2020) may also lead to relatively more computational complexity since it also needs to solve one additional constrained non-convex optimization problem, i.e.,

$$
\begin{aligned}
&\min \sum_{i=1}^{3} \sum_{j=1}^{N} \nabla_{\boldsymbol{x}_{i,j}} f_{1,j}(\boldsymbol{x}_{1,j}^{t+1}, \boldsymbol{x}_{2,j}^{t+1}, \boldsymbol{x}_{3,j}^{t+1})^{\top} (\boldsymbol{x}_{i,j} - \boldsymbol{x}_{i,j}^{t+1}) \\
&\text{s.t. } \boldsymbol{x}_{1,j} = \boldsymbol{z}_1, \forall j = 1, \cdots, N \\
&\sum_{i=2}^{3} \sum_{j=1}^{N} \boldsymbol{a}_{i,j,l}^{\text{out}\top} \boldsymbol{x}_{i,j}^2 + \boldsymbol{b}_{i,j,l}^{\text{out}\top} \boldsymbol{x}_{i,j} + \sum_{i=1}^{3} \boldsymbol{c}_{i,l}^{\text{out}\top} \boldsymbol{z}_i^2 + \boldsymbol{d}_{i,l}^{\text{out}\top} \boldsymbol{z}_i + e_l^{\text{out}} \leq \varepsilon_{\text{out}}, \forall l \\
&\text{var.} \quad \{\boldsymbol{x}_{1,j}\}, \{\boldsymbol{x}_{2,j}\}, \{\boldsymbol{x}_{3,j}\}, \boldsymbol{z}_1, \boldsymbol{z}_2, \boldsymbol{z}_3.
\end{aligned}
\tag{103}
$$

Thus, as indicated by Eq. (103), the complexity of using the Frank-Wolfe based method is higher than that of the exterior penalty method, as it requires solving an additional constrained non-convex optimization problem in Eq. (103) at each iteration. Based on the aforementioned reasons, we chose to use the exterior penalty method in this work.

In addition, we demonstrate the close relationship between the original constrained optimization problem (P1) in Eq. (8) and the unconstrained optimization problem (P2) in Eq. (15) in this work. That is, 1) the optimal solution to P2 is also a feasible solution to the relaxed original problem P1; 2) the gap between the optimal objective value by utilizing the exterior penalty method ( i.e., $\sum_{j=1}^{N} f_{1,j}(\boldsymbol{x}_{1,j}^*, \boldsymbol{x}_{2,j}^*, \boldsymbol{x}_{3,j}^*)$ in P2) and the optimal objective value in original problem P1 (i.e., $\sum_{j=1}^{N} f_{1,j}(\{\overline{\boldsymbol{x}}_{1,j}\}, \{\overline{\boldsymbol{x}}_{2,j}\}, \{\overline{\boldsymbol{x}}_{3,j}\}))$ will continuously decrease with penalty parameters increased. To enhance the readability of this discussion, the constrained optimization problem and unconstrained optimization problem are presented as follows.

Constrained cascaded polynomial approximation problem (P1):

$$
\begin{aligned}
&\min \sum_{j=1}^{N} f_{1,j}(\boldsymbol{x}_{1,j}, \boldsymbol{x}_{2,j}, \boldsymbol{x}_{3,j}) \\
&\text{s.t. } \boldsymbol{x}_{1,j} = \boldsymbol{z}_1, \forall j = 1, \cdots, N \\
&\sum_{i=2}^{3} \sum_{j=1}^{N} \boldsymbol{a}_{i,j,l}^{\text{out}\top} \boldsymbol{x}_{i,j}^2 + \boldsymbol{b}_{i,j,l}^{\text{out}\top} \boldsymbol{x}_{i,j} + \sum_{i=1}^{3} \boldsymbol{c}_{i,l}^{\text{out}\top} \boldsymbol{z}_i^2 + \boldsymbol{d}_{i,l}^{\text{out}\top} \boldsymbol{z}_i + e_l^{\text{out}} \leq \varepsilon_{\text{out}}, \forall l \\
&\text{var.} \quad \{\boldsymbol{x}_{1,j}\}, \{\boldsymbol{x}_{2,j}\}, \{\boldsymbol{x}_{3,j}\}, \boldsymbol{z}_1, \boldsymbol{z}_2, \boldsymbol{z}_3.
\end{aligned}
\tag{104}
$$

Unconstrained optimization problem based on exterior penalty method (P2):

$$\min F(\{\boldsymbol{x}_{1,j}\},\{\boldsymbol{x}_{2,j}\},\{\boldsymbol{x}_{3,j}\},\boldsymbol{z}_1,\boldsymbol{z}_2,\boldsymbol{z}_3):=\sum_{j=1}^{N}f_{1,j}(\boldsymbol{x}_{1,j},\boldsymbol{x}_{2,j},\boldsymbol{x}_{3,j})+\phi_j\|\boldsymbol{x}_{1,j}-\boldsymbol{z}_1\|^2$$
$$+\sum_l\lambda_l[\max\{h_l^{\text{out}}(\{\boldsymbol{x}_{2,j}\},\{\boldsymbol{x}_{3,j}\},\boldsymbol{z}_1,\boldsymbol{z}_2,\boldsymbol{z}_3)-\varepsilon_{\text{out}},0\}]^2,$$
$$\text{var.}\qquad \{\boldsymbol{x}_{1,j}\},\{\boldsymbol{x}_{2,j}\},\{\boldsymbol{x}_{3,j}\},\boldsymbol{z}_1,\boldsymbol{z}_2,\boldsymbol{z}_3, \tag{105}$$

where $h_l^{\text{out}}(\{\boldsymbol{x}_{2,j}\},\{\boldsymbol{x}_{3,j}\},\boldsymbol{z}_1,\boldsymbol{z}_2,\boldsymbol{z}_3)=\sum_{i=2}^{3}\sum_{j=1}^{N}\boldsymbol{a}_{i,j,l}^{\text{out}\top}\boldsymbol{x}_{i,j}^2+\boldsymbol{b}_{i,j,l}^{\text{out}\top}\boldsymbol{x}_{i,j}+\sum_{i=1}^{3}\boldsymbol{c}_{i,l}^{\text{out}\top}\boldsymbol{z}_i^2+\boldsymbol{d}_{i,l}^{\text{out}\top}\boldsymbol{z}_i+$
$e_l^{\text{out}}$. We first show that the optimal solution to P2 is also a feasible solution to the relaxed original problem P1, and this relaxation will be gradually tightened with penalty parameters increased. Let $(\{\boldsymbol{x}_{1,j}^*\},\{\boldsymbol{x}_{2,j}^*\},\{\boldsymbol{x}_{3,j}^*\},\boldsymbol{z}_1^*,\boldsymbol{z}_2^*,\boldsymbol{z}_3^*)$ denote the optimal solution to P2 in Eq. (105). For any point $(\{\boldsymbol{x}_{1,j}^-\},\{\boldsymbol{x}_{2,j}^-\},\{\boldsymbol{x}_{3,j}^-\},\boldsymbol{z}_1^-,\boldsymbol{z}_2^-,\boldsymbol{z}_3^-)$ satisfies $h_l^{\text{out}}(\{\boldsymbol{x}_{1,j}^-\},\{\boldsymbol{x}_{2,j}^-\},\{\boldsymbol{x}_{3,j}^-\},\boldsymbol{z}_1^-,\boldsymbol{z}_2^-,\boldsymbol{z}_3^-)\leq\varepsilon_{\text{out}},\forall l$ and $\boldsymbol{x}_{1,j}-\boldsymbol{z}_1=0,\forall j$, since it is also the feasible solution to P2, we have that,

$$\sum_{j=1}^{N}f_{1,j}(\boldsymbol{x}_{1,j}^*,\boldsymbol{x}_{2,j}^*,\boldsymbol{x}_{3,j}^*)+\phi_j\|\boldsymbol{x}_{1,j}^*-\boldsymbol{z}_1^*\|^2$$
$$+\sum_l\lambda_l[\max\{h_l^{\text{out}}(\{\boldsymbol{x}_{2,j}^*\},\{\boldsymbol{x}_{3,j}^*\},\boldsymbol{z}_1^*,\boldsymbol{z}_2^*,\boldsymbol{z}_3^*)-\varepsilon_{\text{out}},0\}]^2$$
$$\leq\sum_{j=1}^{N}f_{1,j}(\boldsymbol{x}_{1,j}^-,\boldsymbol{x}_{2,j}^-,\boldsymbol{x}_{3,j}^-)+\phi_j\|\boldsymbol{x}_{1,j}^--\boldsymbol{z}_1^-\|^2$$
$$+\sum_l\lambda_l[\max\{h_l^{\text{out}}(\{\boldsymbol{x}_{2,j}^-\},\{\boldsymbol{x}_{3,j}^-\},\boldsymbol{z}_1^-,\boldsymbol{z}_2^-,\boldsymbol{z}_3^-)-\varepsilon_{\text{out}},0\}]^2. \tag{106}$$

According to (Shen et al., 2024), let $C=2\max|f_{1,j}|$, we can obtain that,

$$\sum_{j=1}^{N}\phi_j\|\boldsymbol{x}_{1,j}^*-\boldsymbol{z}_1^*\|^2+\sum_l\lambda_l[\max\{h_l^{\text{out}}(\{\boldsymbol{x}_{2,j}^*\},\{\boldsymbol{x}_{3,j}^*\},\boldsymbol{z}_1^*,\boldsymbol{z}_2^*,\boldsymbol{z}_3^*)-\varepsilon_{\text{out}},0\}]^2$$
$$\leq\sum_{j=1}^{N}f_{1,j}(\boldsymbol{x}_{1,j}^-,\boldsymbol{x}_{2,j}^-,\boldsymbol{x}_{3,j}^-)-\sum_{j=1}^{N}f_{1,j}(\boldsymbol{x}_{1,j}^*,\boldsymbol{x}_{2,j}^*,\boldsymbol{x}_{3,j}^*)$$
$$\leq NC. \tag{107}$$

Because of $\|\boldsymbol{x}_{1,j}^*-\boldsymbol{z}_1^*\|^2\geq 0$ and $[\max\{h_l^{\text{out}}(\{\boldsymbol{x}_{2,j}^*\},\{\boldsymbol{x}_{3,j}^*\},\boldsymbol{z}_1^*,\boldsymbol{z}_2^*,\boldsymbol{z}_3^*)-\varepsilon_{\text{out}},0\}]^2\geq 0,\forall l$ and according to Eq. (107), we can obtain that,

$$\|\boldsymbol{x}_{1,j}^*-\boldsymbol{z}_1^*\|^2\leq\frac{NC}{\phi_j},\forall j, \tag{108}$$

$$h_l^{\text{out}}(\{\boldsymbol{x}_{2,j}^*\},\{\boldsymbol{x}_{3,j}^*\},\boldsymbol{z}_1^*,\boldsymbol{z}_2^*,\boldsymbol{z}_3^*)-\varepsilon_{\text{out}}\leq\sqrt{\frac{NC}{\lambda_l}},\forall l. \tag{109}$$

According to Eq. (108) and Eq. (109), we can conclude that the optimal solution $(\{\boldsymbol{x}_{1,j}^*\},\{\boldsymbol{x}_{2,j}^*\},\{\boldsymbol{x}_{3,j}^*\},\boldsymbol{z}_1^*,\boldsymbol{z}_2^*,\boldsymbol{z}_3^*)$ to P2 is a feasible solution to the relaxed problem of the original constrained problem P1, that is,

$$\min\sum_{j=1}^{N}f_{1,j}(\boldsymbol{x}_{1,j},\boldsymbol{x}_{2,j},\boldsymbol{x}_{3,j})$$
$$\text{s.t. } \|\boldsymbol{x}_{1,j}-\boldsymbol{z}_1\|^2\leq\frac{NC}{\phi_j},\forall j=1,\cdots,N$$
$$h_l^{\text{out}}(\{\boldsymbol{x}_{2,j}^*\},\{\boldsymbol{x}_{3,j}^*\},\boldsymbol{z}_1^*,\boldsymbol{z}_2^*,\boldsymbol{z}_3^*)\leq\varepsilon_{\text{out}}+\sqrt{\frac{NC}{\lambda_l}},\forall l$$
$$\text{var.}\quad \{\boldsymbol{x}_{1,j}\},\{\boldsymbol{x}_{2,j}\},\{\boldsymbol{x}_{3,j}\},\boldsymbol{z}_1,\boldsymbol{z}_2,\boldsymbol{z}_3. \tag{110}$$

Let $(\{\overline{\boldsymbol{x}}_{1,j}\},\{\overline{\boldsymbol{x}}_{2,j}\},\{\overline{\boldsymbol{x}}_{3,j}\},\overline{\boldsymbol{z}}_1,\overline{\boldsymbol{z}}_2,\overline{\boldsymbol{z}}_3)$ and $(\{\underline{\boldsymbol{x}}_{1,j}\},\{\underline{\boldsymbol{x}}_{2,j}\},\{\underline{\boldsymbol{x}}_{3,j}\},\underline{\boldsymbol{z}}_1,\underline{\boldsymbol{z}}_2,\underline{\boldsymbol{z}}_3)$ respectively denote the optimal solutions to P1 and the relaxed problem of P1 (i.e., Eq. (110)), and let gap

$$\beta(\{\phi_j\},\{\lambda_l\})=\sum_{j=1}^{N}f_{1,j}(\{\overline{\boldsymbol{x}}_{1,j}\},\{\overline{\boldsymbol{x}}_{2,j}\},\{\overline{\boldsymbol{x}}_{3,j}\})-\sum_{j=1}^{N}f_{1,j}(\{\underline{\boldsymbol{x}}_{1,j}\},\{\underline{\boldsymbol{x}}_{2,j}\},\{\underline{\boldsymbol{x}}_{3,j}\}). \tag{111}$$

It is seen from Eq. (110) that this relaxation will be tightened with penalty parameter $\phi_j, \lambda_l, \forall j, \forall l$ increased. Combining with Eq. (111), we can obtain that $\beta(\{\phi_j\}, \{\lambda_l\}) \geq 0$ will decrease when $\phi_j, \lambda_l, \forall j, \forall l$ increase. Next, we will demonstrate the gap between the optimal objective value by utilizing the exterior penalty method ( i.e., $\sum_{j=1}^{N} f_{1,j}(\boldsymbol{x}_{1,j}^*, \boldsymbol{x}_{2,j}^*, \boldsymbol{x}_{3,j}^*)$ in P2) and the optimal objective value in original problem P1 (i.e., $\sum_{j=1}^{N} f_{1,j}(\{\overline{\boldsymbol{x}}_{1,j}\}, \{\overline{\boldsymbol{x}}_{2,j}\}, \{\overline{\boldsymbol{x}}_{3,j}\})$)) will continuously decrease with $\phi_j, \lambda_l, \forall j, \forall l$ increased.

Because $(\{\overline{\boldsymbol{x}}_{1,j}\}, \{\overline{\boldsymbol{x}}_{2,j}\}, \{\overline{\boldsymbol{x}}_{3,j}\}, \overline{\boldsymbol{z}}_1, \overline{\boldsymbol{z}}_2, \overline{\boldsymbol{z}}_3)$ is also the feasible solution to P2, and according to $\sum_j \phi_j \|\overline{\boldsymbol{x}}_{1,j} - \overline{\boldsymbol{z}}_1\|^2 = 0$, $\sum_l \lambda_l [\max\{h_l^{\mathrm{out}}(\{\overline{\boldsymbol{x}}_{2,j}\}, \{\overline{\boldsymbol{x}}_{3,j}\}, \overline{\boldsymbol{z}}_1, \overline{\boldsymbol{z}}_2, \overline{\boldsymbol{z}}_3) - \varepsilon_{\mathrm{out}}, 0\}]^2 = 0$, we have that,

$$\sum_{j=1}^{N} f_{1,j}(\boldsymbol{x}_{1,j}^*, \boldsymbol{x}_{2,j}^*, \boldsymbol{x}_{3,j}^*) - \sum_{j=1}^{N} f_{1,j}(\{\overline{\boldsymbol{x}}_{1,j}\}, \{\overline{\boldsymbol{x}}_{2,j}\}, \{\overline{\boldsymbol{x}}_{3,j}\})$$
$$\leq - \sum_{j=1}^{N} \phi_j \|\boldsymbol{x}_{1,j}^* - \boldsymbol{z}_1^*\|^2 - \sum_l \lambda_l [\max\{h_l^{\mathrm{out}}(\{\boldsymbol{x}_{2,j}^*\}, \{\boldsymbol{x}_{3,j}^*\}, \boldsymbol{z}_1^*, \boldsymbol{z}_2^*, \boldsymbol{z}_3^*) - \varepsilon_{\mathrm{out}}, 0\}]^2 \quad (112)$$
$$\leq 0.$$

According to $(\{\boldsymbol{x}_{1,j}^*\}, \{\boldsymbol{x}_{2,j}^*\}, \{\boldsymbol{x}_{3,j}^*\}, \boldsymbol{z}_1^*, \boldsymbol{z}_2^*, \boldsymbol{z}_3^*)$ is a feasible solution to problem in Eq. (110), we can obtain that,

$$\sum_{j=1}^{N} f_{1,j}(\boldsymbol{x}_{1,j}^*, \boldsymbol{x}_{2,j}^*, \boldsymbol{x}_{3,j}^*) \geq \sum_{j=1}^{N} f_{1,j}(\{\underline{\boldsymbol{x}}_{1,j}\}, \{\underline{\boldsymbol{x}}_{2,j}\}, \{\underline{\boldsymbol{x}}_{3,j}\}). \quad (113)$$

By combining Eq. (113) with Eq. (111), we can obtain that,

$$\sum_{j=1}^{N} f_{1,j}(\{\overline{\boldsymbol{x}}_{1,j}\}, \{\overline{\boldsymbol{x}}_{2,j}\}, \{\overline{\boldsymbol{x}}_{3,j}\}) - \sum_{j=1}^{N} f_{1,j}(\boldsymbol{x}_{1,j}^*, \boldsymbol{x}_{2,j}^*, \boldsymbol{x}_{3,j}^*)$$
$$\leq \sum_{j=1}^{N} f_{1,j}(\{\overline{\boldsymbol{x}}_{1,j}\}, \{\overline{\boldsymbol{x}}_{2,j}\}, \{\overline{\boldsymbol{x}}_{3,j}\}) - \sum_{j=1}^{N} f_{1,j}(\{\underline{\boldsymbol{x}}_{1,j}\}, \{\underline{\boldsymbol{x}}_{2,j}\}, \{\underline{\boldsymbol{x}}_{3,j}\}) \quad (114)$$
$$= \beta(\{\phi_j\}, \{\lambda_l\}).$$

By combining Eq. (114) with Eq. (112), we can obtain that,

$$-\beta(\{\phi_j\}, \{\lambda_l\}) \leq \sum_{j=1}^{N} f_{1,j}(\boldsymbol{x}_{1,j}^*, \boldsymbol{x}_{2,j}^*, \boldsymbol{x}_{3,j}^*) - \sum_{j=1}^{N} f_{1,j}(\{\overline{\boldsymbol{x}}_{1,j}\}, \{\overline{\boldsymbol{x}}_{2,j}\}, \{\overline{\boldsymbol{x}}_{3,j}\}) \leq 0. \quad (115)$$

Based on Eq. (115) and $\beta(\{\phi_j\}, \{\lambda_l\}) \geq 0$, we can get that,

$$|\sum_{j=1}^{N} f_{1,j}(\boldsymbol{x}_{1,j}^*, \boldsymbol{x}_{2,j}^*, \boldsymbol{x}_{3,j}^*) - \sum_{j=1}^{N} f_{1,j}(\{\overline{\boldsymbol{x}}_{1,j}\}, \{\overline{\boldsymbol{x}}_{2,j}\}, \{\overline{\boldsymbol{x}}_{3,j}\})| \leq \beta(\{\phi_j\}, \{\lambda_l\}). \quad (116)$$

By combining Eq. (116) with Eq. (110) and Eq. (111), we can conclude the gap between the optimal objective value by utilizing the exterior penalty method (i.e., $\sum_{j=1}^{N} f_{1,j}(\boldsymbol{x}_{1,j}^*, \boldsymbol{x}_{2,j}^*, \boldsymbol{x}_{3,j}^*)$ in P2) and the optimal objective value in original problem P1 (i.e., $\sum_{j=1}^{N} f_{1,j}(\{\overline{\boldsymbol{x}}_{1,j}\}, \{\overline{\boldsymbol{x}}_{2,j}\}, \{\overline{\boldsymbol{x}}_{3,j}\})$)) is bounded and will decrease with penalty parameter $\phi_j, \lambda_l, \forall j, \forall l$ increased.

## I   TLL WITH PARTIAL ZEROTH ORDER CONSTRAINTS

In this work, TLL with *level-wise* zeroth order constraints is considered, where first order information at *each level* is unavailable. In addition, it is worth mentioning that the proposed framework is versatile and can be adapted to a wide range of TLL problems with partial zeroth order constraints, i.e., grey-box TLL, through slight adjustments. The reason we refer to it as grey-box TLL is that the first order information for some levels in TLL is available, while for others it is not (Huang et al.,

Table 4: Comparisons between the proposed DTZO with the state-of-the-art TLL methods based on the applicability to different TLL problems. ✓ represents that the method can be applied to this TLL problem. The proposed DTZO is versatile and can be adapted to a wide range of TLL problems. We use ZOC as an abbreviation for zeroth order constraints.

|  | Betty | Hypergradient | AFTO | **DTZO** |
|---|---|---|---|---|
| Non-distributed TLL without ZOC | ✓ | ✓ | ✓ | ✓ |
| Distributed TLL without ZOC |  |  | ✓ | ✓ |
| TLL with partial ZOC |  |  |  | ✓ |
| TLL with level-wise ZOC |  |  |  | ✓ |

2024b; Beykal et al., 2020; Astudillo and Frazier, 2021; Bajaj et al., 2018). To further show the superiority of the proposed DTZO, we compare it with the state-of-the-art TLL methods (i.e., Betty (Choe et al., 2023), Hypergradient based method (Sato et al., 2021), and AFTO Jiao et al. (2024)) based on their applicability to TLL problems in Table 4. In DTZO, the zeroth order cut takes center stage, driving the construction of cascaded polynomial approximations without the need for gradients or sub-gradients. Notably, zeroth order cut is not only the backbone of DTZO but also opens the door to tackling grey-box TLL problems, seamlessly handling nested functions that combine both black-box and white-box elements. Discussions are provided as follows.

### I.1 TLL WITH SECOND AND THIRD-LEVEL ZEROTH ORDER CONSTRAINTS

In this situation, the first order information at the first-level in TLL problems is accessible. Thus, we can use the exact gradients to replace the zeroth order gradient estimator, i.e., Eq. (16)-(19) can be replaced by,

$$\boldsymbol{x}_{1,j}^{t+1} = \boldsymbol{x}_{1,j}^t - \eta_{\boldsymbol{x}_1}\left(\nabla_{\boldsymbol{x}_{1,j}} f_{1,j}(\boldsymbol{x}_{1,j}^t, \boldsymbol{x}_{2,j}^t, \boldsymbol{x}_{3,j}^t) + 2\phi_j(\boldsymbol{x}_{1,j}^t - \boldsymbol{z}_1^t)\right), \tag{117}$$

$$\boldsymbol{x}_{2,j}^{t+1} = \boldsymbol{x}_{2,j}^t - \eta_{\boldsymbol{x}_2}\nabla_{\boldsymbol{x}_{2,j}} f_{1,j}(\boldsymbol{x}_{1,j}^t, \boldsymbol{x}_{2,j}^t, \boldsymbol{x}_{3,j}^t) - \eta_{\boldsymbol{x}_2}\nabla_{\boldsymbol{x}_{2,j}} o(\{\boldsymbol{x}_{2,j}^t\}, \{\boldsymbol{x}_{3,j}^t\}, \boldsymbol{z}_1^t, \boldsymbol{z}_2^t, \boldsymbol{z}_3^t), \tag{118}$$

$$\boldsymbol{x}_{3,j}^{t+1} = \boldsymbol{x}_{3,j}^t - \eta_{\boldsymbol{x}_3}\nabla_{\boldsymbol{x}_{3,j}} f_{1,j}(\boldsymbol{x}_{1,j}^t, \boldsymbol{x}_{2,j}^t, \boldsymbol{x}_{3,j}^t) - \eta_{\boldsymbol{x}_3}\nabla_{\boldsymbol{x}_{3,j}} o(\{\boldsymbol{x}_{2,j}^t\}, \{\boldsymbol{x}_{3,j}^t\}, \boldsymbol{z}_1^t, \boldsymbol{z}_2^t, \boldsymbol{z}_3^t). \tag{119}$$

By using the gradient descent steps in Eq. (117)-(119), the TLL problems with second and third-level zeroth order constraints can be effectively by the proposed framework.

### I.2 TLL WITH FIRST AND THIRD-LEVEL ZEROTH ORDER CONSTRAINTS

In this situation, the first order information at the second-level in TLL problems is available. Thus, we can use the first order information to generate outer layer cutting plane, e.g., $\rho$-cut (Jiao et al., 2024). By combining the outer layer first order cutting plane with the inner layer zeroth order cut, the proposed framework is capable of constructing the cascaded polynomial approximation. The generated outer layer $\rho$-cut can be expressed as,

$$\nabla\phi_{\text{out}}(\{\boldsymbol{x}_{2,j}^t\}, \{\boldsymbol{x}_{3,j}^t\}, \boldsymbol{z}_1^t, \boldsymbol{z}_2^t, \boldsymbol{z}_3^t)^\top \left(\begin{bmatrix} \{\boldsymbol{x}_{2,j}\} \\ \{\boldsymbol{x}_{3,j}\} \\ \boldsymbol{z}_1 \\ \boldsymbol{z}_2 \\ \boldsymbol{z}_3 \end{bmatrix} - \begin{bmatrix} \{\boldsymbol{x}_{2,j}^t\} \\ \{\boldsymbol{x}_{3,j}^t\} \\ \boldsymbol{z}_1^t \\ \boldsymbol{z}_2^t \\ \boldsymbol{z}_3^t \end{bmatrix}\right)$$

$$+\phi_{\text{out}}(\{\boldsymbol{x}_{2,j}^t\}, \{\boldsymbol{x}_{3,j}^t\}, \boldsymbol{z}_1^t, \boldsymbol{z}_2^t, \boldsymbol{z}_3^t) \tag{120}$$

$$\leq \varepsilon_{\text{out}} + \rho\left(a_1 + (N+1)(a_2 + a_3) + \sum_{i=2}^3 \sum_{j=1}^N ||\boldsymbol{x}_{i,j}^t||^2 + \sum_{i=1}^3 ||\boldsymbol{z}_i^t||^2\right).$$

In Eq. (120), $\rho > 0$ is a parameter in $\rho$-weakly convex function, and $a_i, i = 1, 2, 3$ is the boundness of variable $\boldsymbol{x}_{i,j}, \boldsymbol{z}_i$, as discussed in Jiao et al. (2024). By using the outer layer first order cutting plane, the TLL problems with first and third-level zeroth order constraints can be addressed by the proposed framework.

### I.3 TLL WITH FIRST AND SECOND-LEVEL ZEROTH ORDER CONSTRAINTS

In this situation, the first order information at the third-level in TLL problems is accessible. Similarly, we can utilize the first order information to generate the inner layer cutting plane, e.g., $\rho$-cut. Through combining the inner layer first order cutting plane with the outer layer zeroth order cut, the proposed framework is capable of constructing the cascaded polynomial approximation. The generated inner layer $\rho$-cut can be expressed as,

$$
\nabla\phi_{\text{in}}(\{\boldsymbol{x}_{3,j}^t\}, \boldsymbol{z}_1^t, \boldsymbol{z}_2^t, \boldsymbol{z}_3^t)^\top \left( \begin{bmatrix} \{\boldsymbol{x}_{3,j}\} \\ \boldsymbol{z}_1 \\ \boldsymbol{z}_2 \\ \boldsymbol{z}_3 \end{bmatrix} - \begin{bmatrix} \{\boldsymbol{x}_{3,j}^t\} \\ \boldsymbol{z}_1^t \\ \boldsymbol{z}_2^t \\ \boldsymbol{z}_3^t \end{bmatrix} \right) + \phi_{\text{in}}(\{\boldsymbol{x}_{3,j}^t\}, \boldsymbol{z}_1^t, \boldsymbol{z}_2^t, \boldsymbol{z}_3^t)
$$

$$
\leq \varepsilon_{\text{in}} + \rho \left( (N+1)a_1 + a_2 + a_3 + \sum_{j=1}^N \|\boldsymbol{x}_{3,j}^t\|^2 + \sum_{i=1}^3 \|\boldsymbol{z}_i^t\|^2 \right).
\tag{121}
$$

By using the inner layer first order cutting plane in Eq. (121), the TLL problems with second and third-level zeroth order constraints can be addressed by the proposed framework.

## J  DISCUSSIONS

### J.1  CUTTING PLANE METHOD

Cutting plane method, also called polyhedral approximation (Bertsekas, 2015), is widely used in convex optimization (Franc et al., 2011; Boyd and Vandenberghe, 2007) and distributed optimization (Bürger et al., 2013; Yang et al., 2014). The rationale behind cutting plane method is to use the intersection of a finite number of half-spaces (e.g., $P = \{x|a_l^T x \leq b_l, l = 1, \cdots, L\}$, where $\{x|a_l^T x \leq b_l\}$ represent a half-space (Boyd and Vandenberghe, 2004)) to approximate the feasible region of the original optimization problem (e.g., $x \in \mathcal{X}$). The approximation can be gradually refined by generating additional half-spaces (Bertsekas, 2015). Recently, cutting plane methods have proven effective in tackling distributed multilevel optimization problems. By leveraging these methods, such problems can be transformed into decomposable optimization problems, which greatly simplifies the design of distributed algorithms for nested optimization, as discussed in (Jiao et al., 2023; 2024). In (Jiao et al., 2023), cutting plane methods are applied to solve bilevel optimization problems within a distributed framework. Likewise, (Chen et al., 2024c) utilize the cutting plane method to tackle distributed bilevel optimization challenges in downlink multi-cell systems. Building on this, (Jiao et al., 2024) further extend the approach to address distributed trilevel optimization problems. However, existing cutting plane methods for multilevel optimization rely on the first-order information to generate cutting planes, which are not available in zeroth-order optimization. In this work, we propose a framework capable of generating zeroth-order cuts for multilevel optimization problems **without** the use of first-order information.

### J.2  THE CHOICE OF GRADIENT ESTIMATOR

It is worth noting that the proposed framework is versatile, allowing for the integration of various gradient estimators. For instance, the mini-batch sampling-based gradient estimator (Liu et al., 2020; Duchi et al., 2015) can be employed to replace the two-point gradient estimator, reducing variance. Specifically, with mini-batch sampling, Eq. (10), (12) (19), (20), and (21) can be replaced by the following multi-point gradient estimators.

$$
G_\mu^{\text{in}}(\{\boldsymbol{x}_{3,j}^t\}, \boldsymbol{z}_1^t, \boldsymbol{z}_2^{t'}, \boldsymbol{z}_3^t)
$$
$$
= \frac{1}{\mu} \sum_{p=1}^b [\phi_{\text{in}}(\{\boldsymbol{x}_{3,j}^t + \mu\boldsymbol{\mu}_{x_{3,j}}^p\}, \boldsymbol{z}_1^t + \mu\boldsymbol{\mu}_{z_1}^p, \boldsymbol{z}_2^{t'} + \mu\boldsymbol{\mu}_{z_2}^p, \boldsymbol{z}_3^t + \mu\boldsymbol{\mu}_{z_3}^p) - \phi_{\text{in}}(\{\boldsymbol{x}_{3,j}^t\}, \boldsymbol{z}_1^t, \boldsymbol{z}_2^{t'}, \boldsymbol{z}_3^t)\boldsymbol{\mu}^{\text{in},p}],
\tag{122}
$$

$$G_\mu^{\text{out}}(\{\boldsymbol{x}_{2,j}^t\}, \{\boldsymbol{x}_{3,j}^t\}, \boldsymbol{z}_1^t, \boldsymbol{z}_2^t, \boldsymbol{z}_3^t)$$

$$= \frac{1}{\mu} \sum_{p=1}^{b} [\phi_{\text{out}}(\{\boldsymbol{x}_{2,j}^t + \mu\boldsymbol{\mu}_{x_{2,j}}^p\}, \{\boldsymbol{x}_{3,j}^t + \mu\boldsymbol{\mu}_{x_{3,j}}^p\}, \boldsymbol{z}_1^t + \mu\boldsymbol{\mu}_{z_1}^p, \boldsymbol{z}_2^t + \mu\boldsymbol{\mu}_{z_2}^p, \boldsymbol{z}_3^t + \mu\boldsymbol{\mu}_{z_3}^p) \quad (123)$$

$$- \phi_{\text{out}}(\{\boldsymbol{x}_{2,j}^t\}, \{\boldsymbol{x}_{3,j}^t\}, \boldsymbol{z}_1^t, \boldsymbol{z}_2^t, \boldsymbol{z}_3^t)\boldsymbol{\mu}^{\text{out},p}],$$

$$G_{\boldsymbol{x}_{1,j}}(\{\boldsymbol{x}_{1,j}^t\}, \{\boldsymbol{x}_{2,j}^t\}, \{\boldsymbol{x}_{3,j}^t\}, \boldsymbol{z}_1^t, \boldsymbol{z}_2^t, \boldsymbol{z}_3^t)$$

$$= \frac{1}{\mu} \sum_{p=1}^{b} [f_{1,j}(\boldsymbol{x}_{1,j}^t + \mu\boldsymbol{u}_{k,1}^p, \boldsymbol{x}_{2,j}^t, \boldsymbol{x}_{3,j}^t) - f_{1,j}(\boldsymbol{x}_{1,j}^t, \boldsymbol{x}_{2,j}^t, \boldsymbol{x}_{3,j}^t)\boldsymbol{u}_{k,1}^p] + 2\phi_j(\boldsymbol{x}_{1,j}^t - \boldsymbol{z}_1^t), \quad (124)$$

$$G_{\boldsymbol{x}_{2,j}}(\{\boldsymbol{x}_{1,j}^t\}, \{\boldsymbol{x}_{2,j}^t\}, \{\boldsymbol{x}_{3,j}^t\}, \boldsymbol{z}_1^t, \boldsymbol{z}_2^t, \boldsymbol{z}_3^t) = \nabla_{\boldsymbol{x}_{2,j}} o(\{\boldsymbol{x}_{2,j}^t\}, \{\boldsymbol{x}_{3,j}^t\}, \boldsymbol{z}_1^t, \boldsymbol{z}_2^t, \boldsymbol{z}_3^t)$$

$$+ \frac{1}{\mu} \sum_{p=1}^{b} [f_{1,j}(\boldsymbol{x}_{1,j}^t, \boldsymbol{x}_{2,j}^t + \mu\boldsymbol{u}_{k,2}^p, \boldsymbol{x}_{3,j}^t) - f_{1,j}(\boldsymbol{x}_{1,j}^t, \boldsymbol{x}_{2,j}^t, \boldsymbol{x}_{3,j}^t)\boldsymbol{u}_{k,2}^p], \quad (125)$$

$$G_{\boldsymbol{x}_{3,j}}(\{\boldsymbol{x}_{1,j}^t\}, \{\boldsymbol{x}_{2,j}^t\}, \{\boldsymbol{x}_{3,j}^t\}, \boldsymbol{z}_1^t, \boldsymbol{z}_2^t, \boldsymbol{z}_3^t) = \nabla_{\boldsymbol{x}_{3,j}} o(\{\boldsymbol{x}_{2,j}^t\}, \{\boldsymbol{x}_{3,j}^t\}, \boldsymbol{z}_1^t, \boldsymbol{z}_2^t, \boldsymbol{z}_3^t)$$

$$+ \frac{1}{\mu} \sum_{p=1}^{b} [f_{1,j}(\boldsymbol{x}_{1,j}^t, \boldsymbol{x}_{2,j}^t, \boldsymbol{x}_{3,j}^t + \mu\boldsymbol{u}_{k,3}^p) - f_{1,j}(\boldsymbol{x}_{1,j}^t, \boldsymbol{x}_{2,j}^t, \boldsymbol{x}_{3,j}^t)\boldsymbol{u}_{k,3}^p], \quad (126)$$

where $\boldsymbol{\mu}^{\text{in},p} = [\{\boldsymbol{\mu}_{x_{3,j}}^p\}, \boldsymbol{\mu}_{z_1}^p, \boldsymbol{\mu}_{z_2}^p, \boldsymbol{\mu}_{z_3}^p]$, $\boldsymbol{\mu}^{\text{out},p} = [\{\boldsymbol{\mu}_{x_{2,j}}^p\}, \{\boldsymbol{\mu}_{x_{3,j}}^p\}, \boldsymbol{\mu}_{z_1}^p, \boldsymbol{\mu}_{z_2}^p, \boldsymbol{\mu}_{z_3}^p]$, $\boldsymbol{u}_{k,1}^p$, $\boldsymbol{u}_{k,2}^p$, $\boldsymbol{u}_{k,3}^p$, $p = 1, \cdots b$ are drawn from $\mathcal{N}(0, \mathbf{I})$, and $b$ represents the number of samples used in the multi-point gradient estimator.

## K   FUTURE WORK

This study is the first work that considers how to address the trilevel zeroth order optimization problems. The proposed framework is not only capable of addressing the single-level and bilevel zeroth order learning problems but can also be applied to a broad class of TLL problems, e.g., TLL with partial zeroth order constraints. However, higher-level nested learning problems, specifically those with more than three levels, are not considered in this work and will be addressed in future research.

