# OpenReview forum: "Unlocking Trilevel Learning with Level-Wise Zeroth Order Constraints: Distributed Algorithms and Provable Non-Asymptotic Convergence"
_ICLR.cc/2025/Conference — Submitted to ICLR 2025_

### Official Review · Reviewer_g4re · 2024-10-30

**Soundness:** 2
**Presentation:** 1
**Contribution:** 2
**Rating:** 3
**Confidence:** 5

**Summary:**

This paper studied the trilevel learning with level-wise zeroth-order constraints in the distributed setting, and proposed a distributed trilevel zeroth-order method. It provided the non-asymptotic convergence analysis of the proposed method. It also provided some experimental results on the proposed method.

**Strengths:**

This paper proposed a distributed trilevel zeroth-order method. It provided the non-asymptotic convergence analysis of the proposed method. It also provided some experimental results on the proposed method.

**Weaknesses:**

This paper is very poorly written. For example, the problem (1) in the paper, the authors do not distinguish between the optimal variables $x^*_1$, $x^*_2$ , $x^*_3$ and the variables $x_1$, $x_2$ , $x_3$. Under this case, the proposed algorithm and the provided convergence results are confusing. They may be wrong.

From algorithm 1 of this paper, the proposed algorithm basically combines the existing algorithms with the existing zeroth-order gradient estimators. There do not exist any novelty in the proposed algorithm. Meanwhile, the convergence analysis mainly follows the exiting results.


Assumption 1 in the convergence analysis is very strict. Under this assumption, the contribution of the provided convergence analysis is very limited. Meanwhile, the convergence results can not provide any useful information to verify the specific properties of the proposed method.

In the numerical experiments, many comparisons are missing. For example, many federated bilevel methods should be compared in the numerical experiments. Why the authors only provided two single-level federated methods.

**Questions:**

Please see the above weaknesses.

---

> ### Author Response · Authors · 2024-11-21
> **Author Rebuttal: Part 1**
>
> **General Reply:** We appreciate your comments. Below please find the reply to your questions. Please let us know if you have any additional questions or concerns—we would be happy to address them.
>
> **(W1)**  This paper is very poorly written. For example, the problem (1) in the paper, the authors do not distinguish between the optimal variables $\boldsymbol{x}_1^∗, \boldsymbol{x}_2^*,\boldsymbol{x}_3^*$, and the variables $\boldsymbol{x}_1, \boldsymbol{x}_2 , \boldsymbol{x}_3$. Under this case, the proposed algorithm and the provided convergence results are confusing. They may be wrong.
>
> **(Reply to W1)** Thank you for your feedback. We appreciate your comments. The nested optimization problem investigated in this paper involves complex mathematical concepts [1, 2], to our best knowledge, this is the **first work** that considers how to address the trilevel zeroth order optimization problem. In the initial version of the manuscript, we consistently used $\boldsymbol{x}_2$ and $\boldsymbol{x}_3$ both within and outside the lower-level optimization problem, as outlined in [1], to enhance the readability. Based on your suggestion, we have revised the notations for trilevel optimization to align with the standard formulation presented in [2]. Moreover, the entire manuscript has undergone professional proofreading to ensure grammatical correctness and fluency. We would appreciate it if the reviewer could provide further suggestions to improve the writing of this manuscript.
>
>
>
> **(W2)** From algorithm 1 of this paper, the proposed algorithm basically combines the existing algorithms with the existing zeroth-order gradient estimators. There do not exist any novelty in the proposed algorithm. Meanwhile, the convergence analysis mainly follows the exiting results.
>
> **(Reply to W2)** Thanks for your comments. Trilevel optimization has found diverse applications in various machine learning tasks, ranging from robust hyperparameter optimization to domain adaptation. Existing approaches to trilevel optimization primarily address scenarios where first-order information is available at each level of the optimization problem. However, the study of trilevel optimization in settings where first-order information is unavailable remains **under-explored**. These scenarios are increasingly $\textit{important}$ given the widespread use of black-box models in machine learning, e.g., in many practical scenarios involving LLMs, access to first-order information is often restricted due to the proprietary nature of these models or API constraints (commercial LLM APIs), which typically only support input-output interactions without providing gradient visibility. Compared to single-level and bilevel optimization problems, trilevel optimization problems are significantly harder to solve [18]; in fact, even finding a feasible solution in a linear trilevel learning problem is **NP-hard** [19]. In this work, we consider how to address the trilevel optimization problems when first-order information is unavailable. To our best knowledge, this is the **first work** to address the trilevel zeroth order optimization problems. Please note that simply combining the existing algorithms **can not** achieve the goal. In the proposed algorithm, a $\textit{novel}$ cutting plane, i.e., zeroth order cut, is proposed, which can construct the cascaded polynomial relaxation without relying on first-order information. We demonstrate that the proposed zeroth order cuts are capable of constructing the **polynomial relaxation** for trilevel zeroth order optimization problems, and the relaxation will be gradually tightened as additional cuts are introduced (It should be noted that simply combining existing trilevel algorithms with zeroth-order gradient estimators **cannot achieve** this theoretical guarantee). To the best of our knowledge, this is the **first algorithm** to explore the generation of cutting planes without utilizing first-order information in multilevel optimization. In addition, we provide **theoretical guarantees** for iteration complexity and communication efficiency for the proposed algorithm, simply combining existing methods **can not** ensure these theoretical guarantees. Moreover, the proposed algorithm is $\textit{general}$ and adaptable to a **wide range** of trilevel optimization scenarios (e.g., trilevel optimization problems with partial zeroth order constraints) with slight adjustments. For instance, if second- or third-level gradients are available, first-order-based cuts [4] can be used to construct a cascaded polynomial approximation. When first-level gradients in TLL are accessible, gradient descent steps can replace the zeroth-order gradient estimators.

---

> ### Author Response · Authors · 2024-11-21
> **Author Rebuttal: Part 2**
>
> Please note that our convergence analysis is **distinct from** that of [2, 3]. Different from the iteration complexity analysis in [3], which is for the centralized zeroth-order single-level optimization algorithm, we provide the $\textit{iteration complexity}$ and $\textit{communication complexity}$ analyses for the proposed **distributed trilevel** zeroth order optimization algorithm. Different from the convergence analysis in [2], the provided convergence analyses are for the **zeroth-order** trilevel optimization algorithm. To our best knowledge, this is the **first study** to provide the iteration complexity and communication complexity analyses for the trilevel zeroth order optimization algorithm.
>
> **(W3)** Assumption 1 in the convergence analysis is very strict. Under this assumption, the contribution of the provided convergence analysis is very limited. Meanwhile, the convergence results can not provide any useful information to verify the specific properties of the proposed method.
>
> **(Reply to W3)** Thanks for your comments. In fact, Assumption 1 is **widely-used** in the theoretical analyses in machine learning. First, the bounded domain assumption is **common-used** in machine learning. For example, this assumption can be found in Assumption 1 in [4], Assumption 5.3 in [5], Assumption 2.3 [6],  Assumption 3 in [7], Assumption A2 in [8], and so on. In addition, the lower-bound assumption is also **common-used** in machine learning. For instance, this assumption appears in Theorem 3.2 in [9], Assumption 1 in [10], Theorem 1 in [11], Assumption 3 in [12], Assumption 5 in [13], Assumption A4 in [14], Assumption A3 in [15], and many other works.
>
> Moreover, the iteration complexity and communication complexity are both meaningful and valuable for the proposed framework. For instance, we can flexibly adjust $T_1$ in the proposed framework to control the trade-off between the performance of cascaded polynomial relaxation and convergence rate (i.e., iteration and communication complexities in Theorems 1 and 2). These theoretical discussions are presented under Theorems 1 and 2 on page 8. Additionally, we have included more experimental results on the convergence rate in the revised manuscript, please refer to page 37 in Appendix F. For your convenience, the added content is provided below.
>
> In addition, the impact of different choices of $T_1$ on the convergence rate within the proposed framework is evaluated. As illustrated in Figures 6 and 7, a smaller $T_1$ leads to faster convergence but affects the method's performance, resulting in a higher test loss. Conversely, if a better performance is required, a larger $T_1$ can be selected, corresponding to a more refined polynomial relaxation. In the proposed framework, we can $\textit{flexibly}$ adjust $T_1$ based on distributed system requirements. The results in Figures 6 and 7 are consistent with our theoretical analyses presented under Theorems 1 and 2.
>
>
>
> **(W4)** In the numerical experiments, many comparisons are missing. For example, many federated bilevel methods should be compared in the numerical experiments. Why the authors only provided two single-level federated methods.
>
> **(Reply to W4)** Thanks for comments. In fact, the **state-of-the-art federated bilevel zeroth order method**, FedRZO$_{bl}$ [16], **is already applied** as a baseline method in this work. The existing (first-order) distributed bilevel optimization method can not be applied in the experiment (zeroth-order setting) because these methods require the first-order information to update the variables.
>
> Moreover, we compare the proposed DTZO with the SOTA (first-order) distributed bilevel and distributed trilevel methods (ADBO [17], AFTO [2]). It is important to note that these methods **cannot be directly applied** to the trilevel zeroth-order optimization problem. To adapt them for our experiments, we combine these methods with the two-point gradient estimator [3]—resulting in variants such as ADBO + ZO and AFTO + ZO. However, ADBO + ZO and AFTO + ZO **do not have any theoretical guarantees** (e.g., convergence guarantees). We include them purely as baseline methods, as recommended by the reviewer. The corresponding results are presented below. We are happy to add any additional comparisons if requested by the reviewer.
>
>
>
> | Dataset | FedZOO         | FedRZO$_{\rm{bl}}$ | ADBO + ZO      | AFTO + ZO      | **DTZO**            |
> | ------- | -------------- | ------------------ | -------------- | -------------- | ---------------------- |
> | MNIST   | 52.89 ± 0.49 % | 54.05 ± 0.81 %     | 53.41 ± 0.56 % | 75.01 ± 0.47 % | **79.27** ± **0.19** % |
> | QMNIST  | 52.45 ± 0.88 % | 54.67 ± 0.65 %     | 54.87 ± 0.49 % | 73.89 ± 0.52 % | **78.04** ± **0.37** % |
> | F-MNIST | 48.74 ± 0.61 % | 50.23 ± 0.49 %     | 51.02 ± 0.67 % | 64.48 ± 0.61 % | **70.07** ± **0.45** % |
> | USPS    | 72.77 ± 0.43 % | 73.79 ± 0.56 %     | 73.23 ± 0.41 % | 79.87 ± 0.65 % | **85.13** ± **0.14** % |

---

> > ### Author Response · Authors · 2024-11-21
> > **Author Rebuttal: Part 3**
> >
> > **Reference**
> >
> > [1] Bilevel Optimization and Applications
> >
> > [2] Provably Convergent Federated Trilevel Learning
> >
> > [3] Stochastic First-and Zeroth-Order Methods for Nonconvex Stochastic Programming
> >
> > [4] A Conditional Gradient-based Method for Simple Bilevel Optimization with Convex Lower-level Problem
> >
> > [5] Non-Convex Bilevel Optimization with Time-Varying Objective Functions
> >
> > [6] AdaDelay: Delay Adaptive Distributed Stochastic Optimization
> >
> > [7] Distributionally Robust Federated Averaging
> >
> > [8] Distributed Zeroth-Order Stochastic Optimization in Time-varying Networks
> >
> > [9] Conflict-Averse Gradient Descent for Multi-task Learning
> >
> > [10] Communication-Efficient Stochastic Zeroth-Order Optimization for Federated Learning
> >
> > [11] Distributed Zero-Order Algorithms for Nonconvex Multi-Agent Optimization
> >
> > [12] On Penalty Methods for Nonconvex Bilevel Optimization and First-Order Stochastic Approximation
> >
> > [13] Enhanced Bilevel Optimization via Bregman Distance
> >
> > [14] A Stochastic Linearized Augmented Lagrangian Method for Decentralized Bilevel Optimization
> >
> > [15] SLM: A Smoothed First-Order Lagrangian Method for Structured Constrained Nonconvex Optimization
> >
> > [16] Zeroth-Order Methods for Nondifferentiable, Nonconvex, and Hierarchical Federated Optimization
> >
> > [17] Asynchronous Distributed Bilevel Optimization
> >
> > [18] The Computational Complexity of Multi-Level Linear Programs
> >
> > [19] A Review on Bilevel Optimization: From Classical to Evolutionary Approaches and Applications

---

> > > ### Comment · Reviewer_g4re · 2024-11-28
> > > **Reply to rebuttals**
> > >
> > > I still concern the contributions of this paper. The first-order bilevel or trilevel optimization has been widely studied recently, the authors only use the zeroth-order gradient estimators instead of the first-order gradients in the existing first-order trilevel algorithms. This paper did not provide any new contributions to the bilevel or trilevel optimization such as new learning framework or relax the existing conditions, and also did not provide any new contributions to the zeroth-order algorithms such as a new zeroth-order gradient estimator. This paper only combining the existing trilevel optimization algorithms with the existing zeroth-order gradient estimators.
> > >
> > > It basically stacks the existing zeroth-order and distributed technologies on top of the existing first-order trilevel optimization.
> > >
> > > Meanwhile, I still think that Assumption 1 in the paper is very strict. Please note that this strict assumption did not be used in many bilevel optimization algorithms such as [1,2]
> > >
> > > [1] On Penalty Methods for Nonconvex Bilevel Optimization and First-Order Stochastic Approximation
> > >
> > > [2] Decentralized bilevel optimization
> > >
> > > Thus, I keep my score.

---

> ### Author Response · Authors · 2024-11-26
> **We are happy to address any further concerns**
>
> We sincerely thank you for raising the concerns in the initial reviews. We have tried our best efforts to clarify those concerns in the responses. Given the limited time for discussion, we would really appreciate it if you could let us know in case there is any additional concern.

---

> ### Author Response · Authors · 2024-11-28
>
> Thank you for your feedback. While we sincerely value your review, we must respectfully disagree with your comments. Below, we provide our detailed responses to your comments.
>
> - We respectfully disagree with the reviewer's statement ``this paper did not provide any new contributions to the bilevel or trilevel optimization such as new learning framework or relax the existing conditions, and also did not provide any new contributions to the zeroth-order algorithms such as a new zeroth-order gradient estimator. This paper only combining the existing trilevel optimization algorithms with the existing zeroth-order gradient estimators.'' Trilevel zeroth order optimization is an **unexplored** problem of significant importance in machine learning, particularly in scenarios where black-box LLMs are utilized. Trilevel zeroth order optimization problems are significantly harder to solve, even finding a **feasible solution** in a linear trilevel optimization problem is **NP-hard**. Existing approaches to trilevel optimization primarily address scenarios where first-order information is available at each level of the optimization problem. In machine learning optimization algorithm design, having non-asymptotic convergence guarantees is crucial, as it provides a clearer understanding of the algorithm's convergence rate. This work presents the **first trilevel zeroth-order optimization algorithm with non-asymptotic convergence guarantees**, e.g., iteration complexity and communication complexity. In addition, we propose a novel zeroth-order cut with **theoretical polynomial relaxation guarantees** and employ cascaded polynomial relaxation to address the trilevel zeroth order optimization problems. This is the **first method** capable of **generating cutting planes without requiring first-order information**, while **theoretically ensuring polynomial relaxation**. It is worth mentioning that the proposed zeroth order cuts are also the **first non-linear cuts** in multilevel optimization. To further highlight the novelty of the proposed zeroth order cut, we compare it with existing cutting plane methods used in multilevel optimization in Table 1. Furthermore, this work **introduces the concept of soft constraints into multilevel optimization** for the first time. Based on the soft constraints, exterior penalty based method is employed for the trilevel zeroth order distributed algorithms. Although the proposed framework used the two-point gradient estimator in Eq. (16-18), this work is **totally different from combining the existing trilevel optimization algorithms with the existing zeroth-order gradient estimators**. A straightforward way to validate the contributions of our work is as follows: we would be delighted to see Reviewer g4re combine existing distributed trilevel optimization algorithms with zeroth-order gradient estimators to develop a distributed trilevel zeroth-order optimization method with non-asymptotic convergence guarantees, including iteration and communication complexity, while also guaranteeing cascaded polynomial relaxation. If this cannot be achieved, **we hope that our work will be evaluated objectively by Reviewer g4re.**
>
> - We respectfully disagree with the reviewer's statement ``I still think that Assumption 1 in the paper is very strict. Please note that this strict assumption did not be used in many bilevel optimization algorithms such as [9, 12]''. First, **this assumption is not strict and is widely used** in theoretical analyses within machine learning research, as demonstrated by the examples provided in our earlier responses. Second, **it has been employed in numerous bilevel optimization studies**. For the bounded domain assumption, it is widely used in many bilevel optimization works, such as **Assumption 2 in [1], Assumption 1 in [5], Assumption 5.3 in [6], Assumption 3 in [7], the assumption in Lemma 1 in [8]**. For the lower bound assumption, it is widely used in many bilevel optimization works, such as **Assumption 3 in [9], Assumption 5 in [10], Assumption A4 in [11]**. We understand that the reviewer may not have been exposed to these works and may therefore have misinterpreted the assumption, **while we also hope this work can be evaluated objectively**.

---

> ### Author Response · Authors · 2024-11-28
>
> Table 1. Comparisons Between Existing Cutting Plane Methods
>
> | Cutting plane Method                 | Convex              | Non-convex                       | Non-linear                    | Gradient free        |
> | ----------------------------- | ------------------- | -------------------------------- | ----------------------------- | -------------------- |
> | [1]             | $ \quad \checkmark$ |             |                    |                      |
> | [2]              | $ \quad \checkmark$ |           |                      |                      |
> | [3]              | $ \quad \checkmark$ |                    |                      |                      |
> | [4]              | $ \quad \checkmark$ | $ \\; \\; \\; \quad \checkmark$  |                               |                      |
> | The proposed Zeroth Order Cut | $ \quad \checkmark$ | $ \\; \\; \\;  \quad \checkmark$ | $  \\; \\;  \quad \checkmark$ | $ \qquad \checkmark$ |
>
> **Reference**
>
> [1] Asynchronous Distributed Bilevel Optimization
>
> [2] Robust Beamforming for Downlink Multi-Cell Systems: A Bilevel Optimization Perspective
>
> [3] Tri-Level Navigator: LLM-Empowered Tri-Level Learning for Time Series OOD Generalization
>
> [4] Provably Convergent Federated Trilevel Learning
>
> [5] A Conditional Gradient-based Method for Simple Bilevel Optimization with Convex Lower-level Problem
>
> [6] Non-Convex Bilevel Optimization with Time-Varying Objective Functions
>
> [7] Distributionally Robust Federated Averaging
>
> [8] Robust Optimization over Multiple Domains
>
> [9] On Penalty Methods for Nonconvex Bilevel Optimization and First-Order Stochastic Approximation
>
> [10] Enhanced Bilevel Optimization via Bregman Distance
>
> [11] A Stochastic Linearized Augmented Lagrangian Method for Decentralized Bilevel Optimization
>
> [12] Decentralized bilevel optimization

---

### Official Review · Reviewer_LaXJ · 2024-10-30

**Soundness:** 3
**Presentation:** 2
**Contribution:** 3
**Rating:** 6
**Confidence:** 3

**Summary:**

This paper proposes DTZO, a novel framework that utilizes zeroth order constraints to solve distributed trilevel learning problems. Unlike existing approaches that rely on first-order information for trilevel learning or previous single-level and bilevel zeroth-order methods, DTZO leverages zeroth-order optimization to address the unique challenges of trilevel structures. Experimental results demonstrate the superior performance of DTZO, establishing it as an effective alternative for trilevel learning problems.

**Strengths:**

1. DTZO provides a novel perspective for solving distributed trilevel learning problems by incorporating zeroth-order constraints, distinguishing it from traditional first-order-based trilevel learning approaches.
2. The paper provides a rigorous theoretical analysis of DTZO, specifically addressing its non-asymptotic convergence rate, which strengthens the framework’s validity and offers insights into its efficiency.

**Weaknesses:**

1. While the paper includes a theoretical analysis of DTZO’s non-asymptotic convergence rate, there are no experimental results to analyze this, such as a convergence analysis based on testing loss. Including such results would help validate the theoretical findings empirically.
2. In the experimental section, the paper employs black-box large language models (LLMs) to achieve DTZO. Although the introduction section mentions the importance of zeroth-order constraints when first-order information is unavailable in black-box LLMs, further elaboration on scenarios where first-order information is inaccessible would enhance readers’ understanding of the choice to use black-box LLMs in these experiments.
3. The paper lacks detailed hyperparameter analysis results in the main body, specifically on the effect of parameters like the iteration interval $\mathcal{T}$. Providing these insights would give readers a more comprehensive understanding of DTZO’s sensitivity to hyperparameter settings.

**Questions:**

1. While the paper presents a theoretical analysis of DTZO’s non-asymptotic convergence rate, could additional experimental results, such as a convergence analysis based on testing loss, be included to empirically validate these theoretical findings?
2. In the experimental section, black-box large language models (LLMs) are used to implement DTZO. Could the authors further elaborate on practical scenarios where first-order information is inaccessible, to clarify the rationale behind using black-box LLMs for these experiments?
3. Could the paper provide a more detailed analysis of hyperparameters in the main body, particularly regarding the impact of parameters like the iteration interval $\mathcal{T}$, to offer readers a better understanding of DTZO’s sensitivity to these settings?

---

> ### Author Response · Authors · 2024-11-21
> **Author Rebuttal: Part 1**
>
> **General Reply:** Thanks for your insightful and valuable suggestions. We have provided a point by point reply to your questions as follows. Please let us know if you have any additional questions or concerns—we would be delighted to address them.
>
> **(W1 & Q1)** While the paper includes a theoretical analysis of DTZO’s non-asymptotic convergence rate, there are no experimental results to analyze this, such as a convergence analysis based on testing loss. Including such results would help validate the theoretical findings empirically.
>
> **(Reply to W1 & Q1)** We sincerely appreciate your suggestions. Per your suggestion, experimental results about the convergence analysis based on testing loss are added in Appendix F (page 37) in the revised manuscript. For your convenience, the added contents are attached as follows.
>
> Following [1], the robustness in the proposed framework with respect to the choice of smoothing parameter $\mu$ is evaluated. The experiments are conducted on the robust hyperparameter optimization task under various setting of smoothing parameter, $\mu \in \\{0.01, 0.001, 0.0001\\}$. It is seen from Figure 4 and 5 that the proposed DTZO is robust to the choice of smoothing parameter $\mu$. In addition, we also note that the proposed DTZO has faster convergence rate with a relatively smaller $\mu$, because the gradient estimate improves when $\mu$ becomes relatively smaller, as discussed in [2].
>
> In addition, the impact of different choices of $T_1$ on the convergence rate within the proposed framework is evaluated. As illustrated in Figures 6 and 7, a smaller $T_1$ leads to faster convergence but affects the method's performance, resulting in a higher test loss. Conversely, if a better performance is required, a larger $T_1$ can be selected, corresponding to a more refined polynomial relaxation. In the proposed framework, we can $\textit{flexibly}$ adjust $T_1$ based on distributed system requirements. The results in Figures 6 and 7 are consistent with our theoretical analyses presented under Theorems 1 and 2.
>
>
>
> **(W2 & Q2)** In the experimental section, the paper employs black-box large language models (LLMs) to achieve DTZO. Although the introduction section mentions the importance of zeroth-order constraints when first-order information is unavailable in black-box LLMs, further elaboration on scenarios where first-order information is inaccessible would enhance readers’ understanding of the choice to use black-box LLMs in these experiments.
>
> **(Reply to W2 & Q2)** Thanks for your valuable suggestions. We have added more discussions in the experiment section as suggested. For your convenience, the added contents are attached as follows.
>
> In many practical scenarios involving LLMs, access to first-order information is restricted due to the proprietary nature of these models or API constraints. For instance, commercial LLM APIs only allow input-output interactions and do not provide visibility into gradients. These limitations make zeroth-order optimization particularly relevant and necessary.

---

> > ### Author Response · Authors · 2024-11-21
> > **Author Rebuttal: Part 2**
> >
> > **(W3 & Q3)**  The paper lacks detailed hyperparameter analysis results in the main body, specifically on the effect of parameters like the iteration interval $\mathcal{T}$. Providing these insights would give readers a more comprehensive understanding of DTZO’s sensitivity to hyperparameter settings.
> >
> > **(Reply to W3 & Q3)** We appreciate your insightful and helpful suggestions. Based on your suggestion, more hyperparameter analyses are provided, include the influence of smoothing parameter $\mu$,  the interval $\mathcal{T}$, and the parameter $T_1$. For your convenience, the added contents are attached as follows.
> >
> > Following [1], the robustness in the proposed framework with respect to the choice of smoothing parameter $\mu$ is evaluated. The experiments are conducted on the robust hyperparameter optimization task under various setting of smoothing parameter, $\mu \in \\{0.01, 0.001, 0.0001\\}$. It is seen from Figure 4 and 5 that the proposed DTZO is robust to the choice of smoothing parameter $\mu$. In addition, we also note that the proposed DTZO has faster convergence rate with a relatively smaller $\mu$, because the gradient estimate improves when $\mu$ becomes relatively smaller, as discussed in [2].
> >
> > In addition, the impact of different choices of $T_1$ on the convergence rate within the proposed framework is evaluated. As illustrated in Figures 6 and 7, a smaller $T_1$ leads to faster convergence but affects the method's performance, resulting in a higher test loss. Conversely, if a better performance is required, a larger $T_1$ can be selected, corresponding to a more refined polynomial relaxation. In the proposed framework, we can $\textit{flexibly}$ adjust $T_1$ based on distributed system requirements. The results in Figures 6 and 7 are consistent with our theoretical analyses presented under Theorems 1 and 2.
> >
> > Additionally, we have also tested the impact of various $\mathcal{T}$ on the performance. The experiments are conducted on robust hyperparameter optimization method on USPS dataset. The results, presented in the table below, indicate that the proposed framework is not highly sensitive to $\mathcal{T}$. However, in distributed systems with sufficient communication capabilities, a smaller $\mathcal{T}$ can be selected, leading to a better cascaded polynomial approximation as more cutting planes are generated.
> >
> > |                  | Performance of DTZO |
> > | ---------------- | ------------------- |
> > | $\mathcal{T}=2$  | 85.54               |
> > | $\mathcal{T}=5$  | 85.12               |
> > | $\mathcal{T}=10$ | 84.88               |
> >
> >
> > **Reference**
> >
> > [1] Zeroth-order methods for nondifferentiable, nonconvex, and hierarchical federated optimization
> >
> > [2] A primer on zeroth-order optimization in signal processing and machine learning: Principals, recent advances, and applications

---

> ### Author Response · Authors · 2024-11-26
> **We are happy to address any further concerns**
>
> We sincerely thank you for raising the concerns in the initial reviews. We have tried our best efforts to clarify those concerns in the responses. Given the limited time for discussion, we would really appreciate it if you could let us know in case there is any additional concern.

---

> ### Comment · Reviewer_LaXJ · 2024-11-27
>
> Thank you for your response. From the experimental results on the impact of various interval $\mathcal{T}$, it seems like there is a linear relationship between the $\mathcal{T}$ and the performance of DTZO. Are there any supporting materials or explanations for these results? Another question is about the smoothing parameter $\mu$, in Fig.4 and Fig.5, it looks like the testing loss would converge to the same point, does it make sense? Since in Eq.29, the $T(\epsilon)$ is related to $F_{\mu}$.

---

> ### Author Response · Authors · 2024-11-27
> **We sincerely appreciate your insightful comments**
>
> We sincerely appreciate your insightful comments and are happy to know that you are interested in our experimental results.
>
>
>
> - **(Q1)**  From the experimental results on the impact of various interval $\mathcal{T}$, it seems like there is a linear relationship between the $\mathcal{T}$ and the performance of DTZO. Are there any supporting materials or explanations for these results?
>
>
>
> - **(Reply to Q1)** We appreciate your comments. We agree with the reviewer that there is a relationship between the $\mathcal{T}$ and the  performance of DTZO.  Specifically, in the proposed framework, $\lfloor \frac{T_1}{\mathcal{T}} \rfloor$ zeroth-order cuts will be generated, and a smaller $\mathcal{T}$ generally leads to relatively improved cascaded polynomial performance since more zeroth-order cuts are generated (as shown in Propositions 1 and 2), which aligns with the experimental results. Moreover, the communication complexity is also related to the setting of $\mathcal{T}$. A higher communication complexity is required with a smaller $\mathcal{T}$, as shown in Theorem 2. In the proposed framework, we can $flexibly$ adjust the  setting of $\mathcal{T}$ based on communication capabilities of the distributed system, for example, if the distributed system has sufficient communication capabilities, a smaller  $\mathcal{T}$ can be chosen. However, in the proposed framework, we do not observe a strictly linear relationship between $\mathcal{T}$ and the performance of DTZO.
>
>
>
> - **(Q2)**  Another question is about the smoothing parameter $\mu$, in Fig.4 and Fig.5, it looks like the testing loss would converge to the same point, does it make sense? Since in Eq.29, the $T(\epsilon)$ is related to $F_{\mu}$.
>
>
>
> - **(Reply to Q2)** Thanks for your valuable comments. You are right, $T(\epsilon)$ is related to $F_{\mu}$ in Eq. (29). However, $T(\epsilon)$ represents the **number of iterations** required for the algorithm to converge, as defined in Definition 2. In the proposed framework, different settings of the smoothing parameter $\mu$ will influence the number of iterations needed for convergence, as $\mu$ influences the variance of gradient estimation [1, 2], i.e., we have that $||\nabla F_{\mu}(w)-\nabla F(w)||^2 \le \mu^2 L^2 (d+3)^{3}/4$. It is similar to the mini-batch SGD, if the batch-size is increased, fewer iterations are needed for convergence because the variance of gradient estimation is reduced, with the distinction that SGD is an unbiased estimator of GD. It can be seen from Fig. 4 and Fig. 5 that the proposed algorithm requires relatively more iterations to converge when a relatively larger value of $\mu$ is used. In addition, the proposed algorithm converges to **different solutions**  based on different settings of the smoothing parameter $\mu$. There are slight differences between the converged points of testing loss, which are not very significant since 1) the proposed framework is robust to the setting of $\mu$, 2) relatively small smoothing parameters are used which will not introduce relatively substantial variance [2].
>
> Thank you once again. We hope our responses have adequately addressed your questions.
>
> **Reference**
>
> [1] Stochastic first-and zeroth-order methods for nonconvex stochastic programming
>
> [2] A primer on zeroth-order optimization in signal processing and machine learning: Principals, recent advances, and applications

---

### Official Review · Reviewer_Pih8 · 2024-11-01

**Soundness:** 2
**Presentation:** 2
**Contribution:** 2
**Rating:** 5
**Confidence:** 4

**Summary:**

This paper studies distributed trilevel optimization problem with zeroth-order constraints. It develops an algorithm based on the cutting plane method and provides its convergence rate. Moreover, the experiment validates its performance.

**Strengths:**

1. The trilevel optimization problem is interesting, which is less studied.

2. This paper proposes a feasible approach to solve the trilevel optimization problem.

**Weaknesses:**

1. The proposed method is based on the cutting plane method. However, no background knowledge about cutting plane is provided. I suggest the author to provide some fundamental information about it. Otherwise, it is difficult for the reviewer to follow the proposed method.

2. The proposed method converts a trilevel problem to an unconstrained problem. However, the solution landscape could be different. How does the approximated solution differ from the original solution? It's better to provide some theoretical analysis for this difference.

3. For the cutting plane method, how to determine $a$, $b$, $c$, and $d$? How do they affect the convergence rate?

4. For the penalty coefficient $\lambda$, how does it affect the convergence?

5. In the experiment, $a$, $b$, $c$, $d$, and $\lambda$ are not stated clearly. What values are used in experiments?

**Questions:**

See Weaknesses

---

> ### Author Response · Authors · 2024-11-21
> **Author Rebuttal: Part 1**
>
> **General Reply:** Thank you for your insightful and valuable suggestions. Below please find our point by point reply to the questions you have raised. Please let us know if you have any additional questions or concerns—we would be delighted to address them.
>
> **(W1)** The proposed method is based on the cutting plane method. However, no background knowledge about cutting plane is provided. I suggest the author to provide some fundamental information about it. Otherwise, it is difficult for the reviewer to follow the proposed method.
>
> **(Reply to W1)** Thanks for your helpful suggestions. We have included additional background information on the cutting plane method in Appendix J1 on page 43 of the revised manuscript. For your convenience, the added content is provided below.
>
> Cutting plane method, also called polyhedral approximation [1], is widely used in convex optimization [2, 3] and distributed optimization [4, 5]. The rationale behind cutting plane method is to use the intersection of a finite number of half-spaces (e.g., $P=\\{x|a_l^Tx\le b_l, l=1,\cdots,L\\}$, where $\\{x|a_l^Tx\le b_l\\}$ represent a half-space [6]) to approximate the feasible region of the original optimization problem (e.g., $x \in \mathcal{X}$) . The approximation can be gradually refined by generating additional half-spaces [1]. Recently, cutting plane methods have proven effective in tackling distributed multilevel optimization problems. By leveraging these methods, such problems can be transformed into decomposable optimization problems, which greatly simplifies the design of distributed algorithms for nested optimization, as discussed in [7, 9].  In [7], cutting plane methods are applied to solve bilevel optimization problems within a distributed framework. Likewise, [8] utilize the cutting plane method to tackle distributed bilevel optimization challenges in downlink multi-cell systems. Building on this, [9] further extend the approach to address distributed trilevel optimization problems. However, existing cutting plane methods for multilevel optimization rely on the first-order information to generate cutting planes, which are not available in zeroth-order optimization. In this work, we propose a framework capable of generating zeroth-order cuts for multilevel optimization problems **without** the use of first-order information.
>
> **(W2)** The proposed method converts a trilevel problem to an unconstrained problem. However, the solution landscape could be different. How does the approximated solution differ from the original solution? It's better to provide some theoretical analysis for this difference.
>
> **(Reply to W2)** We appreciate your insightful comments. Existing approaches to trilevel optimization primarily address scenarios where first-order information is available at each level of the optimization problem. However, the study of trilevel optimization in settings where first-order information is unavailable remains **under-explored**. These scenarios are increasingly important given the widespread use of black-box models in machine learning, e.g., in many practical scenarios involving LLMs, access to first-order information is often restricted due to the proprietary nature of these models or API constraints (commercial LLM APIs), which typically only support input-output interactions without providing gradient visibility. To our best knowledge, this is the **first work** to address the trilevel zeroth order optimization problems. Compared to single-level and bilevel optimization problems, trilevel optimization problems are significantly harder to solve, as discussed in [12, 13]. Finding a $\textit{feasible solution}$ in trilevel optimization problem is **NP-hard** [14], thus it's somehow $\textit{unlikely}$ to design a polynomial-time algorithm for original trilevel optimization problem. In addition, the lower-level optimization problem often serves as a **soft constraint** to upper-level optimization problem, which means that this constraint can be violated to a certain extent while still yielding a feasible and meaningful solution [10, 11], making the relaxation of trilevel optimization problem acceptable. We theoretically demonstrated that the proposed zeroth order cuts can construct the relaxation for the feasible region of the original optimization problem and this relaxation will be **gradually tightened** with zeroth order cuts added. Furthermore, we have also theoretically demonstrated that the gap between the unconstrained optimization problem and the original optimization problem decreases as larger penalty parameters $\phi_j, \lambda_l$ are set, please refer to Appendix H for details. For your convenience, the key steps of the demonstration are outlined below.

---

> > ### Author Response · Authors · 2024-11-21
> > **Author Rebuttal: Part 2**
> >
> > Let $(\\{{\boldsymbol{x}\_{1,j}^*}\\}, \\{{\boldsymbol{x}\_{2,j}^*}\\}, \\{{\boldsymbol{x}\_{3,j}^*}\\}, \boldsymbol{z}_1^*, \boldsymbol{z}_2^*, \boldsymbol{z}_3^*)$ and $(\\{{\overline{\boldsymbol{x}}\_{1,j}}\\}, \\{{\overline{\boldsymbol{x}}\_{2,j}}\\}, \\{{\overline{\boldsymbol{x}}\_{3,j}}\\}, \overline{\boldsymbol{z}}_1, \overline{\boldsymbol{z}}_2, \overline{\boldsymbol{z}}_3)$ respectively denote the optimal solutions to the unconstrained optimization problem and the original constrained optimization problem, we can first establish that:
> >
> > $  \sum\limits \_{j = 1}^N  {f \_{1,j}}({\boldsymbol{x} \_{1,j}^*},{\boldsymbol{x} \_{2,j}^*},{\boldsymbol{x} \_{3,j}^*}) -\sum\limits \_{j = 1}^N {f \_{1,j}}(\{{\overline{\boldsymbol{x}} \_{1,j}}\}, \{{\overline{\boldsymbol{x}} \_{2,j}}\}, \{{\overline{\boldsymbol{x}} \_{3,j}}\})  \\ \le 0.  \qquad \qquad  (1)$
> >
> > We can subsequently derive that,
> >
> > $\sum\limits \_{j = 1}^N {f \_{1,j}}(\{{\overline{\boldsymbol{x}} \_{1,j}}\}, \{{\overline{\boldsymbol{x}} \_{2,j}}\}, \{{\overline{\boldsymbol{x}} \_{3,j}}\}) - \sum\limits \_{j = 1}^N  {f \_{1,j}}({\boldsymbol{x} \_{1,j}^*},{\boldsymbol{x} \_{2,j}^*},{\boldsymbol{x} \_{3,j}^*}) \\ \le \beta(\{\phi_j\},\{\lambda_l\}), \qquad \qquad (2)$
> >
> > where $\beta(\{\phi_j\},\{\lambda_l\})$ will decrease when penalty parameters $\phi_j,\lambda_l$ increase. Combining Eq. (1) with Eq. (2), we can obtain that,
> >
> > $| \sum\limits \_{j = 1}^N  {f \_{1,j}}({\boldsymbol{x} \_{1,j}^*},{\boldsymbol{x} \_{2,j}^*},{\boldsymbol{x} \_{3,j}^*}) -\sum\limits \_{j = 1}^N {f \_{1,j}}(\{{\overline{\boldsymbol{x}} \_{1,j}}\}, \{{\overline{\boldsymbol{x}} \_{2,j}}\}, \{{\overline{\boldsymbol{x}} \_{3,j}}\}) | \le \beta(\{\phi_j\},\{\lambda_l\}).\qquad \qquad(3)$
> >
> > It is seen from Eq. (3) that the gap between the optimal objective values in unconstrained optimization problem and original constrained optimization problem will decrease as the penalty parameters $\phi_j,\lambda_l$ increase.
> >
> > **(W3)** For the cutting plane method, how to determine a, b, c, and d? How do they affect the convergence rate?
> >
> > **(Reply to W3)** Thanks for your valuable comments. In the proposed framework,  ${{a} \_{j,l}^{{in}}}$, ${{b} \_{j,l}^{{in}}}$, ${c} \_{i,l}^{{in}}$, ${d} \_{i,l}^{{in}}  $, $e \_{l}^{{in}}$ and  ${{a} \_{i,j,l}^{{out}}}$, ${{b} \_{i,j,l}^{{out}}}$, ${c} \_{i,l}^{{out}}$, ${d} \_{i,l}^{{out}}  $, $e \_{l}^{{out}}$  refer to the parameters for the $l^{th}$ inner-layer and outer-layer cutting planes. In the proposed framework, the inner-layer and outer-layer cutting planes are generated during the iteration to gradually refine the polynomial relaxation. These parameters are determined according to Eq. (9) and Eq. (11). For example, we provide the mathematical forms of ${{a} \_{j,l}^{{in}}}$, ${{b} \_{j,l}^{{in}}}$, ${c} \_{i,l}^{{in}}$, ${d} \_{i,l}^{{in}}  $, $e \_{l}^{{in}}$ as follows.
> >
> > $a^{in} \_{j,l}=-\frac{L+1}{2},\forall j$
> >
> > $b^{in} \_{j,l}={G_\mu^{in} }( \{{{x} \_{3,j}^t}\},{z}_1^t, {{z}_2^t}', {z}_3^t)+(L+1){{x} \_{3,j}^t},\forall j$
> >
> > $c^{in} \_{i,l}=-\frac{L+1}{2},\forall i$
> >
> > $d^{in} \_{1,l}={G_\mu^{in} }( \{{{x} \_{3,j}^t}\},{z}_1^t, {{z}_2^t}', {z}_3^t)+(L+1)z_1^t$
> >
> > $d^{in} \_{2,l}={G_\mu^{in} }( \{{{x} \_{3,j}^t}\},{z}_1^t, {{z}_2^t}', {z}_3^t)+(L+1){z_2^t}'$
> >
> > $d^{in} \_{3,l}={G_\mu^{in} }( \{{{x} \_{3,j}^t}\},{z}_1^t, {{z}_2^t}', {z}_3^t)+(L+1){z_3^t}$
> >
> > $e^{in}\_{l}=\phi \_{\rm{in}}( \{{{x} \_{3,j}^t}\},{z}_1^t, {{z}_2^t}', {z}_3^t)-{G\_{\mu}^{in} }( \{{{x} \_{3,j}^t}\},{z}_1^t, {{z}_2^t}', {z}\_3^t)^T[\\{{x} \_{3,j}^t\\};z_1^t;{z\_2^t}';z\_3^t]-\frac{L+1}{2}(\sum||x \_{3,j}^t||^2-||z_1^t||^2-||{z\_2^t}'||^2-||z\_3^t||^2)-\frac{\mu^2 }{8}L^2{({d_1}+  {d_2} + (N+1){d_3} + 3)^{3}}$
> >
> > where $( \\{{{x} \_{3,j}^t}\\},{z}_1^t, {{z}_2^t}', {z}_3^t)$ is the point at $t^{th}$ iteration.

---

> > > ### Author Response · Authors · 2024-11-21
> > > **Author Rebuttal: Part 3**
> > >
> > > This work is the $\textit{first}$ to explore generating cutting planes in multilevel optimization without relying on first-order information. We demonstrate that the proposed zeroth-order cuts can construct the **theoretical polynomial relaxation** for the original feasible region, please refer to Propositions 1 and 2. In the proposed framework, the convergence rate is influenced by the number of cutting planes, whereas the cutting planes  parameters  ${{a} \_{j,l}^{{in}}}$, ${{b} \_{j,l}^{{in}}}$, ${c} \_{i,l}^{{in}}$, ${d} \_{i,l}^{{in}}  $, $e \_{l}^{{in}}$ and  ${{a} \_{i,j,l}^{{out}}}$, ${{b} \_{i,j,l}^{{out}}}$, ${c} \_{i,l}^{{out}}$, ${d} \_{i,l}^{{out}}  $, $e \_{l}^{{out}}$   $\textit{do not}$ affect the convergence rate. Specifically, incorporating more cutting planes (corresponds to a larger $T_1$) leads to a higher-quality cascaded polynomial approximation but also incurs higher iteration and communication complexity (i.e., a slower convergence rate), which are shown in Theorem 1, 2. In the proposed framework, we can flexibly adjust $T_1$ based on the capabilities of the distributed system. For example, if the distributed system has limited computational and communication capabilities, a smaller value of $T_1$ can be selected. Conversely, if a higher quality of cascaded polynomial approximation is desired, a larger value of $T_1$ can be chosen. In addition, to evaluate the impact of different choices of $T_1$ on the convergence rate within the proposed framework, some new experimental results are provided in the revised manuscript, the added contents are attached as follows for your convenience.
> > >
> > > In addition, the impact of different choices of $T_1$ on the convergence rate within the proposed framework is evaluated. As illustrated in Figures 6 and 7, a smaller $T_1$ leads to faster convergence but affects the method's performance, resulting in a higher test loss. Conversely, if a better performance is required, a larger $T_1$ can be selected, corresponding to a more refined polynomial relaxation. In the proposed framework, we can $\textit{flexibly}$ adjust $T_1$ based on distributed system requirements. The results in Figures 6 and 7 are consistent with our theoretical analyses presented under Theorems 1 and 2.
> > >
> > > **(W4)**  For the penalty coefficient λ, how does it affect the convergence?
> > >
> > > **(Reply to W4)**  We appreciate your insightful comments. In the proposed framework, $\lambda_l$ will not affect the convergence rate (as shown in Theorems 1 and 2). However, the penalty parameters $\lambda_l,\phi_j$ control the gap between original constrained problem and the unconstrained optimization problem. We have demonstrated that the gap between the unconstrained optimization problem and original constrained optimization problem will decrease as $\lambda_l,\phi_j$ increase, the key steps of the demonstration are shown in the Reply to W2, and more details can be found in Appendix H.
> > >
> > > **(W5)** In the experiment, a, b, c, d, and λ are not stated clearly. What values are used in experiments?
> > >
> > > **(Reply to W5)** We truly appreciate your suggestions. We have added the details of the experiments in Appendix F (Table 3 in page 36) in the revised manuscript. For your convenience, the added contents are attached as follows.
> > >
> > > | Dataset | $\eta_{x_1}$ | $\eta_{x_2}$ | $\eta_{x_3}$ | $\mu$ | $\lambda_l$ | $\phi_j$ |
> > > | ------- | ------------ | ------------ | ------------ | ----- | ----------- | -------- |
> > > | SST-2   | 0.01         | 0.001        | 0.001        | 0.001 | 1           | 0.5      |
> > > | COLA    | 0.01         | 0.001        | 0.001        | 0.001 | 1           | 0.5      |
> > > | MRPC    | 0.01         | 0.001        | 0.001        | 0.001 | 1           | 0.5      |
> > > | MNIST   | 0.01         | 0.05         | 0.1          | 0.001 | 1           | 0.5      |
> > > | QMNIST  | 0.01         | 0.05         | 0.1          | 0.001 | 1           | 0.5      |
> > > | F-MNIST | 0.01         | 0.05         | 0.1          | 0.001 | 1           | 0.5      |
> > > | USPS    | 0.01         | 0.5          | 0.1          | 0.001 | 1           | 0.5      |

---

> > > > ### Author Response · Authors · 2024-11-21
> > > > **Author Rebuttal: Part 4**
> > > >
> > > > **Reference**
> > > >
> > > > [1] Bertsekas, Dimitri. Convex optimization algorithms. Athena Scientific, 2015.
> > > >
> > > > [2] Cutting-plane methods in machine learning
> > > >
> > > > [3] Localization and cutting-plane methods
> > > >
> > > > [4] A polyhedral approximation framework for convex and robust distributed optimization
> > > >
> > > > [5] Distributed robust optimization (DRO), part I: Framework and example
> > > >
> > > > [6] Boyd, Stephen, and Lieven Vandenberghe. Convex optimization. Cambridge university press, 2004.
> > > >
> > > > [7] Asynchronous distributed bilevel optimization
> > > >
> > > > [8] Robust beamforming for downlink multi-cell systems: A bilevel optimization perspective
> > > >
> > > > [9] Provably convergent federated trilevel learning
> > > >
> > > > [10] A general stochastic approach to solving problems with hard and soft constraints.
> > > >
> > > > [11] Using hard constraints for representing soft constraints
> > > >
> > > > [12] The computational complexity of multi-level linear programs
> > > >
> > > > [13] Mixed-integer multi-level optimization through multi-parametric programming
> > > >
> > > > [14] A review on bilevel optimization: From classical to evolutionary approaches and applications

---

> ### Author Response · Authors · 2024-11-26
> **We are happy to address any further concerns**
>
> We sincerely thank you for raising the concerns in the initial reviews. We have tried our best efforts to clarify those concerns in the responses. Given the limited time for discussion, we would really appreciate it if you could let us know in case there is any additional concern.

---

### Official Review · Reviewer_C3cz · 2024-11-03

**Soundness:** 2
**Presentation:** 3
**Contribution:** 2
**Rating:** 6
**Confidence:** 2

**Summary:**

In this paper, the authors propose a distributed trilevel zeroth-order (DTZO) algorithm to address the trilevel optimization problem (TLL) when gradients or subgradients of level objectives (functions) are not available. They first reformulate the inner and outer layer constraints as feasible regions approximated by polynomials which are constructed using zeroth order cuts. These  zero order cuts, corresponding polynomial approximations and local variables are updated at each iteration of the algorithm. The method is implemented in the distributed manner. It is supported by a theoretical analysis of iteration and communication complexities, and experimental evaluation in the tasks related to large language models and robust hyperparameter optimization.

**Strengths:**

DTZO is the first algorithm designed to solve trilevel optimization (TLL) problems in distributed settings without requiring gradients or subgradients of level objectives (functions). The authors provide a theoretical analysis of the algorithm’s iteration and communication complexities. The method shows good performance in comparison to the chosen baselines​.

**Weaknesses:**

My major concern is related to the novelty of the DTZO algorithm as it combines the ideas from two previous papers. Indeed, the general structure of the algorithm seems to be borrowed from the recent paper (Jiao et al., 2024). The main difference is that the authors do not use gradients of functions, which are unavailable, but well-known approximations to them, following (S. Ghadimi and G. Lan, 2013). However, this modification does not seem very significant for me.


Moreover, I have concerns regarding the theoretical results. To prove the main of them, i.e., Theorem 1,2, the authors introduce $\mu$-smooth approximation of the final objective functional $F$. Then, similar to (Jiao et al., 2024), the authors conduct the convergence rate analysis using this approximation. However, there is no theoretical result showing how close the solutions of this $F_{\mu}$ approximation are to those of the objective functional $F$? Without such a result, the convergence rate for $F$ seems to be still an open question. I will appreciate it if the authors provide some comments on this topic.

My minor comments are related to the comparison with the baselines in the numerical experiments. The authors tested the approach in the tasks related to LLM and hyperparameters robust optimization and compare its performance with several baseline models, e.g., FEDZOO (Fang et al., 2022) and  FEDRZO$_{bl}$ (Qiu et al., 2023). Both of these algorithms provide the results for MSE or Loss vs iteration number and communication round, it would be great if the authors provide this kind of result too.

*Minor*:
line 328: equation number (16) is missing

**Questions:**

- Can you comment on the connection between the solutions of $F$ and its $\mu$-smooth approximation $F_{\mu}$? Are they $\epsilon$-close to each other?
- I noticed that the zeroth-order gradient estimation method used here is based on (Ghadimi and Lan, 2013). Could you explain why you choose this method over the approach in (Liu et al., 2020), which utilizes mini-batch sampling to reduce variance?

**References.**
1. Jiao, Yang, et al. "Provably Convergent Federated Trilevel Learning." Proceedings of the AAAI Conference on Artificial Intelligence. Vol. 38. No. 11. 2024.

2. Qiu, Yuyang, Uday Shanbhag, and Farzad Yousefian. "Zeroth-order methods for nondifferentiable, nonconvex, and hierarchical federated optimization." Advances in Neural Information Processing Systems 36 (2023).

3. Liu, Sijia, et al. "A primer on zeroth-order optimization in signal processing and machine learning: Principals, recent advances, and applications." IEEE Signal Processing Magazine 37.5 (2020): 43-54.

4. Ghadimi, Saeed, and Guanghui Lan. "Stochastic first-and zeroth-order methods for nonconvex stochastic programming." SIAM journal on optimization 23.4 (2013): 2341-2368.

5. W. Fang, Z. Yu, Y. Jiang, Y. Shi, C. N. Jones, and Y. Zhou. Communication-efficient stochastic zeroth-order optimization for federated learning. IEEE Transactions on Signal Processing, 70, 2022

6. Y. Qiu, U. Shanbhag, and F. Yousefian. Zeroth-order methods for nondifferentiable, nonconvex, and hierarchical federated optimization. Advances in Neural Information Processing Systems, 36, 2023.

---

> ### Author Response · Authors · 2024-11-21
> **Author Rebuttal: Part 1**
>
> **General Reply:** We sincerely appreciate the time and effort you have dedicated to reviewing our work. Below, we provide a point-by-point response to your questions. Please let us know if you have any additional questions or concerns—we would be more than happy to address any further questions and concerns.
>
> **(W1)** My major concern is related to the novelty of the DTZO algorithm as it combines the ideas from two previous papers. Indeed, the general structure of the algorithm seems to be borrowed from the recent paper [4]. The main difference is that the authors do not use gradients of functions, which are unavailable, but well-known approximations to them, following [3]. However, this modification does not seem very significant for me.
>
> **(Reply to W1)** Thank you for your insightful comments. Trilevel optimization has found diverse applications in various machine learning tasks, ranging from robust hyperparameter optimization to domain adaptation. Existing approaches to trilevel optimization primarily address scenarios where first-order information is available at each level of the optimization problem. However, the study of trilevel optimization in settings where first-order information is unavailable remains **under-explored**. These scenarios are increasingly $\textit{important}$ given the widespread use of black-box models in machine learning, e.g., in many practical scenarios involving LLMs, access to first-order information is often restricted due to the proprietary nature of these models or API constraints (commercial LLM APIs), which typically only support input-output interactions without providing gradient visibility. Compared to single-level and bilevel optimization problems, trilevel optimization problems are significantly harder to solve [1]; in fact, even finding a feasible solution in a linear trilevel learning problem is **NP-hard** [2]. In this work, we consider how to address the trilevel optimization problems when first-order information is unavailable. To our best knowledge, this is the **first work** to address the trilevel zeroth order optimization problems. Please note that simply combining the existing algorithms [3, 4] **can not** achieve the goal. In the proposed algorithm, a $\textit{novel}$ cutting plane, i.e., zeroth order cut, is proposed, which can construct the cascaded polynomial relaxation without relying on first-order information. We demonstrate that the proposed zeroth order cuts are capable of constructing the **polynomial relaxation** for trilevel zeroth order optimization problems, and the relaxation will be gradually tightened as additional cuts are introduced (It should be noted that simply combining existing trilevel algorithms with zeroth-order gradient estimators **cannot achieve** this theoretical guarantee). To the best of our knowledge, this is the **first algorithm** to explore the generation of cutting planes without utilizing first-order information in multilevel optimization. In addition, we provided **theoretical guarantees** for iteration complexity and communication efficiency for the proposed algorithm, simply combining existing methods **can not** ensure these theoretical guarantees. Moreover, the proposed algorithm is $\textit{general}$ and adaptable to a **wide range** of trilevel optimization scenarios (e.g., trilevel optimization problems with partial zeroth order constraints) with slight adjustments. For instance, if second- or third-level gradients are available, first-order-based cuts [4] can be used to construct a cascaded polynomial approximation. When first-level gradients in TLL are accessible, gradient descent steps can replace the zeroth-order gradient estimators.

---

> ### Author Response · Authors · 2024-11-21
> **Author Rebuttal: Part 2**
>
> **(W2)** Moreover, I have concerns regarding the theoretical results. To prove the main of them, i.e., Theorem 1,2, the authors introduce $\mu$-smooth approximation of the final objective functional $F$. Then, similar to [4], the authors conduct the convergence rate analysis using this approximation. However, there is no theoretical result showing how close the solutions of this $F_{\mu}$ approximation are to those of the objective functional $F$? Without such a result, the convergence rate for F seems to be still an open question. I will appreciate it if the authors provide some comments on this topic.
>
> **(Reply to W2)** We appreciate your valuable comments. In the theoretical analyses, the iteration complexity and communication complexity are provided for function $F$ **instead of** $F_{\mu}$. Please note that the Stationarity Gap (in Definition1) and $\epsilon$-Stationary Point (in Definition2) are defined with respect to function $F$.  Moreover, more discussions about the smooth approximation $F_{\mu}$ are provided as suggested. $F_{\mu}$ is a $\mu$-smooth approximation of $F$. As shown in Definition 3 that $F_{\mu}(w)=\mathbb{E}[F(w+\mu u)]$, where $w \in \mathbb{R}^d$. The difference between  $F$ and  $F_{\mu}$ will decrease as $\mu$ decreases [3], i.e., for any $w$, it holds that:
>
> $|F_{\mu}(w)-F(w)| \le \mu^2 L d/2$ and $||\nabla F_{\mu}(w)-\nabla F(w)|| \le \mu L (d+3)^{\frac{3}{2}}/2$ .
>
> In addition, more discussions about $F$ and  $F_{\mu}$ are provided in the Reply to Q1 below.
>
> Moreover, it is worth noting that finding a feasible solution for a trilevel optimization problem is **NP-hard** [2], making it unlikely to develop a polynomial-time algorithm for the original trilevel optimization problem. In multilevel optimization, the lower-level problem often acts as a **soft constraint** for the upper-level problem, allowing for some degree of constraint violation while still achieving a feasible and meaningful solution [5, 6]. This property supports the acceptability of relaxing the trilevel optimization problem. Our theoretical analysis demonstrates that the proposed zeroth-order cuts effectively construct a **polynomial relaxation** for the trilevel optimization problem, with this relaxation **gradually tightening** as more zeroth-order cuts are applied.
>
> **(W3)** My minor comments are related to the comparison with the baselines in the numerical experiments. The authors tested the approach in the tasks related to LLM and hyperparameters robust optimization and compare its performance with several baseline models, e.g., [7] and [8]. Both of these algorithms provide the results for MSE or Loss vs iteration number and communication round, it would be great if the authors provide this kind of result too.
>
> **(Reply to W3)** We appreciate your helpful suggestions. Per your suggestions, we have provided additional experimental results (test loss vs iteration numbers) in the revised manuscript, please refer to page 37 in Appendix F.  For your convenience, the added contents are attached as follows.
>
> Following [8], the robustness in the proposed framework with respect to the choice of smoothing parameter $\mu$ is evaluated. The experiments are conducted on the robust hyperparameter optimization task under various setting of smoothing parameter, $\mu \in \\{0.01, 0.001, 0.0001\\}$. It is seen from Figure 4 and 5 that the proposed DTZO is robust to the choice of smoothing parameter $\mu$. In addition, we also note that the proposed DTZO has faster convergence rate with a relatively smaller $\mu$, because the gradient estimate improves when $\mu$ becomes relatively smaller, as discussed in [9].
>
> In addition, the impact of different choices of $T_1$ on the convergence rate within the proposed framework is evaluated. As illustrated in Figures 6 and 7, a smaller $T_1$ leads to faster convergence but affects the method's performance, resulting in a higher test loss. Conversely, if a better performance is required, a larger $T_1$ can be selected, corresponding to a more refined polynomial relaxation. In the proposed framework, we can $\textit{flexibly}$ adjust $T_1$ based on distributed system requirements. The results in Figures 6 and 7 are consistent with our theoretical analyses presented under Theorems 1 and 2.

---

> ### Author Response · Authors · 2024-11-21
> **Author Rebuttal: Part 3**
>
> **(Q1)** Can you comment on the connection between the solutions of $F$ and its $\mu$-smooth approximation $F_{\mu}$? Are they ϵ-close to each other?
>
> **(Reply to Q1)** Thanks for your questions. In this work, all the theoretical analyses are conducted based on $F$ **instead of** $F_{\mu}$. However, we are happy to provide some theoretical analyses about $F$ and $F_{\mu}$ during the ICLR discussion phase.
>
> Let $\mathcal{G}^t$ and $\mathcal{G}\_{\mu}^t$ denote the stationarity gaps with respect to functions $F$ and $F_{\mu}$, we have that,
>
> $$||\mathcal{G}^t- \mathcal{G}\_{\mu}^t||^2 \\ =\sum\_{i=1}^3\sum\_{j=1}^N||\nabla\_{x\_{i,j}}F - \nabla\_{x_{i,j}}F\_{\mu}||^2+\sum\_{i=1}^3||\nabla\_{z\_i}F - \nabla\_{z\_i}F\_{\mu}||^2\\ \le\mu^2 L^2 \sum\_{i=1}^3(N+1)(d\_i+3)^{3}/4 . \qquad \qquad (1) $$
>
> It is seen that $||\mathcal{G}^t- \mathcal{G}_{\mu}^t||^2$ is related to the setting of smoothing parameter $\mu$, i.e., it decreases as we set a smaller $\mu$.
>
> In addition, combining Eq. (1) with the setting of $0\le \mu \le \frac{1}{\sqrt{T(\epsilon)-T_1}} $ and $T(\epsilon) \ge \mathcal{O}(\frac{1}{\epsilon^2}+T_1)$, we can obtain that,
>
> $$||\mathcal{G}^t- \mathcal{G} \_{\mu}^t||^2 \\  \le \epsilon^2 L^2 \sum \_{i=1}^3(N+1)(d \_i+3)^{3}/4 \\ \sim \mathcal{O}(\epsilon^2).  \qquad \qquad (2)$$
>
> It is seen from Eq. (2) that the difference between the stationarity gaps with respect to functions $F$ and $F_{\mu}$ is bounded by $\mathcal{O}(\epsilon^2)$ when the $\epsilon$-stationary point is obtained.
>
>
>
> **(Q2)**  I noticed that the zeroth-order gradient estimation method used here is based on [3]. Could you explain why you choose this method over the approach in [9], which utilizes mini-batch sampling to reduce variance?
>
> **(Reply to Q2)** We sincerely appreciate your suggestion. The two-point estimate is utilized in the proposed framework due to its lower computational complexity in each iteration. Additionally, it is worth mentioning that the proposed framework is general, allowing for the integration of the mini-batch zeroth-order estimator. We have included further discussions on this topic in Appendix J2; please refer to page 43 for details. For your convenience, the added discussions are attached as follows.
>
> It is worth noting that the proposed framework is versatile, allowing for the integration of various gradient estimators. For instance, the mini-batch sampling-based gradient estimator [9, 10, 11] can be employed to replace the two-point gradient estimator, reducing variance. Specifically, with mini-batch sampling, Eq. (10), (12), (19), (20), and (21) can be replaced by the following multi-point gradient estimators.
>
> $$\begin{equation}
> \begin{array}{l}
>      {G_\mu^{\rm{in}} }( \{{\boldsymbol{x}\_{3,j}^t}\}, \boldsymbol{z}_1^t, {\boldsymbol{z}_2^t}', \boldsymbol{z}_3^t) \\
>      = \frac{1}{{\mu}} \sum\limits\_{p=1}^b  [\phi\_{\rm{in}}( \{{\boldsymbol{x}\_{3,j}^t} + \mu \boldsymbol{\mu}\_{x\_{3,j}}^p\}, \boldsymbol{z}_1^t + \mu \boldsymbol{\mu}\_{z_1}^p, {\boldsymbol{z}_2^t}' + \mu \boldsymbol{\mu}\_{z_2}^p, \boldsymbol{z}_3^t + \mu \boldsymbol{\mu}\_{z_3}^p)   -   \phi\_{\rm{in}}( \{{\boldsymbol{x}\_{3,j}^t}\}, \boldsymbol{z}_1^t, {\boldsymbol{z}_2^t}', \boldsymbol{z}_3^t)  \boldsymbol{\mu}^{{\rm{in}},p}],
> \end{array}
> \end{equation}$$
>
> $$\begin{equation}
> \begin{array}{l}
>     {G_\mu^{\rm{out}} }(\{{\boldsymbol{x}\_{2,j}^t}\}, \{{\boldsymbol{x}\_{3,j}^t}\},  \boldsymbol{z}_1^t, \boldsymbol{z}_2^t, \boldsymbol{z}_3^t) \\
>     = \frac{1}{\mu} \sum\limits\_{p=1}^b  [ \phi\_{\rm{out}}(\{{\boldsymbol{x}\_{2,j}^t} + \mu \boldsymbol{\mu}\_{x\_{2,j}}^p\}, \{{\boldsymbol{x}\_{3,j}^t} + \mu \boldsymbol{\mu}\_{x\_{3,j}}^p\}, \boldsymbol{z}_1^t + \mu \boldsymbol{\mu}\_{z_1}^p, \boldsymbol{z}_2^t + \mu \boldsymbol{\mu}\_{z_2}^p, \boldsymbol{z}_3^t + \mu \boldsymbol{\mu}\_{z_3}^p) -   \phi\_{\rm{out}}(\{{\boldsymbol{x}\_{2,j}^t}\}, \{{\boldsymbol{x}\_{3,j}^t}\}, \boldsymbol{z}_1^t, \boldsymbol{z}_2^t, \boldsymbol{z}_3^t)\boldsymbol{\mu}^{{\rm{out}},p}],
> \end{array}
> \end{equation}$$
>
> $$\begin{equation}
> \begin{array}{l}
>   G\_{\boldsymbol{x}\_{1,j}}(\{{\boldsymbol{x}\_{1,j}^t}\}, \{{\boldsymbol{x}\_{2,j}^t}\}, \{{\boldsymbol{x}\_{3,j}^t}\}, \boldsymbol{z}_1^t, \boldsymbol{z}_2^t, \boldsymbol{z}_3^t)
>    \\   =\frac{1}{{\mu }} \sum\limits\_{p=1}^b  [{{{f\_{1,j}}({\boldsymbol{x}\_{1,j}^t} + \mu {\boldsymbol{u}\_{k,1}^p},{\boldsymbol{x}\_{2,j}^t},{\boldsymbol{x}\_{3,j}^t}) -{f\_{1,j}}({\boldsymbol{x}\_{1,j}^t},{\boldsymbol{x}\_{2,j}^t},{\boldsymbol{x}\_{3,j}^t})}} {\boldsymbol{u}\_{k,1}^p}]+  2\phi_j (\boldsymbol{x}\_{1,j}^t - \boldsymbol{z}_1^t),
> \end{array}
> \end{equation}$$

---

> ### Author Response · Authors · 2024-11-21
> **Author Rebuttal: Part 4**
>
> $$\begin{equation}
>  \begin{array}{l}
>   G \_{\boldsymbol{x} \_{2,j}}(\{{\boldsymbol{x} \_{1,j}^t}\}, \{{\boldsymbol{x} \_{2,j}^t}\}, \{{\boldsymbol{x} \_{3,j}^t}\}, \boldsymbol{z}_1^t, \boldsymbol{z}_2^t, \boldsymbol{z}_3^t) = \nabla \_{\boldsymbol{x} \_{2,j}}  o( \{{\boldsymbol{x} \_{2,j}^t}\},  \{{\boldsymbol{x} \_{3,j}^t}\},  \boldsymbol{z}_1^t, \boldsymbol{z}_2^t, \boldsymbol{z}_3^t)  +  \frac{1}{{\mu }} \sum\limits \_{p=1}^b  [{{{f \_{1,j}}({\boldsymbol{x} \_{1,j}^t},{\boldsymbol{x} \_{2,j}^t} + \mu {\boldsymbol{u} \_{k,2}^p},{\boldsymbol{x} \_{3,j}^t}) -{f \_{1,j}}({\boldsymbol{x} \_{1,j}^t},{\boldsymbol{x} \_{2,j}^t},{\boldsymbol{x} \_{3,j}^t})}} {\boldsymbol{u} \_{k,2}^p}],
> \end{array}
> \end{equation}$$
>
> $$\begin{equation}
> \begin{array}{l}
>   G \_{\boldsymbol{x} \_{3,j}}(\{{\boldsymbol{x} \_{1,j}^t}\}, \{{\boldsymbol{x} \_{2,j}^t}\}, \{{\boldsymbol{x} \_{3,j}^t}\}, \boldsymbol{z}_1^t, \boldsymbol{z}_2^t, \boldsymbol{z}_3^t) =  \nabla \_{\boldsymbol{x} \_{3,j}} o( \{{\boldsymbol{x} \_{2,j}^t}\},  \{{\boldsymbol{x} \_{3,j}^t}\},  \boldsymbol{z}_1^t, \boldsymbol{z}_2^t, \boldsymbol{z}_3^t) + \frac{1}{{\mu }} \sum\limits \_{p=1}^b  [ {{{f \_{1,j}}({\boldsymbol{x} \_{1,j}^t},{\boldsymbol{x} \_{2,j}^t},{\boldsymbol{x} \_{3,j}^t} + \mu {\boldsymbol{u} \_{k,3}^p}) -{f \_{1,j}}({\boldsymbol{x} \_{1,j}^t},{\boldsymbol{x} \_{2,j}^t},{\boldsymbol{x} \_{3,j}^t})}} {\boldsymbol{u} \_{k,3}^p}],
> \end{array}
> \end{equation}$$
>
> where $\boldsymbol{\mu}^{{\rm{in}},p}=[\{\boldsymbol{\mu}\_{x_{3,j}}^p\}, \boldsymbol{\mu}\_{z_1}^p, \boldsymbol{\mu}\_{z_2}^p, \boldsymbol{\mu}\_{z_3}^p]$, $\boldsymbol{\mu}^{{\rm{out}},p}=[\{\boldsymbol{\mu}\_{x_{2,j}}^p\},\{\boldsymbol{\mu}\_{x_{3,j}}^p\}, \boldsymbol{\mu}\_{z_1}^p, \boldsymbol{\mu}\_{z_2}^p, \boldsymbol{\mu}\_{z_3}^p]$,  $\boldsymbol{u}\_{k,1}^p$, $ \boldsymbol{u}\_{k,2}^p$, $ \boldsymbol{u}\_{k,3}^p, p=1,\cdots b$ are drawn from $\mathcal{N}(0, {\bf{I}})$, and $b$ represents the number of samples used in the multi-point gradient estimator.
>
>
>
> **Minor**: line 328: equation number (16) is missing.
>
> (Reply to minor) Thanks for your comments. Corrected as suggested.
>
>
> **Reference**
>
> [1] The computational complexity of multi-level linear programs
>
> [2] A review on bilevel optimization: From classical to evolutionary approaches and applications
>
> [3] Stochastic first-and zeroth-order methods for nonconvex stochastic programming
>
> [4] Provably convergent federated trilevel learning
>
> [5] A general stochastic approach to solving problems with hard and soft constraints.
>
> [6] Using hard constraints for representing soft constraints
>
> [7] Communication-efficient stochastic zeroth-order optimization for federated learning
>
> [8] Zeroth-order methods for nondifferentiable, nonconvex, and hierarchical federated optimization
>
> [9] A primer on zeroth-order optimization in signal processing and machine learning: Principals, recent advances, and applications
>
> [10] Optimal rates for zero-order convex optimization: The power of two function evaluations
>
> [11] Zeroth-order online ADMM: Convergence analysis and applications

---

> ### Author Response · Authors · 2024-11-26
> **We are happy to address any further concerns**
>
> We sincerely thank you for raising the concerns in the initial reviews. We have tried our best efforts to clarify those concerns in the responses. Given the limited time for discussion, we would really appreciate it if you could let us know in case there is any additional concern.

---

> > ### Comment · Reviewer_C3cz · 2024-11-27
> > **Answer to the Authors**
> >
> > Thank you for additional clarifications. I remain concerned about the novelty of your algorithm and the proposed zero-order cuts. But since my other concerns are addressed, I increase my score.

---

> ### Author Response · Authors · 2024-11-27
> **Thank you for your feedback: Part 1**
>
> Thank you for your feedback, we sincerely appreciate it! We provide a detailed clarification below regarding the novelty of the proposed zeroth order cut and the proposed algorithm.
>
> - The Novelty of The Proposed Zeroth Order Cut
>
> This work is the first to explore the use of cutting plane methods in zeroth order optimization. We theoretically demonstrate that the proposed zeroth order cuts are capable of constructing the **cascaded polynomial relaxation** without relying on first-order information, and this relaxation will be gradually tightened as additional cuts are introduced. Additionally, it is worth mentioning that the proposed zeroth order cuts **do not require the convexity** of the function and are also the **first non-linear cuts** in multilevel optimization. Compared to linear cutting planes, nonlinear cuts usually offer better approximation capabilities for complex functions [5], providing new insights for the further development of cutting plane methods in multilevel optimization. Please note that simply combining the existing algorithms can not achieve this goal. To further highlight the novelty of the proposed zeroth order cut, we compare it with existing cutting plane methods used in multilevel optimization.
>
>
>
> | cutting plane                 | Convex       | Non-convex   | Non-linear   | Gradient free |
> | ----------------------------- | ------------ | ------------ | ------------ | ------------- |
> | [1]                           | $ \quad \checkmark$ |              |              |               |
> | [2]                           | $ \quad \checkmark$ |              |              |               |
> | [3]                           | $ \quad \checkmark$ |              |              |               |
> | [4]                           | $ \quad \checkmark$ | $ \\; \\; \\; \quad \checkmark$ |              |               |
> | The proposed Zeroth Order Cut | $ \quad \checkmark$ | $ \\; \\; \\;  \quad \checkmark$ | $  \\; \\;  \quad \checkmark$ |$ \qquad \checkmark$  |
>
> **Reference**
>
> [1] Asynchronous Distributed Bilevel Optimization, ICLR 2023
>
> [2] Robust Beamforming for Downlink Multi-Cell Systems: A Bilevel Optimization Perspective, AAAI 2024
>
> [3] Tri-Level Navigator: LLM-Empowered Tri-Level Learning for Time Series OOD Generalization, NeurIPS 2024
>
> [4] Provably Convergent Federated Trilevel Learning, AAAI 2024
>
> [5] Temlyakov. "Nonlinear methods of approximation." Foundations of Computational Mathematics

---

> ### Author Response · Authors · 2024-11-27
> **Thank you for your feedback: Part 2**
>
> - The Novelty of The Proposed Algorithm
>
> Trilevel zeroth order optimization is significantly important given the widespread use of black-box models in machine learning, such as commercial LLMs APIs.  The study of trilevel zeroth order optimization remains **under-explored** and **challenging**, especially since finding a feasible solution in a linear trilevel learning problem is **NP-hard**. To the best of our knowledge, this work takes an **initial step** in addressing trilevel zeroth order optimization problems while providing theoretical guarantees, such as iteration complexity and communication complexity. Notably, simply combining existing algorithms **cannot** achieve these theoretical guarantees. Moreover, this algorithm is the **first to explore** the use of cutting plane methods in zeroth order optimization. We theoretically demonstrate that the proposed zeroth order cuts are capable of constructing the **cascaded polynomial relaxation** without relying on first-order information, and this relaxation will be gradually tightened as additional cuts are introduced. Simply combining the existing algorithms can not ensure these theoretical guarantees. In addition, in the proposed algorithm, we explore the idea that the lower-level optimization problem can often serve as the **soft constraint** for the upper-level optimization problem in multilevel optimization. This work represents the **first introduction** of the concept of soft constraints into multilevel optimization, offering a novel perspective for this field.
>
> We sincerely appreciate the time and effort you have devoted to reviewing our work, and we hope our responses adequately address your concerns. Thank you once again!

---

### Author Response · Authors · 2024-11-22
**We are happy to address any further concerns**

Dear Reviewers and AC,

We sincerely appreciate the time and effort you have dedicated to reviewing and handling our work. The rebuttals and the revised manuscript have been provided. In the revised manuscript, the revised contents are highlighted in blue. We hope our responses have sufficiently addressed all your concerns, and we would be happy to address any additional questions or comments you may have.


Best regards,

Authors #13705

---

### Meta-Review · Area_Chair_XQ5a · 2024-12-18

**Metareview:**

This paper proposes a distributed trilevel zeroth-order (DTZO) algorithm to tackle trilevel optimization problems in scenarios where gradients or subgradients of the level objectives are unavailable. The algorithm is supported by a theoretical analysis of iteration and communication complexities, along with experimental evaluations on tasks related to large language models and robust hyperparameter optimization.

The proposed method employs smoothing techniques to replace gradient oracles with finite-difference estimators. A key technical contribution is the introduction of zeroth-order cuts, which enable the construction of cascaded polynomial relaxations without relying on first-order information.

The reviewers agree that trilevel optimization problems are underexplored and recognize the value of the presented results in advancing the state of the art. However, they debated the novelty of the techniques, noting that smoothing is a well-established approach for replacing first-order oracles with zeroth-order information. As a result, the advancement over prior work, such as Jiao et al. (2024), appears limited.

While the paper introduces advanced concepts, including a distributed implementation of the method, the reviewers found that it lacks sufficient discussion of the tightness of the results and the algorithm's sensitivity to parameter choices. This limits the overall novelty and insight provided by the work.

**Additional Comments On Reviewer Discussion:**

The discussion between the authors and the reviewers helped clarify several technical questions.

Reviewer g4re raised a concern about Assumption 1 (boundedness of the iterates), stating that it appears to be a very strong assumption. The authors responded by arguing that this assumption is common in related literature. No consensus was reached during the discussion. Both perspectives seem reasonable, and I do not believe the reviewer's opinion violates the ICLR review guidelines.

---

### Decision · Program_Chairs · 2025-01-22

Reject